# Retrievals of ice microphysical properties using dual-wavelength polarimetric radar observations during stratiform precipitation events

Eleni Tetoni[1], Florian Ewald[1], Martin Hagen[1], Gregor Köcher[2], Tobias Zinner[2], and Silke Groß[1]

[1]Deutsches Zentrum für Luft- und Raumfahrt (DLR), Institut für Physik der Atmosphäre, Oberpfaffenhofen, Germany
[2]Meteorologisches Institut, Ludwig-Maximilians-Universität, Munich, Germany

*Correspondence to*: Eleni Tetoni (Eleni.Tetoni@dlr.de)

**Abstract.** Ice growth processes within clouds affect the type as well as the amount of precipitation. Hence, the importance of an accurate representation of ice microphysics in numerical weather and numerical climate models has been confirmed by several studies. To better constrain ice processes in models, we need to study ice cloud regions before and during monitored precipitation events. For this purpose, two radar instruments facing each other were used to collect complementary measurements. The C-band POLDIRAD weather radar from the German Aerospace Center (DLR), Oberpfaffenhofen and the Ka-band MIRA-35 cloud radar from the Ludwig Maximilians University of Munich (LMU) were used to monitor stratiform precipitation in the vertical cross-section area between both instruments. The logarithmic difference of radar reflectivities at two different wavelengths (54.5 and 8.5 mm), known as dual-wavelength ratio, was exploited to provide information about the size of the detected ice hydrometeors, taking advantage of the different scattering behavior in the Rayleigh and Mie regime. Along with the dual-wavelength ratio, differential radar reflectivity measurements from POLDIRAD provided information about the apparent shape of the detected ice hydrometeors. Scattering simulations using the T-matrix method were performed for oblate and horizontally aligned prolate ice spheroids of varying shape and size using a realistic particle size distribution and a well-established mass-size relationship. The combination of dual-wavelength ratio, radar reflectivity and differential radar reflectivity measurements as well as scattering simulations was used for the development of a novel retrieval for ice cloud microphysics. The development of the retrieval scheme also comprised a method to estimate the hydrometeor attenuation in both radar bands. To demonstrate this approach, a feasibility study was conducted on three stratiform snow events which were monitored over Munich in January 2019. The ice retrieval can provide ice particle shape, size and mass information which is in line with differential radar reflectivity, dual-wavelength ratio and radar reflectivity observations, respectively, when the ice spheroids are assumed to be oblates and to follow the mass-size relation of aggregates. When combining two spatially-separated radars to retrieve ice microphysics, the beam width mismatch can locally lead to significant uncertainties. However, the calibration uncertainty is found to cause the largest bias for the averaged retrieved size and mass. Moreover, the shape assumption is found to be equally important to the calibration uncertainty for the for retrieved size, while it is less important than the calibration uncertainty for the retrieval of ice mass. A furthermore finding is the importance of the differential radar reflectivity for the particle size retrieval directly above the MIRA-35 cloud radar. Especially for that observation geometry, the simultaneous slantwise observation from the polarimetric weather radar POLDIRAD can reduce ambiguities in retrieval of the ice particle size by constraining the ice particle shape.

## 1 Introduction

The ice phase is the predominant cloud phase at mid and higher latitudes (Field and Heymsfield, 2015). Ice clouds are known to reflect the shortwave, incoming solar radiation, but they can also trap the longwave, terrestrial radiation interfering with the Earth's energy budget (Liou, 1986). Their influence on the radiation budget of the climate system strongly depends on their top height as well as on ice crystals habits and effective ice crystal size (Zhang et al., 2002). Ice growth processes such

as deposition, riming and aggregation, play a leading role in the formation of precipitation and are a central topic in many ice cloud studies. A misrepresentation of these processes in numerical weather models can lead to high uncertainties and therefore, they need to be constrained as accurately as possible. Brdar and Seifert (2018) presented the novel Monte-Carlo microphysical model, McSnow, aiming for a better representation of aggregation and riming processes of ice particles. When numerical weather models are used to predict microphysics information about ice hydrometeors (e.g., Predicted Particle Properties (P3), Part I, Morrison and Milbrandt, 2015), we need to investigate under which conditions each ice growth process occurs. To better understand these mechanisms and improve their representation in models, more precise microphysics information (e.g., size, shape and mass) through ice retrievals based on measurements, is needed.

Many studies showed how millimeter-wave radar measurements can be used to retrieve ice water content (IWC) profiles in clouds (e.g., Hogan et al., 2006). However, stand-alone single-frequency radar measurements cannot constrain microphysical properties such as ice particle size and shape simultaneously without using empirical relations. Dealing with more parameters (e.g., IWC, size and shape) more measurements are needed. Thus, observations or simulated radar parameters are often combined with other remote sensing instruments, e.g., with lidars, to retrieve microphysics properties such as the effective radius of cloud ice particles (Cazenave et al., 2019), or with infrared radiometers (Matrosov et al., 1994) to retrieve the median diameter of the ice particles size distribution. Another way to gain microphysics information is to use multi-frequency radar observations (described in detail in Sect. 1.1) as they exploit the scattering properties of ice particles in both Rayleigh and non-Rayleigh regime. To this end, frequencies are chosen with respect to the prevalent ice particle size. In the case of dual-frequency techniques, one frequency is chosen to be in the Rayleigh regime (e.g., S-, C- or X-band), where particle size is much smaller than the radar wavelength, and the other is chosen to be in the Mie regime (e.g., Ka-, Ku- or W-band), where particle size is comparable or larger than the radar wavelength (e.g., Matrosov 1998, Hogan et al., 2000 and many more). The scattering of radar waves is sensitive to the size and number concentration of particles. The radar reflectivity factor $z$ is defined as the sixth moment of the particle size distribution $N(D)$ and is thus designed to be proportional to the Rayleigh scattering cross section of small – size much smaller comparing to the radar wavelength – liquid spheres:

$$z \text{ [mm}^6 \text{ m}^{-3}] = \int_0^\infty N(D)D^6 \mathrm{d}D \tag{1a}$$

where,

$z$: the radar reflectivity in linear scale,

$N(D)$: the number concentration,

$D$: the geometric diameter of the particles.

This formula can be also expressed in logarithmic terms:

$$Z \text{ [dBZ]} = 10\log_{10}(z). \tag{1b}$$

This definition, however, cannot be directly applied to snow due to the varying density, the irregular shape and larger size of ice particles which cause deviations from the Rayleigh into the Mie scattering regime. Moreover, $N(D)$ for ice particles is referring to the size distribution of their melted diameters. Nevertheless, an equivalent radar reflectivity factor $Z_e$ can be derived from the measured radar reflectivity $\eta$ ($\eta = \sum_{Vol} \sigma_n$; normalized to a specific volume summation of backscattering cross-section, $\sigma_n$, of all detected hydrometeors) when the dielectric factor of water $|K|^2 = 0.93$ is assumed:

$$z_e \text{ [mm}^6 \text{ m}^{-3}] = \eta \frac{\lambda^4}{\pi^5 |K|^2} \text{ and } Z_e \text{ [dBZ]} = 10\log_{10}(z_e) \tag{1c}$$

where,

$\lambda$: the radar wavelength. In the Rayleigh regime, the radar reflectivity factor $Z$ or the equivalent radar reflectivity factor $Z_e$ (for simplicity reasons referred also as radar reflectivity in this paper) is proportional to the sixth power of the particle size, while in the Mie regime $Z_e$ scales with the second power of the particle size. In both regimes $Z_e$ scales linearly with the particle number concentration.

**1.1 Size and shape microphysics retrievals**

Using the ratio of radar reflectivities at two different radar wavelengths (Eq. 2; dual-wavelength ratio, DWR), we can infer size information about hydrometeors observed within the radar beams. This parameter increases with the particle size when the shorter radar wavelength is equal or shorter than the particle size:

$$\mathrm{DWR}_{\lambda_1,\lambda_2}\,[\mathrm{dB}] = 10\log_{10}\left(\frac{z_{e,\lambda_1}}{z_{e,\lambda_2}}\right) \text{ or } \mathrm{DWR}_{\lambda_1,\lambda_2}\,[\mathrm{dB}] = Z_{e,\lambda_1}\,[\mathrm{dBZ}] - Z_{e,\lambda_2}\,[\mathrm{dBZ}]. \tag{2}$$

In Eq. (2), $\lambda_1 > \lambda_2$ are the two radar wavelengths, $z_{e,\lambda_1}$, $z_{e,\lambda_2}$ the radar reflectivities at the two radar wavelengths in linear scale (units: $\mathrm{mm}^6\,\mathrm{m}^{-3}$) and $Z_{e,\lambda_1}$, $Z_{e,\lambda_2}$ the radar reflectivities in logarithmic scale (units: dBZ). Recent studies (e.g., Trömel et al., 2021) have underlined that multi-wavelength (also known as multi-frequency) measurements should be combined with other types of radar observations, e.g., polarimetric variables or Doppler velocity to improve our understanding of ice microphysics. For ice particle density, in particular, DWR provides only limited information, while Doppler velocity measurements can better

constrain the particle density as the fall speed is strongly connected to it. Specifically, Mason et al. (2018) used vertically pointing Ka- and W-band cloud radars to combine DWR and Doppler measurements to provide information about the particle size distribution (PSD) and an ice particles density factor which is connected to ice particles shape and mass, but also terminal velocity and backscatter cross-section. However, the DWR approach has been widely used in many studies in the past providing microphysics information without Doppler velocity measurements. In particular, the DWR method has been used in

ice studies to estimate the snowfall rate $R$ or for the Quantitative Precipitation Estimation (QPE). Matrosov (1998) developed a DWR method to estimate $R$, supplementing experimental $Z_e$-R relations with a retrieved median size. In other studies, such as Hogan and Illingworth (1999) and Hogan et al. (2000), DWR from airborne and ground-based radars was used to obtain information about ice crystals sizes as well as IWC for cirrus clouds. In recent years, the combination of multiple DWR measurements has been explored to provide more microphysics information, e.g., ice particles habits or density. Kneifel et al.

(2015) developed a triple-frequency method ($\mathrm{DWR}_{X,Ka}$ and $\mathrm{DWR}_{Ka,W}$) to derive ice particle habits information from three snowfall events measured during the Biogenic Aerosols Effects on Clouds and Climate (BAECC) field campaign (Petäjä et al., 2016). The triple-frequency method was also used by Leinonen et al. (2018b) to develop an algorithm that retrieves ice particle size and density as well as number concentration using airborne radar data from the Olympic Mountains Experiment (OLYMPEX, Houze et al., 2017). In Mason et al. (2019), the PSD and morphology of ice particles were thoroughly explored

using the triple-frequency method to improve ice particle parameterizations in numerical weather prediction models. In the same study, it was also found that for heavily rimed ice particles, the triple-frequency radar observations can constrain the shape parameter $\mu$ of the PSD. Recently, Mroz et al. (2021) used single-frequency (X-band), triple-frequency radar measurements (X-, Ka-, W-band) as well as triple-frequency combined with Doppler velocity radar measurements to develop different versions of an algorithm that retrieves the mean mass-weighted particle size, IWC and the degree of riming. The

multi-frequency versions of the algorithm retrieved IWC with lower uncertainties compared to the single-frequency version. Additionally, with the multi-frequency approaches, the algorithm was also able to provide ice particle density information as well as mean mass-weighted diameter information for larger snowflakes in contrast to single-frequency approach which could only constrain the mean mass-weighted diameter for snowflakes up to 3 mm size. Overall, the multi-frequency versions of the algorithm performed better as the retrieved parameters agreed better with in-situ measurements.

Beyond multi-frequency techniques, ice microphysics information can be obtained from polarimetric radar measurements. In previous studies, polarimetry was commonly used for snowfall rate estimation. Bukovčić et al. (2018), for instance, used

polarimetric radar variables to study the IWC and the resulting snow water equivalent rate. Besides these precipitation rate studies, polarimetry is an advantageous tool to obtain information about the size distribution and the shape of ice particles. Additional characteristics, like the particle orientation and their canting angle distribution, as well as the variable refractive index of melting or rimed ice crystals have a further influence on polarimetric radar signals. To untangle some of these particle properties, polarimetric weather radars can provide several parameters such as differential radar reflectivity (ZDR), linear depolarization ratio (LDR), reflectivity difference (ZDP), cross-correlation coefficient (ρHV), differential propagation phase (φDP) and specific differential phase (KDP). A description of the aforementioned polarimetric radar variables can be found in Straka et al. (2000) and Kumjian (2013). The different sensitivities of these parameters have been widely used in classification schemes of atmospheric hydrometeors. Höller et al. (1994) developed one of the first algorithms to distinguish between rain, hail, single or multi-cells using ZDR, LDR, KDP and ρHV measurements during the evolution of a thunderstorm while moving from the west towards southern Germany. Subsequently, this algorithm was extended to estimate hydrometeor mass concentrations (Höller, 1995). Later, Straka et al. (2000) summarized the characteristics of different hydrometeors types depending on their radar signatures at a wavelength of 10 cm. One prominent polarimetric parameter in ice microphysics studies is known to be ZDR, a parameter which is defined as:

$$\text{ZDR [dB]} = 10\log_{10}\left(\frac{z_H}{z_V}\right),$$
(3)

where,

$z_H$: the signal received or reflectivity factor at horizontal polarization,

$z_V$: the signal received or reflectivity factor at vertical polarization. Following the definition of ZDR, it is zero if the received signal in both polarization states is the same, i.e., for spherical targets. For elongated, azimuthally oriented particles ZDR is found to be greater (oblate particles) or less than zero (vertically aligned prolates), depending on the orientation of their rotational axis to the horizontal polarization state (e.g., Straka et al., 2000). In Moisseev et al. (2015), ZDR along with KDP has been used to investigate growth processes of snow and their signatures on dual-polarization and Doppler velocity radar observations. Later on, Tiira and Moisseev (2020) exploited vertical profiles of ZDR combined with KDP and $Z_e$ for the development of an unsupervised classification of snow and ice crystals particles. In that study, the most important growth processes of ice particles were studied using several years of the Ikaalinen C-band radar data, in Hyytiälä forestry station in Juupajoki, Finland.

Although the size of atmospheric hydrometeors is strongly correlated to DWR, many studies have shown that DWR is also sensitive to the shape of ice hydrometeors. This sensitivity of DWR to shape was shown in e.g., Matrosov et al. (2005), where they estimated the increased uncertainty in particle size retrievals when the particles are assumed to be spherical only. One solution to that problem was offered by Matrosov et al. (2019), who stated that the shape of ice hydrometeors can be disentangled from DWR by studying the effect of radar elevation angle on DWR. Non-spherical ice hydrometeors should show a strong influence of elevation angle on DWR compared to spherical ice particles. Besides this scanning approach, the combination with polarimetry from collocated or nearby radar instruments could offer a promising solution to disentangle the contribution of size and shape in DWR measurements. While the shape can be constrained by ZDR measurements, the size of the detected particles can be determined using DWR.

**1.2 Representation of ice atmospheric hydrometeors using spheroids**

Single scattering simulations are an indispensable tool to bridge the gap between microphysical properties of hydrometeors and polarimetric radar observations. In the case of ice particles, the calculation of scattering properties can be challenging due to their large complexity, variety in shape, structure, size and density. One of the most sophisticated methods, the Discrete-Dipole Approximation (DDA; Draine and Flatau, 1994), can be used to calculate the scattering properties of realistic ice

crystals and aggregates. However, this approximation can be computational demanding. To reduce computation cost and complexity, ice particles are often assumed to be spheres and their scattering properties are calculated using the Mie theory or they are assumed to be spheroids and their scattering properties are calculated using the T-Matrix method (Waterman, 1965) or the Self-Similar Rayleigh-Gans Approximation (SSRGA; e.g., Hogan and Westbrook, 2014; Hogan et al., 2017; Leinonen et al., 2018a). The SSRGA was developed to consider the distribution of the ice mass throughout the particle's volume in scattering simulations. As we aim for a simple ice particle model, we extensively used the T-Matrix method in this study, assuming the ice particles to be *soft spheroids*. It is a common approach in model studies that ice particles are represented by homogeneous spheroids with density equal or smaller of bulk ice. Due to its simplicity, the limitations of the spheroid approximation have been a heavily researched and debated topic in the last decade. While Tyynelä et al. (2011) showed an underestimation of the backscattering for large snowflakes, Hogan et al. (2012) suggested that horizontally aligned oblate spheroids with a sphericity ($S$; minor to major axis ratio) of 0.6 can reliably reproduce the scattering properties of realistic ice aggregates which are smaller than the radar wavelength. The same study also concluded that, spheroids are more suitable to represent larger particles (maximum diameter up to 2.5mm) in simulations, rather than Mie spheres can, as the latter can lead to a strong overestimation of $Z_e$. Leinonen et al. (2012) on the other hand showed that the spheroidal model cannot always explain the radar measurements as more sophisticated particle models do, e.g., snowflake models. Later on, Hogan and Westbrook (2014) indicated that the soft spheroid approximation underestimates the backscattered signal of large snowflakes (1 cm size) – measured with a 94 GHz radar – up to 40 and 100 times for vertical and horizontal incidence, respectively. In contrast, the simple spheroidal particle model could successfully explain measurements of slant-45° linear depolarization ratio, SLDR, as well as SLDR patterns on the elevation angles (Matrosov, 2015) during the Storm Peak Laboratory Cloud Property Validation Experiment (StormVEx). In Liao et al. (2016) it was found that randomly oriented oblate ice spheroids could reproduce scattering properties in Ku- and Ka-band similar to these from scattering databases when large particles were assumed to have a density of $0.2 \text{ g cm}^{-3}$ and a maximum size up to 6 mm. Although Schrom and Kumjian (2018) showed that homogeneous reduced-density ice spheroids or plates cannot generally represent the scattering properties of branched planar crystals, the simple spheroidal model has been used in recent studies to represent ice aggregates as in Jiang et al. (2019), to simulate DWR for snow rate estimation studies as in Huang et al. (2019) or to retrieve shape from LDR as in Matrosov (2020). In all these studies, it is recognized that the spheroidal model requires less assumed parameters compared to more complex particle models.

Although more complex ice particle and scattering models are available, this work will use the soft spheroid approximation out of the following reasons: (1) In this work we aim to provide a feasibility study to combine two spatially separated radars to better constrain the ice crystal shape in microphysical retrievals using simultaneous DWR and ZDR observations from an oblique angle. Besides instrument coordination, the actual measurements and the assessment of measurement errors, the ice crystal and scattering model are just one component. Due to its simple and versatile setup, this work will utilize the soft spheroid approximation to study the benefit of additional ZDR measurements and the role of the observation geometry. (2) More importantly, to our knowledge, the more accurate SSRGA described by Hogan and Westbrook (2014) does not (yet) provide polarimetric variables used in this study, namely the ZDR. (3) In anticipation of a prognostic aspect ratio of ice crystals in bulk microphysical models (e.g., the adaptive habit prediction; Harrington et al., 2013), we aim to keep a minimal set of degrees of freedom to remain comparable with these modelling efforts. (4) Using ice spheroids, we are able to vary parameters such as median size, aspect ratio and ice water content independently, which serve as degrees of freedom of the ice spheroids, and calculate their scattering properties without much computational cost as in other scattering algorithms (e.g., DDA) that are used in more realistic ice crystal shapes simulations. Due to the independent parameters describing a spheroid, we can better study the dependence between each variable and the forward-simulated radar variables.

Due to this simplification, this study will focus on the feasibility to combine DWR and ZDR from spatially separated radar instruments into a common retrieval framework. Due to the missing internal structure and the near-field ice dipoles interactions

of soft spheroids, the known underestimation of the radar backscatter and generally lower ZDR for larger snowflakes will limit this study to ice aggregates with sizes in the millimeter regime. This will include the onset of ice aggregation within clouds above the melting layer (ML) but will exclude heavy snowfall close to the ground. Anyhow, this region is rarely included in the measurement region with an overlap between the two scanning radar instruments.

**1.3 Scientific objective and outline of this study**

Although vertically pointing radars are useful for Doppler spectra observations (e.g., Kneifel et al., 2016; Kalesse et al., 2016), they cannot provide slant-wise polarimetric measurements of ZDR, which can be useful to estimate the shape of the ice particles, due to their observation geometry. In this study we want to investigate the feasibility to combine two spatially separated radars to derive observations of DWR and ZDR for size, shape and mass retrievals of ice cloud particles and aggregates detected above the ML. In the scope of the priority program "Polarimetric Radar Observations meet Atmospheric Modelling (PROM; Trömel et al., 2021)", funded by the German Research Foundation (DFG), we explore the added value when operational weather radars are augmented with cloud radar measurements. By using ZDR measurements from a polarimetric weather radar, we estimate the shape of the ice hydrometeors. To estimate particle size, we use simultaneous range-height indicator (RHI) scans from a scanning cloud radar, 23 km apart from the weather radar, to obtain dual-wavelength observations for the same observation volume. As we aim to use DWR and ZDR measurements from two different locations, we are focused on case studies with homogeneous cloud scenes and in which hydrometeor attenuation can considered to be negligible. Therefore, we selected cloud cross-sections of cloud scenes with stratiform snowfall where water hydrometeors are unlikely to occur. To exclude liquid hydrometeors and melting layers, an ice mask was developed and applied to the observational dataset. Future studies will also include wet particles to improve the representation of melting and riming processes in numerical weather models. Combining ice scattering simulations and radar measurements we present an ice microphysics retrieval that resolves the ice water content, the median size and the apparent shape of the detected ice particles. The apparent shape, for simplicity the term shape will be used throughout this study, is described by the average observed aspect ratio which is strongly connected to the orientation of ice particles including their flutter around this preferential orientation. Our approach considers single RHI scans from each radar instrument resulting in a single radar cross-section. In the special case when the wind direction in this area is aligned to our radars cross-section, we can monitor the evolution of precipitation and the development of fall streaks inside the clouds by performing continuous RHI scans according to the precipitation rate. In another approach, to deeply investigate the initiation of convection as well as to better observe ice microphysical processes in clouds, we performed sector range-height indicator (S-RHI) using POLDIRAD and MIRA-35 to monitor precipitation cells during convection. In this way, a first scan was executed towards the cell of interest at a specific azimuth. Then, two additional fast RHI scans were executed from each radar deviated ±2° from the initial azimuth. This approach can result in nine vertical profiles within the precipitation cell providing additional microphysical information (Köcher et al., 2022, their Fig. 1).

Our approach of combining two radars located at different areas has multiple advantages. First of all, we can exploit the non-Rayleigh scattering, which usually complicates $Z_e$-only retrievals, by using the DWR to constrain the size of the atmospheric hydrometeors. DWR has been used so far in many conventional retrievals to retrieve the particles size - usually by making an a-priori assumption of the ice particles shape, e.g., the aspect ratio. In our approach, we augment this technique with polarimetric measurements (e.g., ZDR). Especially when the other scanning radar is pointing upwards, ZDR creates added value when obtained from a second, scanning radar. For the multi-wavelength technique, oblate ice particles appear like spheres when it is applied to vertically pointing radars. In this study we advocate that spatially separated radars are suited to provide this kind of measurement. Operational weather radars located throughout Germany could therefore be used in synergy with already established cloud radar sites to monitor precipitation but also to obtain microphysical properties of atmospheric hydrometeors.

This manuscript is organized as follows: In Sect. 2 the instruments used to produce the measurements dataset are described. In Sect. 3 the measurement strategy and the error assessments of the radar observations as well as the T-Matrix scattering simulations are presented in detail. Section 3 also demonstrates the methodology to combine DWR and polarimetric

measurements along with the scattering simulations in order to retrieve microphysical properties of ice particles. In addition, the attenuation correction methods are described. In Sect. 4, retrieval results along with their uncertainties as well as statistical results of the ice microphysics retrieval are presented. Furthermore, limitations of this study, comparisons to other methods and the performance of the ice retrieval in different areas of the radars cross-section are fully discussed. In Sect. 5, the conclusions for the presented approach are drawn.

**2 Instruments**

This feasibility study to combine two spatially separated weather and cloud radars was conducted in the scope of the IcePolCKa project (Investigation of the initiation of convection and the evolution of precipitation using simulations and polarimetric radar observations at C- and Ka-band), which is part of the PROM priority program (Trömel et al., 2021). For the DWR dataset the synergy of two polarimetric radars, the C-band POLDIRAD weather radar at German Aerospace Center

(DLR) in Oberpfaffenhofen and the Ka-band MIRA-35 cloud radar at Ludwig Maximilians University of Munich (LMU), Munich was used. POLDIRAD and MIRA-35 performed coordinated RHI scans towards each other (azimuth angle constant for both radars) at a distance of 23 km between DLR and LMU, monitoring stratiform precipitation events.

**2.1 POLDIRAD**

POLDIRAD (Fig. 1, left) is a polarization diversity Doppler weather radar operating at C-band at a frequency of 5.504

GHz ($\lambda = 54.5$ mm, $\lambda_1$ in Eq. 2). The radar is located at DLR, Oberpfaffenhofen, 23 km southwest of Munich at 48°05'12" N and 11°16'45" E at an altitude of 602.5 m above mean sea level (MSL). Since 1986, POLDIRAD has been operated at the roof of Institute of Atmospheric Physics (IPA), DLR for meteorological research purposes (Schroth et al., 1988). The weather radar consists of a parabolic antenna with a diameter of 4.5 m and a circular beam width of 1°. A magnetron transmitter with a power peak of 400 kW and a Selex ES Germatronik GDRX digital receiver with both linear and logarithmic response are synchronized

with the polarization network of the receiver, which can record the linear, elliptic and circular polarization of each radar pulse (Reimann and Hagen, 2016). POLDIRAD has the capability to receive the co- and cross-polar components of the horizontal, vertical, circular and elliptical polarized transmitted electromagnetic waves. In this way it provides several polarimetric variables, e.g., ZDR, ρHV etc., which can be used to obtain additional information about the size, shape, phase, and falling behavior of the hydrometeors in the atmosphere (Straka et al., 2000; Steinert and Chandra, 2009). Depending on its operational

mode, the maximum range that can be reached is 300 km (for a pulse repetition frequency of 400 Hz, a pulse duration of 2 μs and a range resolution of 300 m), making it a suitable instrument for nowcasting in the surrounding area of Munich. For the present study POLDIRAD's maximum range was 130 km with a pulse repetition frequency of 1150 Hz, a pulse duration of 1 μs and a range resolution of 150 m. The system can also be operated in the STAR mode (simultaneous transmission and reception). Here, we used the alternate-HV mode (alternate horizontally and vertically polarized transmitted electromagnetic

waves) which allows measuring the cross-polar components of the back-scatter matrix. The elevation velocity during the RHI scans was 1°/s. The technical characteristics of POLDIRAD are presented in Table 1.

**2.2 MIRA-35**

MIRA-35 (Fig. 1, right) is a Ka-band scanning Doppler cloud radar developed by Metek (Meteorologische Messtechnik GmbH, Elmshorn, Germany) with a frequency of ca. 35.2 GHz and a wavelength $\lambda = 8.5$ mm (Görsdorf et al., 2015), which is

$\lambda_2$ in Eq. (2). The cloud radar, which is operated by the Meteorological Institute Munich (MIM) as part of the Munich Aerosol

Cloud Scanner (MACS) project (also referred as miraMACS, Ewald et al., 2019), is located on the roof of the institute at the LMU at 48°08'52.2" N and 11°34'24.2" E and 541 m above MSL. The transmitter consists of a magnetron with a power peak of 30 kW which typically transmits radar pulses of 0.2 μs with a pulse repetition frequency of 5 kHz, corresponding to a range resolution of 30 m. The 1 m diameter antenna dish produces a beam width of 0.6°. The MIRA-35 cloud radar emits horizontally polarized radiation and measures both vertical and horizontal components of the backscattered wave. Hence, it has the capability to perform LDR measurements. The cloud radar usually points to the zenith, but can also perform RHI scans at different azimuths with elevation velocity 4°/s and plan-position indicator (PPI) scans at different elevations angles. The technical characteristics of MIRA-35 are presented in Table 1.


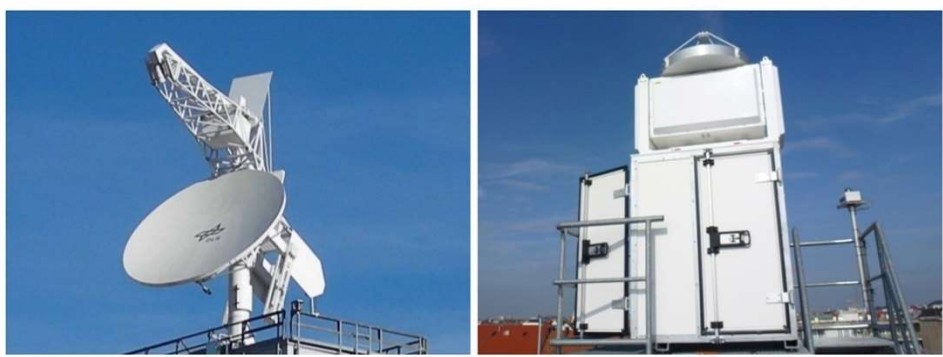


**Figure 1:** C-band POLDIRAD weather radar (left, photo: Dr.rer.nat. Martin Hagen) and Ka-band MIRA-35 cloud radar (right, photo: Prof. Dr. Bernhard Mayer).

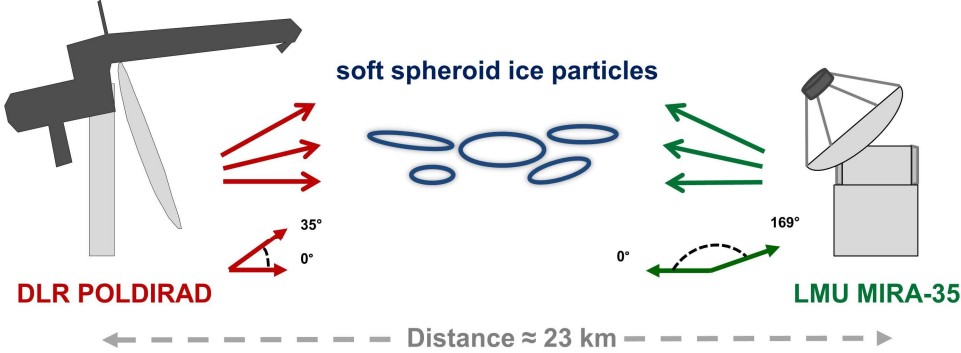

**Figure 2:** Geometry of the radar setup. The range of elevation angles is 0°–35° and 0°–169° for POLDIRAD and MIRA-35, respectively.

**Table 1:** POLDIRAD and MIRA-35 technical characteristics.

|  | POLDIRAD | MIRA-35 |
|---|---|---|
| **frequency/wavelength** | 5.5 GHz/ 54.5 mm | 35.2 GHz/ 8.5 mm |
| **peak transmitted power** | 400 kW | 30 kW |
| **antenna diameter** | 4.5 m diameter | 1 m diameter |
| **beam width** | 1.0º | 0.6º |
| **transmit mode** | pulse duration: 1 μs<br>pulse repetition frequency: 1150 Hz<br>max. range: 130 km<br>range resolution: 150 m | pulse duration: 0.2 μs<br>pulse repetition frequency: 5000 Hz<br>max. range: 24 km<br>range resolution: 30 m |

## 3 Methods

This study intends to investigate the synergy of two spatially separated radars to retrieve microphysical properties of ice hydrometeors detected in clouds, that are known to affect the type of precipitation (e.g., stratiform or convective), aiming to improve their representation in numerical weather models. To address this, an ice microphysics retrieval scheme has been developed. In this way, the microphysical properties of ice hydrometeors are revealed for stratiform snow precipitation cases. In this section, our approach is presented in detail and demonstrated using a case study example from 30[th] January 2019 when a snowfall event took place over the Munich area. At 04:00 UTC of that night, an ice cloud started forming at an altitude of 9 km. During the time of our coordinated measurements the cloud's vertical extension was up to 7 km. Throughout that day, the ambient temperature was mostly below 0°. The wind speed at the surface was very low, while at higher altitudes exceeded 15 m s$^{-1}$ at some cases. The vertical gradient of the wind favored the development of fall streaks (also shown in our radar observations in Fig. 3) and thus, ice particle growth within the ice cloud.

### 3.1 Measurements strategy and data preprocessing

Coordinated RHI measurements with POLDIRAD and MIRA-35 have been collected during three snowfall days on 9[th], 10[th] and 30[th] January 2019, with some ice particles reaching the ground where both radars are located (602.5 m for POLDIRAD and 541 m for MIRA-35, both heights above MSL). However, only ice particles above the melting layer were investigated in the present study. 59 RHI scans were executed from the two radars at almost the same time (time difference between RHIs was estimated less than 15 s) with a temporal resolution which was adjusted to the precipitation rate. POLDIRAD scanned between 0°–35° elevation towards MIRA-35 (northeast direction, azimuth of 73°), while MIRA-35 scanned between 0°–90° elevation towards POLDIRAD (southwest direction, azimuth of 253°) as well as 90°–169° elevation in a backward northeast direction but still inside the common cross-section (Fig. 2). With this setup, the cross-section between the two radars as well as beyond the MIRA-35 position was fully covered to record the development and microphysics of precipitation cells and fall streaks. During the snow events, $Z_e$ measurements from the two radars were performed and interpolated, using the nearest-neighbor interpolation method, onto a common rectangular grid (50 × 50 m). The 0-height of this grid is defined to be the height above MSL, while POLDIRAD and MIRA-35 locate at 602.5 m and 541 m height above MSL. In Fig. 3a and Fig. 3c, the measured $Z_e$ from the two radar systems during the RHI scans from 30[th] January 2019 at 10:08 UTC is presented. For the MIRA-35 $Z_e$ measurements we applied a calibration offset of 4 dBZ as derived in Ewald et al. (2019). Studying only snow cases no strong effects of hydrometeor attenuation are expected (e.g., Nishikawa et al., 2016). However, an iterative method to estimate hydrometeor attenuation has been developed. Additionally, both $Z_e$ datasets are corrected for gaseous attenuation using the ITU-R P.676-12 formulas provided by International Telecommunication Union (ITU) in August 2019 (ITU-R P.676-12, 2019). Both methods are fully described in Sect. 3.3. After the interpolation of both radar reflectivities in the common radar grid, we calculated the DWR (Fig. 3b) using Eq. (2). Since DWR is defined as the ratio of $Z_e$ at two wavelengths, it is independent of number concentration $N$. Therefore, it exploits the difference in the received radar signal due to Mie effects to give size information. To avoid unwanted biases by measurement artefacts, DWR values lower than –5 dB and higher than 20 dB were excluded. Furthermore, errors from other sources, e.g., beam width mismatch effects (beam width 1º for POLDIRAD and 0.6º for MIRA-35), are considered (fully explained in Sect. 3.1.2). Besides DWR measurements, polarimetric observations were used to study the shape of ice particles. POLDIRAD provided polarimetric measurements of ZDR, but only ZDR values between –1 dB and 7 dB were considered to be atmospheric hydrometeors signatures. The ZDR calibration was validated using additional measurements, described in detail in Sect. 3.1.2 and Appendix A. For the ZDR panel (Fig. 3d), reasonable boundaries for optimal visualization purposes were used in the colormap.

When $Z_e$, ZDR and DWR measurements are combined (Fig. 3), one can already get a first glimpse on the prevalent ice microphysics. Especially below 3 km height, between 20–30 km from POLDIRAD, the large values of $Z_e$ accompanied with the large values of DWR (greater than 5 dB) and the low values of ZDR (lower than 1 dB) indicate the presence of large and

quite spherical ice particles. In the following, quantitative ice microphysics will be revealed by the combination of $Z_e$, DWR and ZDR measurements with scattering simulations for a variety of ice particles.

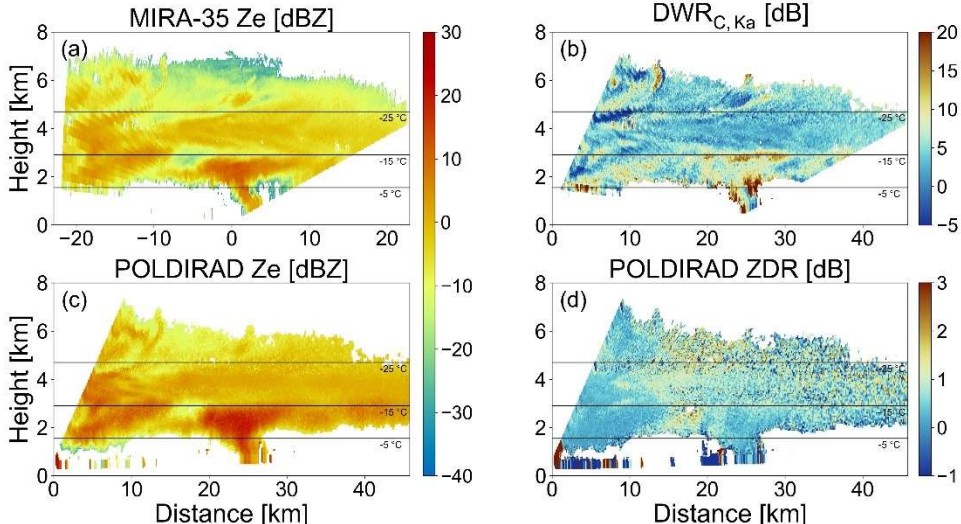

**Figure 3:** Radar observations of (a, c) MIRA-35 and POLDIRAD $Z_e$, (b) DWR and (d) POLDIRAD ZDR from 30th January 2019 at 10:08 UTC. The –5 °C, –15 °C and –25 °C temperature levels are plotted with black solid lines (source: Deutscher Wetterdienst, data provided by University of Wyoming; http://weather.uwyo.edu/upperair/sounding.html, last access: 08 April 2022).

### 3.1.1 Application of the ice mask

As already mentioned, the current version of the ice microphysics retrieval only accounts for ice particles that are detected into clouds above ML. Hence, radar datasets should be filtered accordingly and an ice mask should be applied. The implementation of the ice mask using threshold from polarimetric radar variables, i.e., MIRA-35 LDR, POLDIRAD ZDR and ρHV, as well as temperature sounding data (shown in Fig. 4), are fully presented in Appendix B.

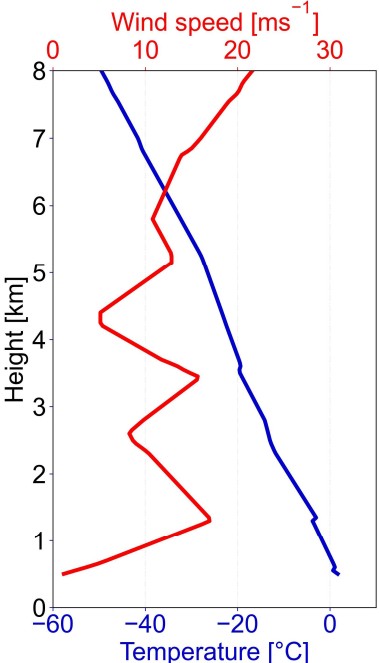

**Figure 4:** Temperature and wind speed data from Oberschleißheim sounding station (about 13 km north of Munich, source: Deutscher Wetterdienst, data provided by University of Wyoming; http://weather.uwyo.edu/upperair/sounding.html, last access: 08 April 2022) at 12:00 UTC are presented.

### 3.1.2 Assessment of radar observations errors

Radar measurements are often affected by systematic or random errors. To assess their impact on the ice microphysics retrieval developed in this study we need to investigate possible errors in POLDIRAD and MIRA-35 observations as well as all their sources.

The absolute radiometric calibration of both instruments is an important error source in DWR measurements. While the error of the absolute radiometric calibration of POLDIRAD is estimated to be ±0.5 dB following the validation with an external device (Reimann, 2013), the budget laboratory calibration of MIRA-35 following Ewald et al. (2019) is estimated to be ±1.0 dB.

In order to test for a systematic ZDR bias, we exploited POLDIRAD measurements during vertically pointing scans (also known as birdbath scans, e.g., Gorgucci et al., 1999) in a liquid cloud layer performed on the 4$^{th}$ April 2019. The measurements indicated that ZDR has an offset of about +0.15 dB as ZDR values are expected to be near 0 dB for this case due to the spherical apparent shape of liquid droplets. Although the examined calibration study from the 4$^{th}$ April 2019 was conducted three months later, we consider this ZDR offset to be reliable since calibration efforts showed similar values over the past years. Recent studies (Ryzhkov et al., 2005; Frech and Hubbert, 2020; Ferrone and Berne, 2021) confirm the stability of ZDR offsets for even longer time periods as long as the integrity of the antenna is maintained and wet radome effects are avoided. In Fig. A1 (Appendix A), examples of radar reflectivity $Z_e$, differential reflectivity ZDR as well as a scatter plot showing the average ZDR offset are presented. The scatters in the last panel (Fig. A1c) indicate the median ZDR value averaged over the full measurement period, shown in Fig. A1a, for each vertical radar bin within the cloud layer. The data were acquired by super sampling the 150 m pulse in 75 m range steps to enhance the signal statistics. To further ensure the stability of ZDR bias, an additional calibration validation was conducted following the Ryzhkov and Zrnic (2019) approach (described in their Sect. 6.2.4). Our measurement dataset from January 2019 was filtered for large $Z_e$ regions and intermediate temperatures for dry and large aggregates. This analysis yielded a median ZDR = 0.2 dB for these areas, where ice aggregates are expected, indicating that POLDIRAD was well calibrated during the period of this study.

Another error that should be considered is the random error, especially for ZDR measurements at low signal levels. To detect and filter out regions with high ZDR noise we compare the local (3 range gates) standard deviation ZDR$_{stdv}$ with the local mean ZDR$_{mean}$. Subsequently, we only include regions where the signal ZDR$_{mean}$ exceeds the noise ZDR$_{stdv}$ by one order of magnitude. An example of this approach can be found in Fig. A2 (Appendix A).

In our case of spatially separated radar instruments, an azimuthal misalignment between both instruments had to be excluded to obtain meaningful DWR measurements. To this end, we performed several solar scans with both instruments in spring 2019 to confirm their azimuthal pointing accuracy (e.g., Reimann and Hagen, 2016). Here, we found an azimuth offset of –0.2° for POLDIRAD and an azimuth offset of +0.1° for MIRA-35. Consecutive solar scans confirmed the azimuthal pointing accuracy within ±0.1°. Despite the small azimuthal misalignment, the radar beam centroids of both instruments were clearly within the respective other beam width during our measurement period in 2019.

Besides an azimuthal misalignment, we also analyzed the temporal mismatch between both RHIs as well as the volumetric mismatch in the context of non-uniform beam filling. Although the RHIs from the radars were scheduled to be executed simultaneously, regions within the RHIs are measured at slightly different times by both instruments. This temporal mismatch can lead to slightly different $Z_e$ radar observations from both radars in the context of horizontal advection of an inhomogeneous cloud scene. In the following we used this temporal mismatch to estimate the resulting DWR error for the example case shown in Fig. 3. Using wind data (Fig. 4) from the Oberschleißheim sounding station (source: Deutscher Wetterdienst, data provided by University of Wyoming; http://weather.uwyo.edu/upperair/sounding.html, last access: 08 April 2022), we converted the temporal mismatch (Fig. 5a) between the radar measurements for each pixel in the common radar grid to a spatial difference (Fig. 5c). To estimate the impact of this spatiotemporal mismatch (hereafter spatiotemporal error) we subsequently used these spatial differences to calculate DWR errors between pixels in the spatially higher resolved MIRA-35 $Z_e$ measurements (Fig.

5e). Concluding the DWR error assessment, we also analyzed the volumetric mismatch caused by the different beam widths of the two radars. For spatially heterogeneous scenes, this volumetric mismatch can lead to artificial DWR signatures caused by a non-uniform beam filling. Here, the spatially higher resolved MIRA-35 $Z_e$ measurements (30 m range gate length) along the RHI cross section were used as a proxy to obtain the spatial heterogeneity of $Z_e$ perpendicular to the RHI cross section. In a first step, the local beam diameters for each pixel in the common grid are calculated for POLDIRAD (Fig. 5b) and MIRA-35 (Fig. 5d). Then, moving averages along the $Z_e$ cross sections from MIRA-35 are performed using the corresponding local beam diameters. Hence, at each pixel of the common radar grid two averaged MIRA-35 $Z_e$ values are obtained; one corresponding to the local beam diameter of MIRA-35 and one corresponding to the local beam diameter of POLDIRAD. Subtracting the averaged $Z_e$ for each pixel, we were able to estimate the error caused by the volumetric mismatch between both radar beams (Fig. 5f). We apply the retrieval to all cloud regions, except for those filtered out by the ice mask and noise thresholds. The aforementioned errors are considered during the statistical aggregation of retrieval results (Sect. 4.2).

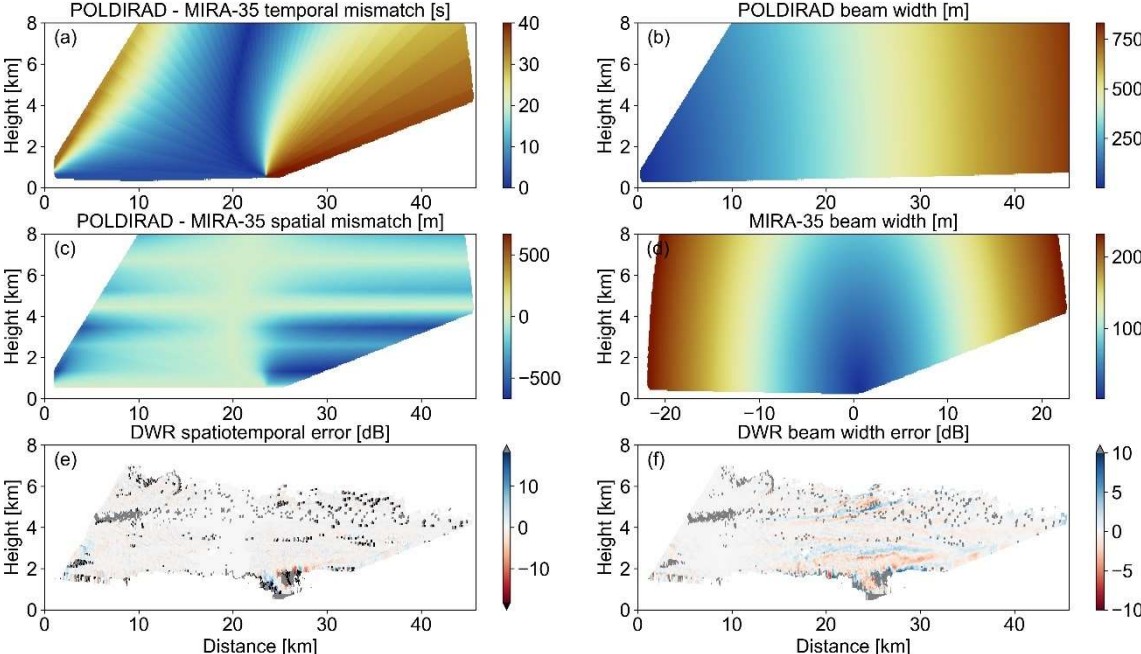

**Figure 5:** DWR error assessment due to temporal mismatch (left panels) and volumetric mismatch (right panels). In (a), (c) and (e) panels, the POLDIRAD and MIRA-35 temporal and spatial mismatch, as well as the spatiotemporal error in dB are plotted. In (b) and (d) panels, the POLDIRAD and MIRA-35 beam widths are presented, while in panel (f) the estimated DWR error due to the volumetric mismatch is shown. For this plot the data from 30th January 2019 at 10:08 UTC are used. The ice masked and noise-filtered values in (e) and (f) are plotted with grey color. Black color in panel (e) denotes the additional missing values due to the spatial shift of the radar grid. For better visualization purposes the –5 ºC, –15 ºC and –25 ºC temperature levels are not plotted here.

## 3.2 Numerical methods

Complementary to measurements, the numerical methods used in this work are introduced in the following section. First, an ice crystal model needs to be assumed which can be used in a scattering algorithm to simulate the backscattering of these crystals. On this basis, radar variables can be computed which can then be compared to radar measurements. As we intended to retrieve the apparent shape, the size and the mass of the detected ice hydrometeors, we used aspect ratio (hereafter referred as AR), median mass diameter ($D_m$) of the PSD, and ice water content as three degrees of freedom of the simulated ice particles for the development of look-up tables (LUTs). Different LUTs for several angles of radars geometry (Fig. 2) were created. Their values were then interpolated to fit all possible radar viewing geometries and used in the ice retrieval. The a-priori assumptions used in the simulations are fully described in the following sub-sections.

### 3.2.1 Soft spheroid model

For the scattering simulations we assumed that ice hydrometeors can be represented by ice spheroids. These so-called *soft*
*spheroids* are assumed to be homogeneous ice particles composed of an ice-air mixture with a real refractive index close to 1.

**Refractive index**

Our soft spheroid model uses the effective medium approximation (EMA) to model the refractive index of the composite
material as an ice matrix with inclusions of air following the Maxwell-Garnett (MG) mixing formula given in Garnett and
Larmor (1904):

$$\frac{e_{\text{eff}} - e_i}{e_{\text{eff}} + 2e_i} = f_i \frac{e_i - e_m}{e_i + 2e_m} \qquad (4)$$

with,

$e_m$, $e_i$: the permittivities of the medium and the inclusion, respectively,

$e_{\text{eff}}$: the effective permittivity,

$f_i$: the volume fraction of the inclusions.

The complex refractive index, $m_{\text{EMA}}$, is then calculated from $m_{\text{EMA}} = \sqrt{e_{\text{eff}}}$. In the framework of the EMA, the
electromagnetic interaction of an inhomogeneous dielectric particle (components with different refractive indices) can be
approximated with one effective refractive index of a homogeneous particle (e.g., Liu et al., 2014; Mishchenko et al., 2016).
In Liu et al. (2014), internal mixing was proven to best represent the scattering properties of hydrometeors. Here, the refractive
index is modelled as an internal mixing of ice with air inclusions which are arranged throughout the ice particle. The same
work also pointed out that the size parameter $D_{\text{crit}} = \frac{\pi d}{\lambda}$ for each of these air inclusions should not be larger than 0.4 (with $d$
as the diameter of the inclusion).

**Aspect ratio**

The shape of the particles is defined using the aspect ratio, AR. In this study, AR is defined as the ratio of the horizontal
to rotational axis of the particle. From the description of the simulated ice spheroids in Fig. 6, it is obvious that oblate (shaped
like lentil) and prolate particles (shaped like rice) have AR larger and lower than 1.0, respectively, as *z* axis is selected to be
the rotational axis. Using this principle, the representative value of sphericity $S = 0.6$ for oblate ice spheroids from Hogan et
al. (2012) is calculated as AR = 1.67 in this study and therefore, this number was used as a reference value for the simulation
plots (Fig. 7, Fig. 8 and Fig. 9a). In this work, we used $S$ additionally to AR to compare retrieval results when the oblate and
prolate shape assumption is used. $S$ for oblates and prolates is found to be smaller than 1, while for spheres is equal to 1. Here,
all ice particles were assumed to fall with their maximum diameter aligned to the horizontal plane. Hence, all ice prolates
(hereafter referred as horizontally aligned prolates or horizontally aligned prolate ice spheroids) are rotated 90° (mean canting
angle) in the yz plane (Fig. 6), while ice oblates are not rotated (0° mean canting angle). The variability of the canting angle,
i.e., the angle between the particle's major dimension and the horizontal plane, of the falling hydrometeors has been the topic
of several studies. This value in nature is not so easy to estimate and thus, a standard deviation (e.g., 2°–23° as in Melnikov,
2017) is often additionally used. Here, we used a fixed standard deviation of 20° to describe the oscillations of the particles
maximum dimension around the selected canting angle. Then the calculation of the scattering properties is performed using
an adaptive integration technique for all possible particle's geometries, ignoring the Euler angles alpha and beta of the
scattering orientation.

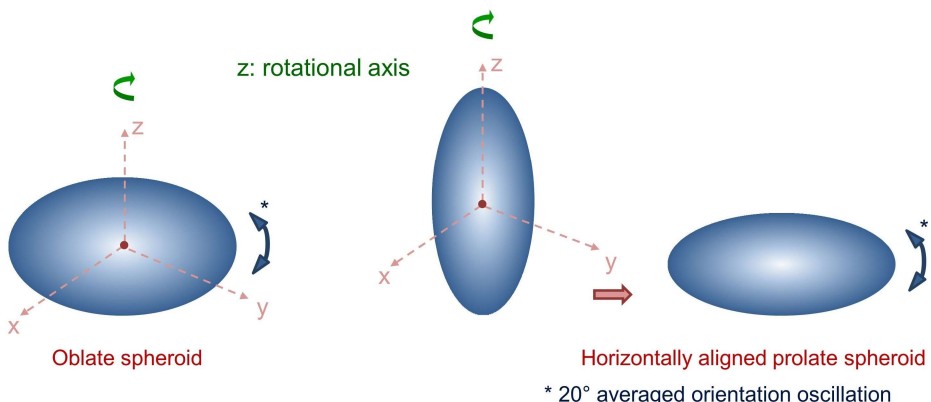

z: rotational axis

Oblate spheroid

Horizontally aligned prolate spheroid

* 20° averaged orientation oscillation

**Figure 6:** Description of simulated oblate, vertically aligned prolate and horizontally aligned (rotated 90° in the yz plane) prolate ice spheroids. Only oblate and horizontally aligned prolate ice spheroids were used in the scattering simulations with a 20° standard deviation out of the horizontal plane.

**Mass-size relation**

The maximum dimension, $D_{max}$, and the sphericity values for the spheroids were a-priori defined and their mass was calculated according to the formula that describes the relation between mass and $D_{max}$, i.e., mass-size relation. This formula can provide information about the mass of the ice particles and therefore, their effective density with respect to their size. Mass $m$ of an ice particle is usually connected to its maximum diameter $D_{max}$ with a power-law formula,

$$m(D_{max}) = aD_{max}^b \qquad (5)$$

where,

$a$: the prefactor of the $m(D_{max})$, refers to the density scaling at all particles sizes,

$b$: the exponent of the $m(D_{max})$, relates to the particles shape and growth mechanisms. With the mass and the spheroid dimensions known, the density of the ice spheroid was calculated. In the special case when the density was found to exceed that of solid ice (0.917 g cm$^{-3}$), the mass of the spheroid was clipped and its density was set equal to 0.917 g cm$^{-3}$.

For the mass of the ice particles, the modified mass-size relation of Brown and Francis (Brown and Francis, 1995) as presented in Hogan et al. (2012) , hereinafter referred to as BF95, is initially used in this study,

$$m(D_{max}) = 480D_{max}^3, \quad D_{max} < 6.6 \times 10^{-5} \text{ m}$$
$$m(D_{max}) = 0.0121D_{max}^{1.9}, \quad D_{max} \geq 6.6 \times 10^{-5} \text{ m} \qquad (6)$$

where,

$D_{max}$: maximum dimension of a spheroid in meters (m),

$m$: mass of the particle in kilograms (kg). While the effective density of a spheroid decreases strongly with its size due to the exponent b = 1.9 in BF95, we contrast this with a second $m(D_{max})$ with a higher and constant density. To that end, we borrowed the $m(D_{max})$ from the irregular aggregate model from Yang et al. (2000) to create soft spheroids with an analog mass-size ratio. Originally, the construction of these aggregates was fully described in Yang and Liou (1998) as an aggregated collection of geometrical hexagonal columns. In our study, this second soft spheroid model only emulates the maximum dimension and mass of the underlying aggregates. Assuming spheroids to represent the ice aggregates, the density and thus, the mass of the particles can be calculated via the melted-equivalent diameter $D_{eq}$ using $D_{max}$ in Eq. (7). The $D_{eq}$ is used to describe the diameter of a spherical water particle with the same mass as an ice particle with maximum dimension $D_{max}$.

$$m(D_{max}) = \frac{\pi \rho_w D_{eq}^3}{6} = \frac{\pi \rho_w}{6} e^{\sum_{n=0}^{4} b_n (\ln(D_{max}))^n 3} \tag{7}$$

where $b_n$ is taken from Table 2 in Yang et al. (2000), the water density $\rho_w = 1$ g cm$^{-3}$ and $D_{eq}$ as well as $D_{max}$ are in microns.

**Particle size distribution**

In all calculations of our study, ice particle sizes were assumed to follow the normalized Gamma particle size distribution of Bringi and Chandrasekar (2001) with a shape parameter $\mu = 0$ (exponential PSD), a typical value for snow aggregates (e.g., Tiira et al., 2016; Matrosov and Heymsfield, 2017 and many more):

$$N(D) = N_w f(\mu) \left(\frac{D}{D_0}\right)^\mu e^{\frac{-(3.67+\mu)D}{D_0}} \text{ with } f(\mu) = \frac{6}{3.67^4}(3.67 + \mu)^{\frac{(\mu+4)}{\Gamma(\mu+4)}}, \tag{8}$$

where,

$N_w$: the intercept parameter,

$\mu$: the shape parameter,

$D_0$: the Median Volume Diameter,

$D$: the melted-equivalent diameter of the ice particles (defined as $D_{eq}$ in this study). The Median Volume Diameter ($D_0$) is one of the three parameters used to define the Gamma PSD for the scattering simulations and is the size which separates the PSD in half with respect to volume (defined as: $\int_0^{D_0} D^3 N(D) dD = \frac{1}{2} \int_0^{D_{max\_PSD}} D^3 N(D) dD$). However, the use of Median Mass Diameter is more common in ice studies. Median Mass Diameter, or Equivalent Median Diameter of the ice particles which have been melted, or simple, $D_m$, is the size that splits the PSD in half with respect to mass (defined as: $\int_0^{D_m} m(D)N(D) dD = \frac{1}{2}$IWC, Ding et al., 2020). Although DWR can be used to retrieve median size $D_0$ of PSD without $D_0$ being much affected by the density of the ice particles (e.g., Matrosov, 1998; Hogan et al., 2000), it can be also used to retrieve $D_m$ when a mass-size relation is investigated as $D_m$ is significantly affected by the used m(D$_{max}$). For instance, Leroy et al. (2016) found that $D_m$ is significantly affected by the $b$ exponent of the m(D$_{max}$) and thus, by the mass and the density of ice hydrometeors. As we aim to investigate how the choice of different parameters affects the results of the ice retrieval (mass, shape and median size), we were also focused on the $D_m$ median size. Along with the shape parameter $\mu$ and intercept parameter $N_w$, soft spheroids with a defined AR were used to calculate $Z_e$ and specific attenuation $A$ at both radar wavelengths, and ZDR only at 54.5 mm as this radar variable is only provided by POLDIRAD. For the refractive index calculation, the Maxwell-Garnett mixing formula was used (e.g., Garnett and Larmor, 1904). In addition to $Z_e$, $A$ and ZDR simulations, the IWC of the PSD was calculated. The $N_w$ that corresponds to this value of IWC served as a factor for rescaling to the desired IWC values used for the simulations. The rescale factor was used for the new estimation of $Z_e$, $A$ and ZDR. In Fig. 7 an example of the Gamma PSD for intercept parameter $N_w = 1 \times 10^3$, shape parameter $\mu = 0$ (exponential PSD), different $D_m$ values and constant AR = 1.67 is presented, showing how $D_m$ and the shape of the PSD are related. For all calculations, a minimum and a maximum diameter of $2 \times 10^{-2}$ mm and 20 mm were used as integration boundaries in the PSD of the ice particles, as we aim to retrieve microphysics only for ice particles detected into clouds and above ML.

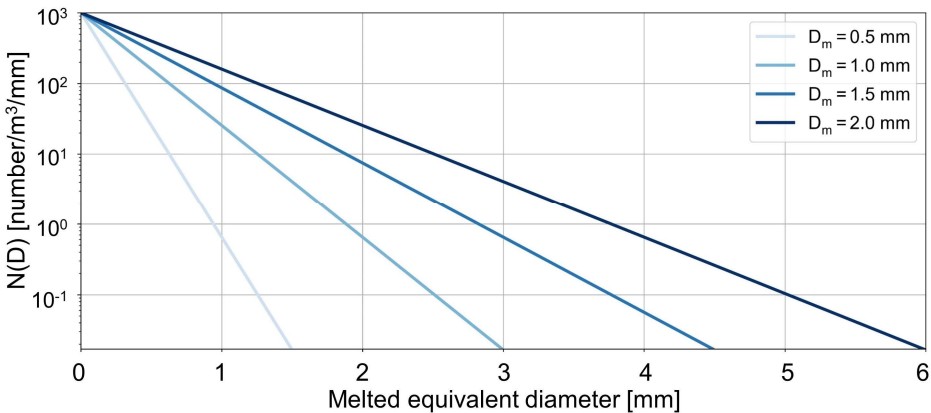

**Figure 7:** Gamma PSD for different values of $D_m$, AR = 1.67, $N_w = 1\times10^3$, and $\mu = 0$.

### 3.2.2 Scattering simulations

The single scattering properties of the ice spheroids were calculated using the T-matrix scattering method as described by e.g., Waterman (1965), Mishchenko and Travis (1994) or Mishchenko et al. (1996). The averaging over particle orientations and the calculation of radar variables for whole size distributions are done using PyTMatrix (Leinonen, 2014) since the simple Rayleigh approximation of Eq. (1a) cannot be used for soft spheroids. PyTMatrix is a package that can be easily adjusted to the user's needs via functions and classes regarding the desired preferences for particle shape, size, orientation, particle size distribution (PSD) and wavelength.

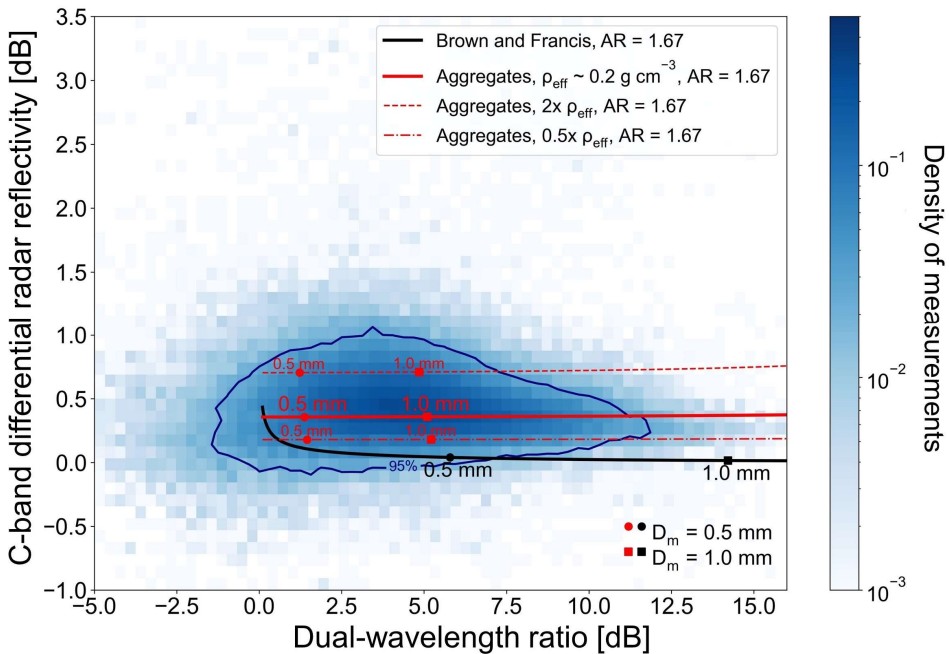

**Figure 8:** Radar observations between 0°–5° elevation angles and scattering simulations for ice spheroids with $\mathbf{m(D_{max})}$ corresponding to aggregates (red) and BF95 (black) for AR = 1.67, IWC = 0.50 g m$^{-3}$ and both radar beams simulated to be emitted horizontally. With scatters, the $\boldsymbol{D_m}$ = 0.5 mm and $\boldsymbol{D_m}$ = 1.0 mm are denoted. The 95$^{th}$ percentile of the 2d density histogram is drawn with a dark blue isoline. With red dashed and dash-dotted lines simulations for ice spheroids with double and half the density of aggregates are plotted.

Combining the PSD with the m(D$_{max}$) relationships of BF95 and aggregates, scattering simulations show that ice spheroids with m(D$_{max}$) analog to aggregates produce more pronounced polarimetric signatures for larger ice particles due to their higher density and in turn, higher real refractive index. This is illustrated by scattering simulations using both m(D$_{max}$) assumptions which are shown along with our radar observations in Fig. 8 (BF95 and aggregates line is plotted with black and solid red color, respectively). These calculations were done for horizontally emitted radar beams and for an aspect ratio of 1.67 and an IWC of 0.50 g m$^{-3}$. Here, larger DWR values are an indication of larger particles, while ZDR values around 0 are an indication of spherical particles. The same figure also shows scattering simulations for ice spheroids with double and half the

density of aggregates $m(D_{max})$ (red dashed and dash-dotted line, respectively). This influence of density on retrieval results will be further discussed in a sensitivity study presented in Sect. 4.3.1. In addition, Fig. 8 also shows our DWR-ZDR

measurements for low elevation angles (0°–5°) and for all 59 RHI coordinated scans as a blue shaded density histogram. The dark blue isoline frames the 95$^{th}$ percentile of our radar observations. In Fig. 8 it becomes apparent that the BF95 $m(D_{max})$ relationship assumed for our ice spheroids cannot explain our radar observations for large ice hydrometeors as ZDR values drop fast with increasing DWR due to the fast decrease of density with size. Therefore, BF95 will be excluded from further analysis. To compare BF95 with aggregates, some retrieval results using BF95 can be found in Sect. 4.3.1. The mass-size

relationship for aggregate ice particles can obviously better explain the density histogram of our DWR-ZDR dataset, especially for particles with DWR > 4 dB.

**Look-up tables structure**

Using ice spheroids that follow the $m(D_{max})$ of aggregates, we proceeded to the development of LUTs for different values

of $D_m$, AR, IWC and geometries covering the radar elevation angles presented in Fig. 2. $D_m$ of the PSD was varied between 0.1–3.02 mm in a logarithmic grid of 150 points. A minimum sensitivity limit of DWR = 0.1 dB was used in the simulations leading to different minimum retrievable $D_m$ according to the $m(D_{max})$ and the AR used, but also the radar viewing geometry (more details about this topic can be found in Appendix C). IWC was varied between 0.00001−1 g m$^{-3}$ in a logarithmic grid of 101 points. Scattering properties for spheroid oblate and horizontally aligned prolate ice particles were calculated and saved

in separated LUTs with the aspect ratio ranging between 0.125−1.0 (values: 0.125, 0.16, 0.21, 0.27, 0.35, 0.45, 0.6, 0.8, 1.0) for the horizontally aligned prolates and the inverse values for the oblate particles. Two examples of the scattering simulations are presented in Fig. 9. For the creation of both panels we assumed the simulated radar beams to be transmitted horizontally towards each other (horizontal-horizontal geometry). For Fig. 9a the AR was chosen 1.67. Radar reflectivity $Z_e$ at C-band as well as DWR were calculated for different $D_m$ values and different values of IWC of the PSD. Larger values of radar reflectivity

$Z$ at C-band are observed for larger values of $D_m$ and larger IWC. Furthermore, as $D_m$ increases, DWR increases as well, indicating the sensitivity of DWR to the size. An important remark is that for constant $D_m$, DWR remains invariant to varied IWC. For Fig. 9b we chose IWC to be 0.50 g m$^{-3}$. ZDR values are found to be invariant for all simulated values of IWC when AR, $D_m$ and shape parameter $\mu$ of PSD as well as the $m(D_{max})$ remained the same. All the aforementioned principles are then used to implement a method for retrieving ice microphysics information from radar measurements.

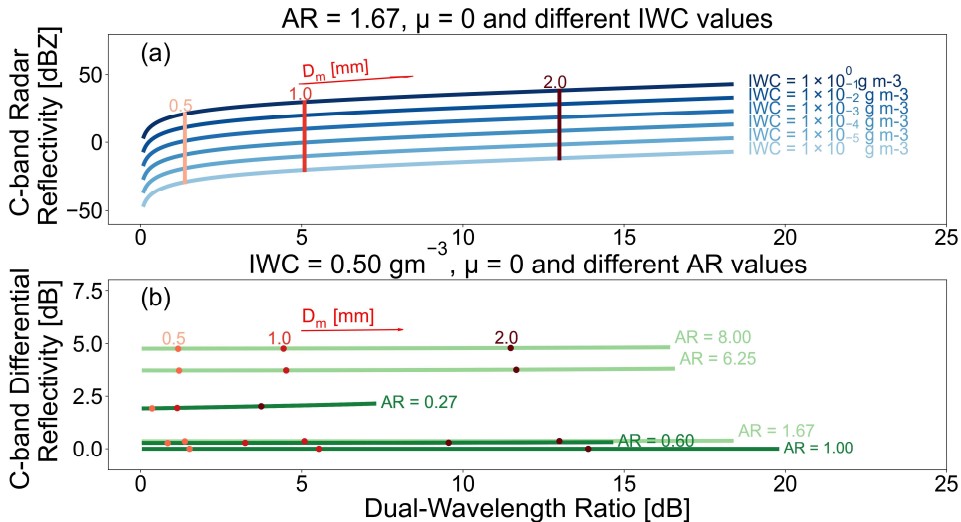

**Figure 9:** Scattering simulations for (a) radar reflectivity and (b) differential radar reflectivity vs. dual-wavelength ratio for horizontally aligned spheroid ice particles, horizontal-horizontal geometry, shape parameter $\mu = 0$ and **m(D$_{max}$)** of aggregates. For the upper panel the AR was chosen 1.67, while for the bottom panel the IWC was chosen 0.50 g m$^{-3}$. In panel (b) the light green and the dark green color lines

denote simulations for oblates and horizontally aligned prolates, respectively.

### 3.3 Correction of attenuation

Before using the radar observations for the development of the ice retrieval algorithm, they need to be corrected for beam propagation effects. One major influence is the attenuation by atmospheric gases and by hydrometeors. This holds especially true for the Ka-band radar measurements. Although snow attenuation in C-band can be mostly neglected especially for low density particles and low snowfall rates (Battan, 1973; Table 6.4), the corrections will be done in both radar bands for reliability purposes.

#### 3.3.1 Gaseous attenuation

Both MIRA-35 and POLDIRAD radar reflectivities are corrected for attenuation caused by atmospheric gases. Atmospheric water vapor can cause considerable attenuation of radar signals especially at the higher frequency (35.2 GHz) of our instrumentation. The gaseous attenuation for both radar bands is calculated using line-by-line formulas proposed by ITU-R P.676-12 model (ITU-R P.676-12, 2019). The corrections are implemented for oxygen and water vapor lines where the attenuation is expected to be significant. The gaseous attenuation formulas use atmospheric pressure, temperature and relative humidity for each RHI, obtained from the Copernicus Climate Change Service (C3S) Climate Data Store (CDS) ECMWF ERA5 reanalysis data (Hersbach et al., 2018).

#### 3.3.2 Hydrometeors attenuation

Next to the gaseous attenuation, the hydrometeor attenuation needs to be considered, too. For this purpose, an iterative approach using the ice microphysics results is developed. In this way, both radar reflectivities are corrected to mitigate the impact of hydrometeor attenuation on the ice microphysics retrieval. For this approach, the retrieval algorithm is used twice. A more detailed description of this method will be presented in Sect. 3.4 along with the developed ice retrieval scheme.

### 3.4 Development of ice microphysical retrieval

For the development of the ice retrieval scheme, radar measurements of $Z_e$, ZDR and DWR are compared with the PyTMatrix scattering simulations described in Sect. 3.2. The retrieved parameters are IWC in g m$^{-3}$, $D_m$ of the PSD in mm, and AR of the measured hydrometeors. Considering their different ranges, we used normalized differences between simulated and measured values of DWR as well as $Z_e$ and ZDR at C-band. By minimizing these differences, the best-fitting microphysical parameters are found. The microphysics retrieval is implemented in two steps using the minimization of the two following cost functions $J_1$ and $J_2$:

$$\min J_1(D_m, AR) = \text{norm}\big(\Delta ZDR(D_m, AR)\big) + \text{norm}\big(\Delta DWR(D_m, AR)\big)$$

$$\min J_2(IWC) = \text{norm}\left(\Delta Z_{e,C}(IWC)\right) \tag{9}$$

where with $\Delta$ the difference between simulated and measured parameter is denoted.

Both ZDR and DWR are invariant to IWC when same values of $D_m$ and AR are used. Therefore, $D_m$ and AR are found in the first step, whilst the IWC is constrained in the second step. While the DWR contributes to the retrieval of $D_m$, the ZDR measurement merely narrows down the solution of aspect ratio of the ice particles. As $Z_e$ at C-band is less affected by attenuation compared to Ka-band, it is better suited to estimate the IWC. After the retrieval of size $D_m$ and shape AR in the first step, the algorithm continues with these values with the retrieval of IWC in the second step by minimizing the cost function $J_2$ in the LUT. Completing these two steps, the microphysics retrieval has retrieved not only preliminary $D_m$, AR and IWC but also the specific attenuation $A$ at both radar bands which is used for the total attenuation estimation. As the ice retrieval produces results using radar measurements interpolated onto a cartesian grid, the retrieved $A$ at C- and Ka-band needs to be converted from cartesian to the original polar coordinates for the calculation of the total attenuation for each radar band. After

*A*, in polar coordinates, is integrated along the radar beams, the total attenuation for each radar dataset is calculated and converted back from polar to cartesian coordinates. Then, it is used to correct $Z_e$ for both radars. In the next step, the final microphysical parameters such as AR, IWC and $D_m$ are retrieved using the corrected $Z_e$ from both bands as well as ZDR from POLDIRAD. Figure 10 shows the process of attenuation correction and retrieval in more detail. An output example of the ice microphysics retrieval scheme for the already introduced case study from 30th January 2019 at 10:08 UTC (Fig. 11–12) can be found in Sect. 4.1. The total attenuation for this case study is presented as a supplement material accompanying this paper.

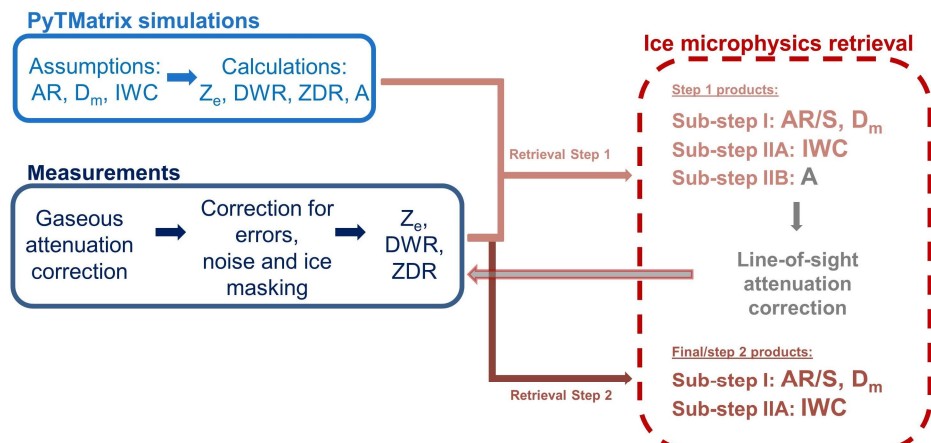

**Figure 10:** Ice microphysics flowchart. The dark blue color refers to radar observations. The light blue color is used for scattering simulations and the red dotted rounded rectangle gives information about the ice microphysics retrieval scheme. With gray color the total attenuation correction method is described.

## 4 Results

### 4.1 Retrieval of ice microphysics

59 pairs of coordinated RHI measurements from POLDIRAD and MIRA-35 were investigated. Here, we use a case study from 30th January 2019 at 10:08 UTC, already presented before in Fig.3, to demonstrate the output of the ice microphysics retrieval scheme. For all the presenting results, we anticipated that the ice hydrometeors can be represented by ice spheroids that follow the aggregates mass-size relation and the a-priori defined exponential PSD. The microphysical properties of the detected hydrometeors are shown in Fig. 11 (assuming oblate ice spheroids and LUTs for different radar viewing geometries) and Fig. 12 (assuming horizontally aligned prolate ice spheroids and LUTs for different radar viewing geometries). In Fig. 11a and Fig. 12a, the retrieved AR is presented. Both plots suggest that in the cross-section of the cloud between the two radars and especially, in the area which is below 3 km height at a distance 0–12 km away from POLDIRAD, more spherical ice hydrometeors are present. Further away at a distance 12–20 km from POLDIRAD, more aspherical particles with AR around 4.0 and AR around 0.5, for oblates and horizontally aligned prolates respectively, were found. The same result is also supported from *S* plots in Fig. 11c and Fig. 12c where $S > 0.6$ for the spherical particles between 0–12 km distance and $S < 0.6$ for the aspherical particles between 12–20 km distance. The retrieved AR and *S* could explain well the ZDR measurements in Fig. 3d where more spherical particles have ZDR < 0.5 dB, while aspherical particles have ZDR > 0.5 dB. Overall, the ZDR measurements could be replicated better with the retrieval results using oblate ice spheroids with RMSE = 0.19 dB (with the term RMSE, the mean Root Mean Square Error over all grid points is meant) between the fitted and measured ZDR for the whole scene, against RMSE = 0.25 dB when horizontally aligned prolate ice spheroids were used. With the a-priori assumptions for PSD and mass-size relation, the retrieved $D_m$ increasing towards the ground is an indication that large ice particles are present below 3 km height compared to smaller particles that are dominant at higher altitudes. This is obvious in

both oblates and horizontally aligned prolates results (Fig. 11b and Fig. 12b). Comparing this plot with the DWR measurements from Fig. 3b, we observe that the retrieved $D_m$ could reasonably explain DWR. The correlation between DWR and $D_m$ is found again to be better when oblate ice spheroids are used. The RMSE for the fitted-simulated and measured DWR is 0.50 dB when ice oblates are used in the simulations, while RMSE = 0.61 dB when the ice particles were assumed horizontally aligned prolates. Although DWR and ZDR measurements are combined for the shape and size retrieval (minimization of $J_1$ in Eq. 9),

the spatial patterns agreement between DWR-$D_m$ and ZDR-AR/$S$ plots indicate the strong correlation of DWR and ZDR with size and shape, respectively. Figure 11d and Fig. 12d show the results of the retrieved IWC for oblates and horizontally aligned prolate ice spheroids described by m($D_{max}$) of aggregates and an exponential PSD. Areas with positive POLDIRAD $Z_e$ values in Fig. 3c correspond to IWC values higher than $1 \times 10^{-3}\,\mathrm{g\,m^{-3}}$. Hence, the sensitivity of $Z_e$ to mass of the ice particles is again indicated for both spheroid shapes (oblates and horizontally aligned prolates). Nevertheless, the $Z_e$ RMSE for horizontally

aligned prolate ice particles is 0.36 dB, whilst the RMSE is found 0.20 dB when ice oblates are used. All RMSE which serve as residual values for the ice retrieval are collected in Table 2. The lowest RMSE are found when oblate ice spheroids are assumed.

        Figure 13 shows averaged profiles of $D_m$ and IWC for the whole cloud cross-section measured on 30[th] January 2019 at 10:08 UTC when different error sources and different shape assumptions are considered. In Fig. 13a and Fig. 13b, the averaged

$D_m$ and IWC profile for oblate ice spheroids, as they are calculated from Fig. 11b and Fig. 11d with only accounting for ice masked and noise-filtered measurements, are plotted in dark red and dark blue, respectively. In the same panels, the averaged $D_m$ and IWC profiles are plotted with different red and blue shades for different combinations of calibration errors for POLDIRAD (±0.5 dBZ) and MIRA-35 (±1.0 dBZ). Figure 13a indicates that the lowest values of $D_m$ are retrieved when the calibration for POLDIRAD and MIRA-35 would be $dZ_{e,\,C} = -0.5$ dBZ and $dZ_{e,\,Ka} = +1.0$ dBZ respectively, resulting in a DWR

bias of -1.5 dB. Due to the smaller $D_m$ retrieval, the retrieved IWC profile in Fig. 13b is the largest in this case. In the lower panels of the same figure, the same profiles of $D_m$ (Fig. 13c) and IWC (Fig. 13d) are plotted again, now including the additional errors caused by the spatiotemporal and beam width mismatch discussed in Sec. 3.1.2. While the beam width mismatch can locally lead to the most significant deviations (shown in Fig. 5), the calibration uncertainty (red and blue shades) in the worst case (for $dZ_{e,\,C} = -0.5$ dBZ, $dZ_{e,\,Ka} = +1.0$ dBZ vs. $dZ_{e,\,C} = +0.5$ dBZ, $dZ_{e,\,Ka} = -1.0$ dBZ) can lead to the largest bias throughout

the profile. With increasing microphysical heterogeneity within a cloud, the DWR error due to the volumetric mismatch between the instruments increases. Here, criteria would need to be defined where the estimated DWR errors indicate a non-applicability of the multi-wavelength technique. The selection of such criteria, however, would require an in-depth sensitivity study using model clouds and in-situ data which is beyond the scope of our study. Our error estimation therefore only serves as an indication in which areas the retrieval results should be taken with caution. The lower panels of Fig. 13 also show the

averaged $D_m$ and IWC profile (dashed lines) as they are calculated from Fig. 12b and Fig. 12d when horizontally aligned prolate ice spheroids are assumed. Between the two shape assumptions the horizontally aligned prolates yields a larger $D_m$ profile (+0.31 mm) on average, while oblate ice spheroids yield a slightly larger IWC profile (+0.002 g m$^{-3}$). With the influence of the calibration uncertainty on the retrieved $D_m$ and IWC profile with ±0.41 mm and ±0.02 g m$^{-3}$, respectively, the shape assumption is of equal significance for the retrieval of $D_m$ while it is less important for the retrieval of IWC.


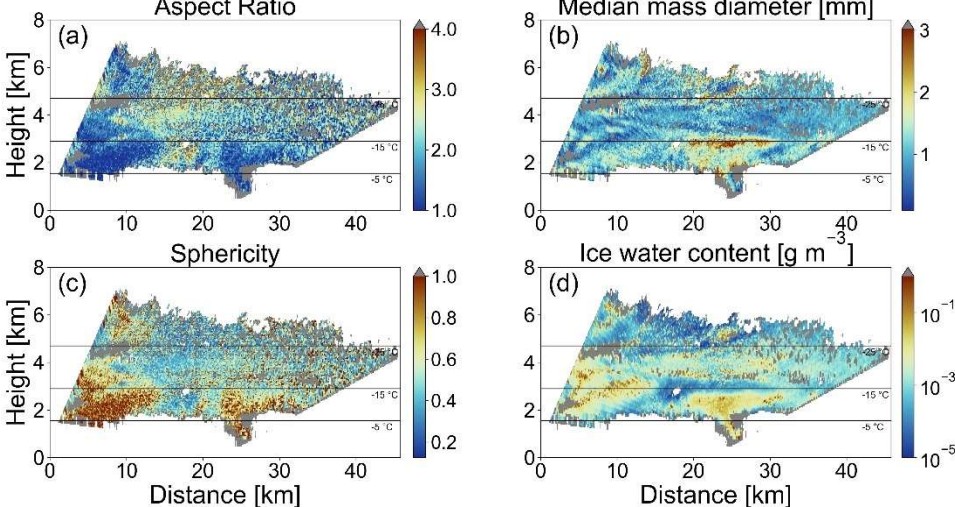

**Figure 11:** Retrieved (a) AR, (b) $D_m$, (c) $S$ and (d) IWC for 30th January 2019 at 10:08 UTC with ice spheroids assumed to be oblates and their $\mathbf{m(D_{max})}$ corresponding to aggregates from Yang et al. (2000). The –5 ºC, –15 ºC and –25 ºC temperature levels are plotted with black solid lines (source: Deutscher Wetterdienst, data provided by University of Wyoming; http://weather.uwyo.edu/upperair/sounding.html, last access: 08 April 2022). Areas where ice masked and noise-filtered measurement values locate are plotted with grey color.

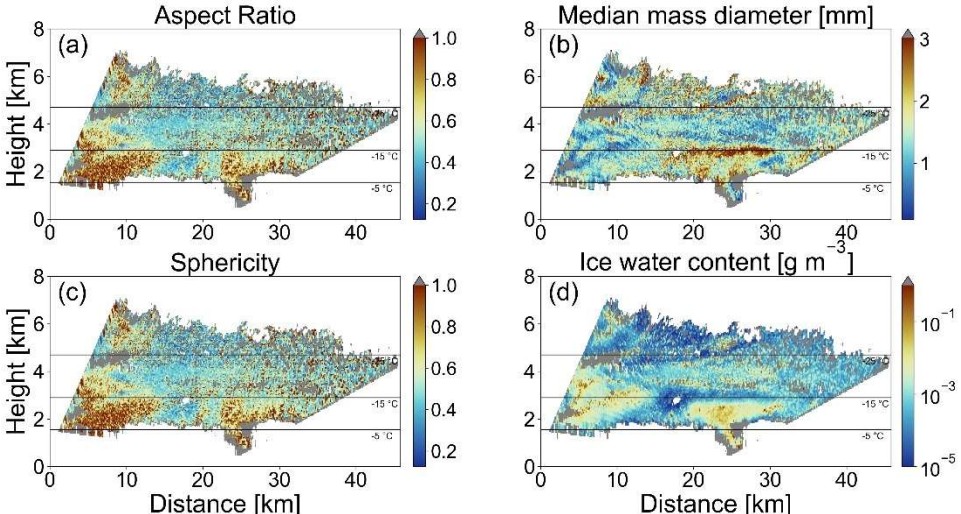

**Figure 12:** Retrieved (a) AR, (b) $D_m$, (c) $S$ and (d) IWC for 30th January 2019 at 10:08 UTC with ice spheroids assumed to be horizontally aligned prolates and their $\mathbf{m(D_{max})}$ corresponding to aggregates from Yang et al. (2000). The –5 ºC, –15 ºC and –25 ºC temperature levels are plotted with black solid lines (source: Deutscher Wetterdienst, data provided by University of Wyoming; http://weather.uwyo.edu/upperair/sounding.html, last access: 08 April 2022). Areas where ice masked and noise-filtered measurement values locate are plotted with grey color.

**Table 2:** RMSE values between simulated and observed ZDR, DWR, $Z_e$ values for the whole radar cross-section after running the retrieval for 30th January 2019 at 10:08 UTC using oblate and horizontally aligned prolate ice spheroids and assuming their $\mathbf{m(D_{max})}$ to be the aggregates from Yang et al. (2000).

| Shape assumption | Parameter | RMSE |
|:---:|:---:|:---:|
| | DWR | 0.50 dB |
| **Oblates** | ZDR | 0.19 dB |
| | $Z_e$ | 0.20 dB |
| | DWR | 0.61 dB |
| **Horizontally aligned prolates** | ZDR | 0.25 dB |
| | $Z_e$ | 0.36 dB |

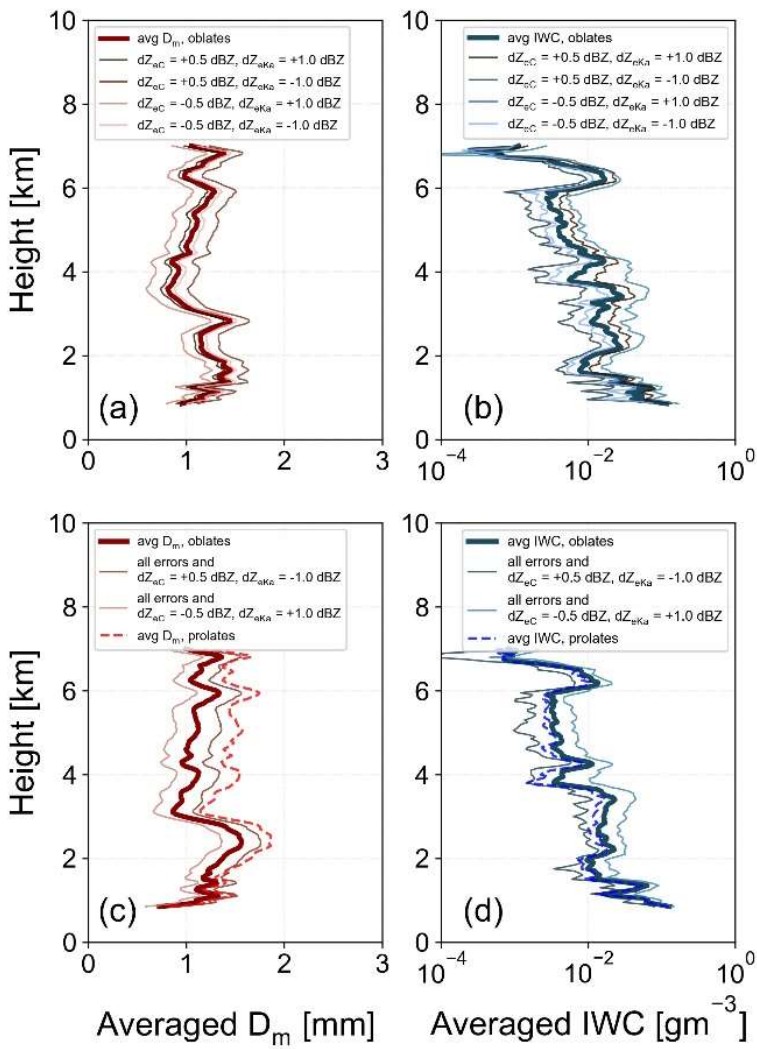

**Figure 13:** Averaged profiles of the retrieved (a) $D_m$ and (b) IWC as derived from Fig. 11b and Fig. 11d for oblate ice spheroids, with (thinner lines) and without (thicker line) considering the calibration error for both radars. (c, d) Same as upper panels but now the beam width error, the spatiotemporal error as well as dDWR = +0.5 dB or dDWR = –0.5 dB is considered. With dashed lines the retrieved $D_m$ and IWC as derived from Fig. 12b and Fig. 12d for the horizontally aligned prolate assumption is plotted. All panels refer to the case study from 30th January 2019 at 10:08 UTC as well as to aggregate mass-size relationship and exponential particle size distribution.

## 4.2 Statistical overview

After investigating 59 pairs of RHI scans from three different snow events (9th January 2019 between 11:18−15:08 UTC, 10th January 2019 between 09:08−17:08 UTC and 30th January 2019 between 10:08−12:38 UTC), we created stacked histograms with respect to temperature for a deeper insight of the retrieval. Particularly, all RHI measurements from these days were compared to scattering simulations in LUTs for oblate and horizontally aligned prolate ice particles. Statistical results of the retrieved $S$, $D_m$ as well as IWC are presented in Fig. 14. For these results, all errors/biases described in Sect. 3.1.2 are considered for the RHI measurements. For the statistics it is also assumed that ice hydrometeors are represented by ice spheroids following an exponential PSD and with $m(D_{max})$ corresponding to this of aggregates. In the first three panels of this figure, results for the retrieved parameters assuming oblate ice spheroids are presented, while in the last three panels, the same kind of results for horizontally aligned prolate ice spheroids are shown. At first glance, the majority of ice hydrometeors are found to be neither very spherical nor very elongated (green color panel plots, first column in Fig. 14). When oblate ice spheroids are used in the scattering simulations, the greater part of retrieved $S$ values is found to range from 0.3 to 0.6. With the assumption that ice hydrometeors can be represented by horizontally aligned prolate ice spheroids, the distribution is narrower with the majority of the detected particles to have $S$ values ranging between 0.4−0.6. From the $D_m$ retrieval (red color

panel plots, second column in Fig. 14) the results for oblates showed a narrower distribution shifted towards lower median mass diameters, while for horizontally aligned prolates the retrieved values are more broadly distributed towards larger values of $D_m$ (median value of both distributions can be found in Table 3). The histograms for the retrieved IWC (blue color plot panels, third column in Fig. 14) are plotted using logarithmic $x$ axis for visualization purposes. The statistical results showed that the greater part of the detected ice hydrometeors is found to have IWC values $3 \times 10^{-4}$–$3 \times 10^{-1}$ g m$^{-3}$ ($-3.5$ to $-0.5$ in the logarithmic axis) when oblate ice spheroids were assumed. For horizontally aligned prolate ice particles, most of the detected ice hydrometeors are found to have IWC values between $1 \times 10^{-4}$–$1 \times 10^{-1}$ g m$^{-3}$ ($-4$ to $-1$ in the logarithmic axis). The spikes in both $D_m$ and IWC histograms are merely caused from the strong discrepancies between simulated and measured radar variables during the minimization of $J_1$ and $J_2$ in Eq. (9), i.e., negative measured values of DWR, while the minimum value 0.1 dB was used in the simulations (see also Appendix C). The different color shades in all panel plots denote the different temperature groups in which the detected hydrometeors are separated. For both shape assumptions, it is observed that when temperature drops below $-25$ °C ice hydrometeors populations with IWC $< 1 \times 10^{-2}$ g m$^{-3}$ dominate the particles distribution. Furthermore, for higher temperatures the greater part of the $\log_{10}$IWC distribution is shifted towards larger values in the logarithmic axis, denoting larger retrieved IWC.

For better interpretation of the ice retrieval results during the three snow events, we further proceeded with the calculation of some descriptive statistics presented in Table 3, always under the assumption that the detected ice hydrometeors can be represented by ice spheroids whose m($D_{max}$) corresponds to that of aggregates and they follow a PSD with $\mu = 0$. The median of the retrieved properties for the observed particles distributions was calculated. Anticipating that the detected ice particles can be represented by oblate spheroids, we calculated the median retrieved $S = 0.45$, the median retrieved $D_m = 0.80$ mm and the median retrieved IWC $= 13 \times 10^{-3}$ g m$^{-3}$. On the contrary, when the observed hydrometeors were assumed to be horizontally aligned prolate spheroids, the median retrieved sphericity, the median retrieved median mass diameter and the median retrieved ice water content, were found $S = 0.45$, $D_m = 1.08$ mm and IWC $= 5 \times 10^{-3}$ g m$^{-3}$, respectively. Although the two median $S$ are the same, there are differences in the median $D_m$ and IWC between oblates and horizontally aligned prolates. For the latter, the median $D_m$ was calculated larger and the IWC was calculated lower than the respective values for oblate ice spheroids. Therefore, the shape assumption seemed to affect the retrieved microphysical properties of the ice particles (also shown in Sect. 4.1). In Table 3 the 10[th] and 90[th] percentile of the detected ice hydrometeors retrieved parameters can be also found.

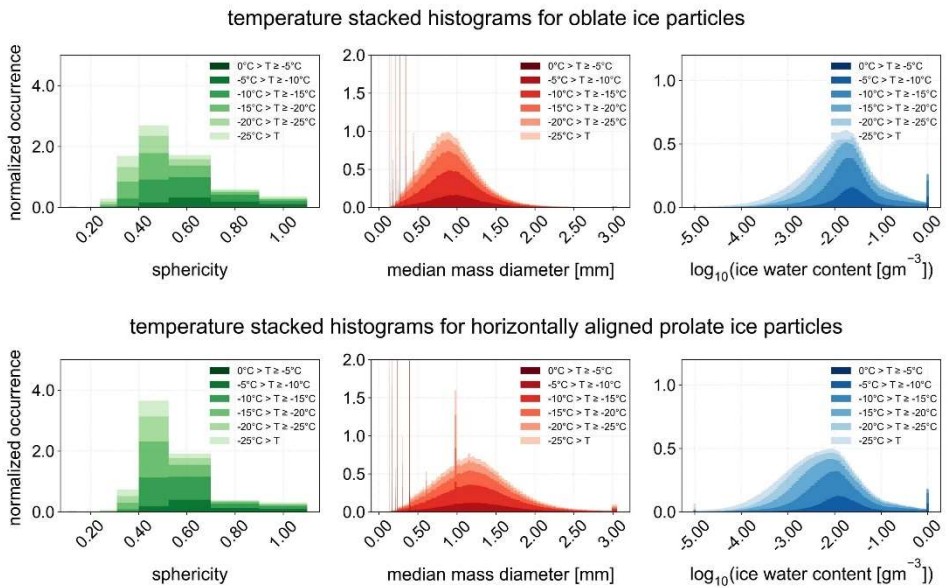

**Figure 14:** Temperature stacked histograms for all RHI scans on 9[th], 10[th] and 30[th] January 2019 for oblate and horizontally aligned prolate ice particles using the retrieval output for ice spheroids m($D_{max}$) to be the aggregates from Yang et al. (2000).

**Table 3:** Statistical description of the retrieved parameters for oblate and horizontally aligned prolate ice spheroids that follow mass-size relation of aggregates from Yang et al. (2000) for all RHI scans on 9th, 10th and 30th January 2019.

| Shape assumption | Statistical Description | Sphericity | Median Mass Diameter [mm] | Ice Water Content [g m⁻³] |
|---|---|---|---|---|
| **Oblates** | Median | 0.45 | 0.80 | $13 \times 10^{-3}$ |
| | 10th percentile | 0.35 | 0.27 | $11 \times 10^{-4}$ |
| | 90th percentile | 0.80 | 1.36 | $11 \times 10^{-2}$ |
| **Horizontally aligned prolates** | Median | 0.45 | 1.08 | $5 \times 10^{-3}$ |
| | 10th percentile | 0.45 | 0.40 | $4 \times 10^{-4}$ |
| | 90th percentile | 0.80 | 1.82 | $6 \times 10^{-2}$ |

## 4.3 Discussion

One limitation of the current version of the ice retrieval is the need to make some significant a-priori assumptions about the particle properties. At first, we selected the ice spheroid model, as the model with small number of parameters to be pre-defined, to represent the detected ice particles. Then, we decided about the PSD that the detected ice particles follow. For this decision, several studies argue that a typical PSD is described by a shape parameter close to 0 for ice particles (e.g., Matrosov and Heymsfield, 2017). Therefore, we also chose an exponential PSD for the simulated ice spheroids. The use of the two aforementioned assumptions were not further investigated in this study and were used as already described. The third assumption (discussed in Sect. 4.1) concerns the choice if we assume oblate or horizontally aligned prolate ice spheroids. In addition to the shape, we have to assume a suitable $m(D_{max})$ relationship for the prevalent ice particles. For the three investigated snow events the selection of $m(D_{max})$ of aggregates over the BF95 for ice spheroids has been partially discussed in Sect. 3.2. An extended explanation for this selection is also presented in Sect. 4.3.1.

## 4.3.1 Unknown mass-size relationship

From the two $m(D_{max})$ relations initially used for ice spheroids in the scattering simulations, only the $m(D_{max})$ of aggregates was systematically used for the statistical analysis of this study. The BF95 mass-size relation could not represent our radar observations, especially for large ice hydrometeors. Although ice crystals are known to have lower densities when they grow larger (except for graupel or hail), BF95 prescribes near-zero density values for large particles especially when they are assumed homogeneous soft spheroids. The homogeneity that this ice model suggests along with the missing internal structures and the near-field ice dipoles interactions that the realistic ice particles have, leads to very low simulated ZDR values with increasing particles size (Fig. 8) and decreasing particles density. The ice retrieval results for the examined case study, using LUTs for oblate ice spheroids, an exponential PSD and BF95 mass-size relation, are presented in Fig. 15 along with the residual values of ZDR, DWR and $Z_e$, expressed using RMSE values in Table 4. The RMSE for ZDR and $Z_e$ are quite low and generally in the same order of magnitude like the RMSE using $m(D_{max})$ of aggregates for both oblate and horizontally aligned prolate shape assumptions (Table 2). However, with the aforementioned assumptions, the retrieved AR and $S$ (Fig. 15a and Fig. 15c) could not really explain ZDR measurements (Fig. 3d). The AR is almost unrealistically high, suggesting e.g., plates, for the greater part of the cloud cross-section using BF95 $m(D_{max})$. The RMSE for DWR with 2.27 dB was found to be quite high, suggesting that the retrieved $D_m$ (Fig. 15b) could not replicate DWR measurements (Fig. 3b). On the other hand, IWC (Fig. 15d) showed a good agreement to the radar measurements as it could explain well POLDIRAD $Z_e$ (Fig. 3c). The retrieved values were found to be larger than the IWC values retrieved using the $m(D_{max})$ of aggregates for both shape assumptions. Overall, the plots of retrieved parameters as well as the RMSE for ZDR, DWR and $Z_e$ reveal that the output of the ice retrieval using an exponential PSD, the $m(D_{max})$ of aggregates and LUTs for oblate ice spheroids (Fig. 11 and Table 2, but also in Table 4) were found to better explain the radar observations compared to the BF95 assumption. Figure 16 shows the residuals between the simulated and measured DWR for aggregates (Fig. 16a) and BF95 (Fig. 16b) $m(D_{max})$. For ice spheroids that

follow the m(D$_{max}$) of aggregates, the residuals are evenly distributed around 0 (mean value of +0.08 dB) suggesting that this mass-size relation can better explain our measurements in this case. In contrast, the measured DWR appeared to be higher than the simulated one for BF95 for the larger part of the cloud cross-section (reddish areas) with a mean value of –0.923 dB.

To further investigate the significance of the m(D$_{max}$) relation for the retrieval result, we conducted a small sensitivity study using the aggregates assumption from Yang et al. (2000) which suggests an almost constant effective density, $\rho_{eff}$,

(approximately $\rho_{eff}$ = 0.2 g cm$^{-3}$) of ice particles with increasing size. Using this value as a reference, we created LUTs for oblate ice particles, 1) with twice and 2) with half the density of the aggregates mass-size relation (simulations shown also in Fig. 8 with red dashed and dash-dotted lines), always with the assumption that the ice spheroids follow an exponential PSD. Retrieval results for ice oblates with half, equal and twice the density of aggregates, are shown in (a)-(c), (d)-(f) and (g)-(i) panels of Fig. 17, respectively. Corresponding RMSE values for $Z_e$ are given in Table 4. Focusing on the IWC retrieval, we

obtain lower IWC values (with an RMSE = 0.28 dB for $Z_e$) for ice particles with twice the density of aggregates than the IWC values retrieved in Fig. 11d or Fig. 17f. Analogously, we retrieve larger IWC (with an RMSE = 0.23 dB for $Z_e$) for ice particles with half the density of aggregates. In Table 4 the residual values expressed as RMSE for DWR and ZDR can also be found. When the ice spheroids are denser with doubled the aggregates $\rho_{eff}$, the DWR RMSE is 0.50 dB, while the DWR RMSE is 0.54 dB for the less dense ice spheroids. The RMSE for ZDR are found to be similar with 0.21 dB and 0.20 dB when ice

spheroids with twice and half the density of aggregates, respectively, are assumed. However, the denser ice spheroids assumption (doubled effective density of aggregates mass-size relation) suggests the presence of more spherical particles compared to more aspherical particles when less dense ice spheroids (halved effective density of aggregates) are assumed.

The aforementioned examples indicate the limitations of the ice retrieval to provide realistic microphysics information when different assumptions have to be considered. To overcome the weakness of the fixed mass-size relation, but still keep

our approach simple, a combination of m(D$_{max}$) relations considering the different behavior of large/softer and small/denser ice particles would help the retrieval producing more realistic microphysics results. Additional measurements (e.g., Doppler velocity), in a possible extension of this method, might be able to replace the fixed m(D$_{max}$) assumption used in this approach with a whole set of m(D$_{max}$) depending on the average ice particles density affected by the environment in which they are formed.

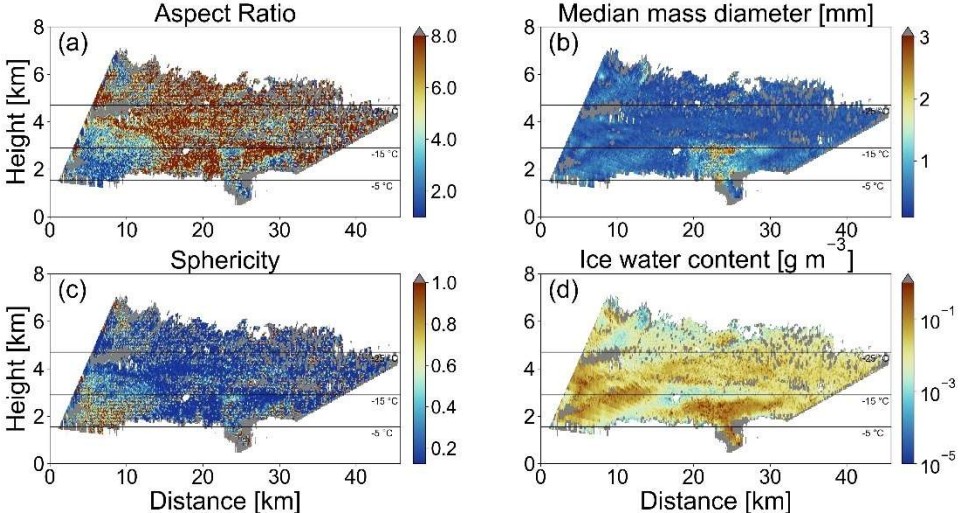

**Figure 15:** Retrieved (a) AR, (b) $\boldsymbol{D_m}$, (c) $S$ and (d) IWC for oblate ice particles for 30th January 2019 at 10:08 UTC using BF95 $\mathbf{m(D_{max})}$. The –5 ºC, –15 ºC and –25 ºC temperature levels are plotted with black solid lines (source: Deutscher Wetterdienst, data provided by University of Wyoming; http://weather.uwyo.edu/upperair/sounding.html, last access: 08 April 2022). Areas where ice masked and noise-
825 filtered measurement values locate are plotted with grey color.

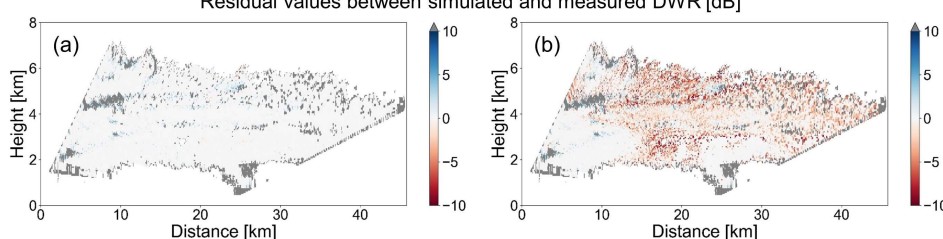

**Figure 16:** Difference (residuals) between simulated and measured values of DWR for ice spheroids that follow m(D$_{max}$) of (a) aggregates and (b) BF95 for 30$^{th}$ January 2019 at 10:08 UTC. For better visualization purposes the –5 ºC, –15 ºC and –25 ºC temperature lines are not plotted here. Areas where ice masked and noise-filtered measurement values locate are plotted with grey color.

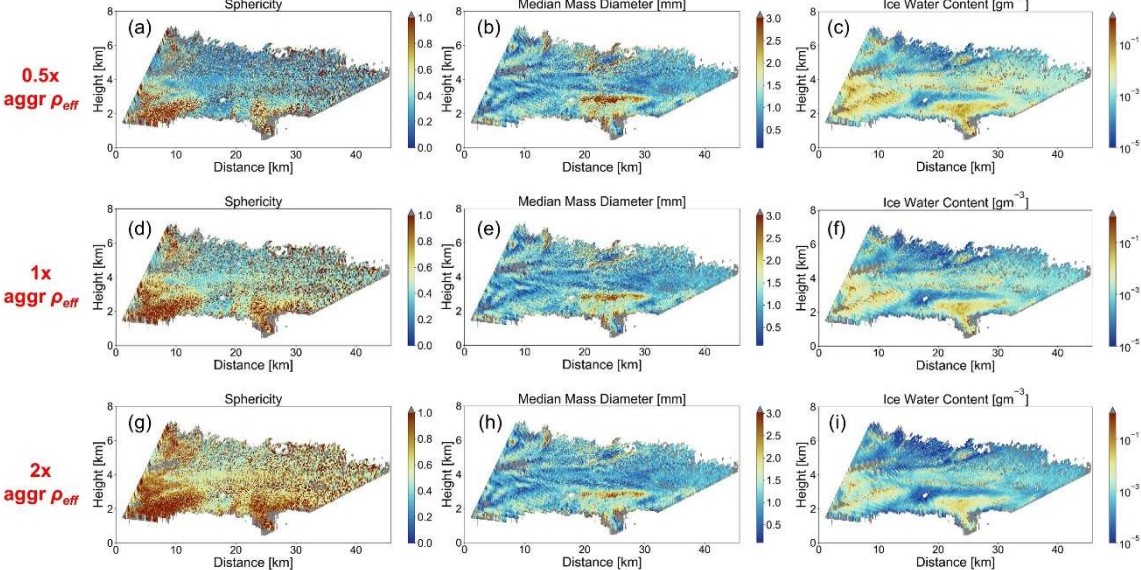

**Figure 17:** Ice microphysics results for oblate ice spheroids; sphericity, median mass diameter and ice water content using (a, b, c) 0.5x $\rho_{eff}$, (d, e, f) 1x $\rho_{eff}$ and (g, h, i) 2x $\rho_{eff}$ from Yang et al. (2000) m(D$_{max}$) of aggregates. For better visualization purposes the –5 ºC, –15 ºC and –25 ºC temperature levels are not plotted here. Areas where ice masked and noise-filtered measurement values locate are plotted with grey color.

**Table 4:** RMSE values for simulated ZDR, DWR, $Z_e$ compared to original observations for the whole radar cross-section after running the retrieval for 30$^{th}$ January 2019 at 10:08 UTC for oblate ice spheroids and different **m(D$_{max}$)** assumptions.

| m(D$_{max}$) assumption for ice spheroids | Parameter | RMSE |
|---|---|---|
| **BF95** | DWR | 2.27 dB |
| | ZDR | 0.25 dB |
| | $Z_e$ | 0.20 dB |
| **Aggregates Yang et al. (2000)** | DWR | 0.50 dB |
| | ZDR | 0.19 dB |
| | $Z_e$ | 0.20 dB |
| **Aggregates 2 times denser than Yang et al. (2000)** | DWR | 0.50 dB |
| | ZDR | 0.21 dB |
| | $Z_e$ | 0.28 dB |
| **Aggregates 0.5 times less dense than Yang et al. (2000)** | DWR | 0.54 dB |
| | ZDR | 0.20 dB |
| | $Z_e$ | 0.23 dB |

### 4.3.2 Comparisons to other ice retrievals

In the absence of collocated in-situ observations, we proceeded to compare our retrieval results with already established methods to infer ice microphysics. As a first step, we used the different aggregates mass-size relation assumptions from the

previous section to retrieve IWCs and to calculate the corresponding ice water path (IWP). Then, we compared our results to IWP data from MODIS MYD06_L2 product (Platnick, S., Ackerman, S., King, M. et al., 2017) for our measurements region. From MODIS, an averaged value of IWP ~ 90 g m$^{-2}$ was estimated for the whole radar cross-section. Using our retrieved IWC for the three assumptions (0.5x, 1x, 2x aggr. $\rho_{eff}$) we obtain IWP values of approximately 46 g m$^{-2}$, 80 g m$^{-2}$ and 137 g m$^{-2}$, respectively. In comparison, the original mass-size relation (1x aggr. $\rho_{eff}$) shows the best agreement with the results from MODIS.

To further evaluate our retrieved IWC, we compared our results with the IWC formula of Bukovčić et al. (2018) for dry snow (IWC (KDP, $Z_e$) = 0.71KDP$^{0.65}Z_e^{0.28}$, their Eq. 28). For this comparison, we used our C-band radar KDP along with $Z_e$ (and not S-band as the aforementioned literature suggests) for the presented case study (Fig. 3) to calculate IWC and IWP. The method of Bukovčić yields a much higher mean IWP (~ 2308 g m$^{-2}$) compared to our IWP (~ 80 g m$^{-2}$). A deeper investigation showed that their method assumes a much smaller melted equivalent particle diameter (~300 μm vs. our retrieved ~1 mm). As Bukovčić's method has to assume the particle size, we consider our IWP results more reasonable for a moderate snowfall case since our method explicitly retrieves the particles size along with the IWC.

Furthermore, we compared our scattering calculations to Matrosov et al. (2017), their Fig. 3b, to examine if we can reproduce their aspect ratio retrieval using the BF95 mass-size relation. Comparing the simulation lines we observe some differences between our calculations and those of the aforementioned literature. For the BF95 mass-size relation, a canting angle of 3° and a median volume diameter = 0.02 cm, ZDR = 1 dB would give an aspect ratio of 0.55 using Matrosov's approach, while our retrieval yields an aspect ratio of 0.45 and thus 20% lower values. In our manuscript we mainly use the aggregates m(D$_{max}$) from Yang et al. (2000). For this assumption, ZDR = 1 dB is in line with an aspect ratio of ~ 0.38 for the same median volume diameter of 0.02 cm. For both mass-size relation assumptions, our retrieved aspect ratio is lower than Matrosov's approach. Some of this disagreement could be attributed to a different modelling of the composite refractive index.

### 4.3.3 Particles shape and viewing geometry

The retrieval results in both Fig. 11 and Fig. 12 showed that some areas are affected more than others from the shape assumption inside the cloud cross-section. For instance, in the region above the Ka-band radar, the retrieved $D_m$ was found to be lower when oblate ice spheroids were used compared to horizontally aligned prolates. Conversely, the lower values of $D_m$ for ice oblates lead to a higher retrieved IWC. In these areas, the use of the polarimetric signature from POLDIRAD, i.e., ZDR, is furthermore found to be crucial: not only to constrain the shape but also to reduce ambiguities between size and mass of the detected ice hydrometeors. To further evaluate the performance of the retrieval, 2d density histograms between retrieved $D_m$ and measured DWR were created for the oblate shape assumption, including all 59 RHI scans. The histograms are presented in Fig. 18 for elevation angles $\theta_C = \theta_{Ka} = 30°$ (Fig. 18a) and $\theta_C = 10°$, $\theta_{Ka} = 90°$ (Fig 18b). The first observation geometry (Fig. 18a) is a region located between both radar instruments, while the second one (Fig. 18b) located directly above the Ka-band radar site. On top the density histograms, the DWR-$D_m$ simulations for different values of AR are plotted with grey lines. In (Fig. 18a), the simulations as well as the retrieved $D_m$ are more closely distributed than in (Fig. 18b). The close distribution of the DWR-$D_m$ lines in Fig. 18a suggests that the shape and the size retrieval are not strongly correlated in the region between both radar systems since the simulated DWR does not change much with AR. In the region above the Ka-band cloud radar, however, polarimetric measurements from the C-band weather radar POLDIRAD (i.e., ZDR) help to narrow down the solution space of the size retrieval (Fig. 18b) by providing information about the ice particle shape. This behavior is fully explained in Fig. 19 where the radar beams passing through ice oblate spheroids are drawn. In Fig. 19a, the radar beams from the two instruments penetrate oblate spheroids with different AR with the same elevation angle $\theta_C = \theta_{Ka} = 30°$. From the radar viewing geometry this is supposed to happen in cloud regions located between both radar instrument. In Fig. 19b, the elevation angle for C-band is $\theta_C = 10°$, while the Ka-band points to zenith with $\theta_{Ka} = 90°$. In both cases, the radar beams penetrate three different shaped ice oblates that are aligned with their maximum dimension in the horizontal plane and which are chosen to

have the same $D_{max}$. In Fig. 19a, the length of the Ka-band beam does not change dramatically inside the oblate ice particle.
In Fig. 19b, however, the MIRA-35 beam length through the oblate ice particle, and hence the DWR, is very sensitive on the aspect ratio. Therefore, the DWR-$D_m$ relationship becomes quite sensitive to AR in this area, especially when particles are assumed to be horizontally oriented. From similar geometric considerations, the region between both radars at very low elevation angles is another region in which the size retrieval benefits from the AR constraint. In the case of variable ice crystal shapes, ZDR from POLDIRAD is, thus, very helpful for the $D_m$ estimation.

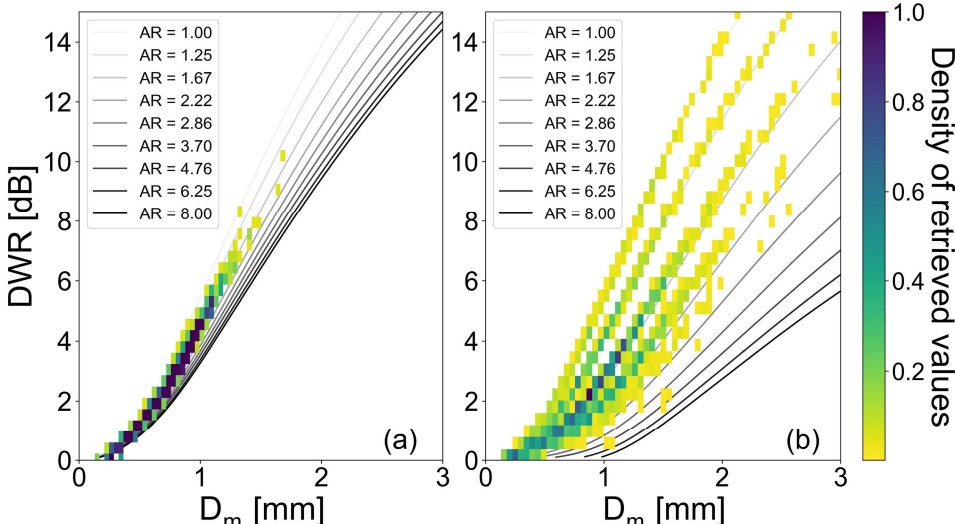

**Figure 18:** 2d density histograms between retrieved $\boldsymbol{D_m}$ and measured DWR for different observation geometries. (a) Between both radars with $\theta_C = \theta_{Ka} = 30°$ and (b) above the Ka-band radar $\theta_C = 10°$, $\theta_{Ka} = 90°$. With grey lines the DWR and $\boldsymbol{D_m}$ simulations are plotted for different values of AR using oblate ice spheroids with $\mathbf{m(D_{max})}$ of aggregates.

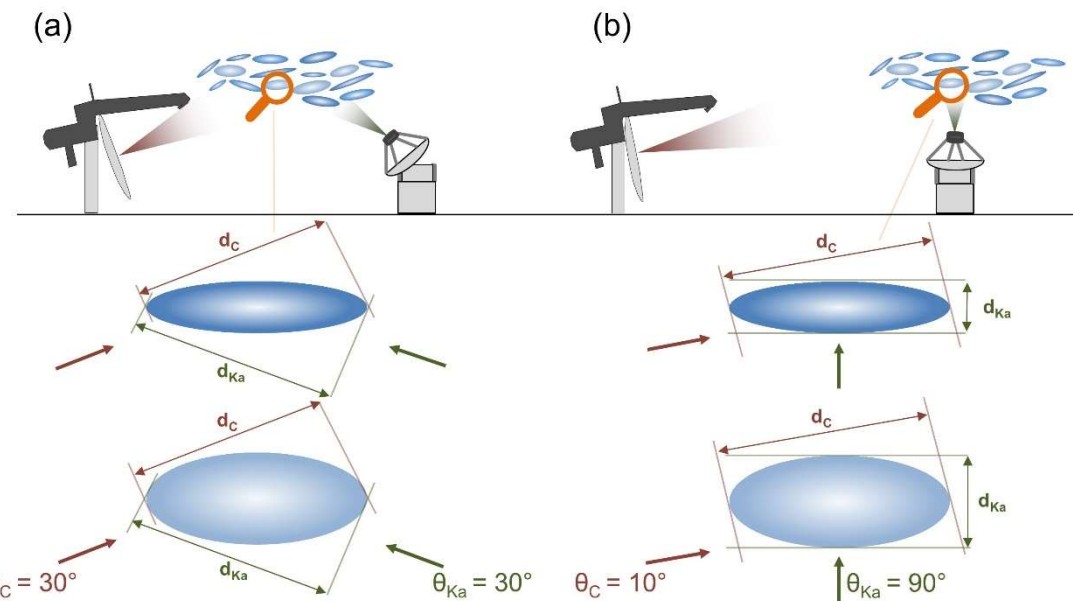

**Figure 19:** Radar beam geometries through oblate ice spheroids with different AR values for (a) $\theta_C = \theta_{Ka} = 30°$ and (b) $\theta_C = 10°$, $\theta_{Ka} = 90°$.

## 5 Conclusion

In the present study, we exploited dual-wavelength radar observations from the spatially separated weather radar POLDIRAD and cloud radar MIRA-35 to estimate the size of ice hydrometeors. Introducing a novel approach, we used the differential radar reflectivity from the weather radar to constrain the particle shape during the particle size retrieval. To this

end, we calculated scattering properties for a variety of ice particles using PyTMatrix with AR/$S$, $D_m$ and IWC as degrees of freedom for the simulated ice spheroids. Scattering simulations for all possible viewing geometries between the cross-section of the radar instruments were then compiled in LUTs and compared to radar observations implementing an ice microphysics retrieval scheme. In this scheme, the ice particles were selected to be represented by soft spheroids. Using the microphysics retrieval and making some a-priori assumptions about the shape, the PSD and the mass-size relation of the ice spheroids, we obtained AR/$S$, $D_m$ and IWC in ice cloud regions. Next to these parameters, we also calculated the attenuation by ice hydrometeors and corrected our radar observations. Besides attenuation, the uncertainty of the radar calibration has been considered. In addition, the impact of the spatiotemporal mismatch between RHI scans and the volumetric mismatch between the radar beams on the measured DWR were analyzed. All aforementioned errors were subsequently propagated through the retrieval to obtain an error estimation.

Three snow events from January 2019 were used to test the ice microphysical retrieval. The retrieved parameters for shape, size and mass could reasonably explain the radar measurements of ZDR, DWR and $Z_e$ when the detected ice particles were assumed to be represented by oblate spheroids (smaller RMSE and retrieval errors than for horizontally aligned prolates) that follow an exponential PSD and the m($D_{max}$) of aggregates from Yang et al. (2000). It was also found that the well-known BF95 assumption could not represent our dataset for large particles as the density of large ice spheroids from BF95 was calculated lower than this of the detected ice hydrometeors. This is merely caused when BF95 is used along with the ice soft spheroid model. The homogeneity that this model suggests, along with the lack of the near-field ice dipoles interactions that the realistic ice particles have against soft spheroids, lead to low simulated effective density and lower simulated ZDR values. For that reason, BF95 m($D_{max}$) combined with the soft spheroidal model could not produce pronounced polarimetric signals matching our ZDR measurements. Although the assumption of aggregates m($D_{max}$) for ice spheroids could better explain our ZDR-DWR observations, it suggests an almost constant density with increasing particle size, i.e., small ice crystals (columns or plates) appear to have the same density as larger ice particles (aggregates). Therefore, we still need a m($D_{max}$) relation which describes a more realistic function between density and size. Here, additional measurements, e.g., Doppler velocity of ice hydrometeors, could be exploited in future studies to provide a more variable m($D_{max}$) relation instead of a fixed one. In conclusion, the assumption of the m($D_{max}$) relation can have a great impact on the retrieval. Moreover, our approach for the retrieval's error assessment showed that, although the beam width mismatch can locally lead to significant deviations, the calibration uncertainty can lead to the largest bias throughout the averaged $D_m$ and IWC profile. Non-uniform beam filling effects, however, can locally have strong impacts (of several dB) on DWR measurements. Subsequent studies certainly need to explore this effect for spatially separated radars in more detail and need to develop techniques to detect and filter out these regions. Further analysis on the error of the ice retrieval showed that the shape assumption is equally important to the calibration uncertainty for the for retrieved $D_m$ profile, while it is less important than the calibration uncertainty for the retrieval of IWC.

Nevertheless, promising microphysics information can be obtained from the combination of dual-wavelength and polarimetric measurements from spatially separated radars. This combination, i.e., DWR and ZDR, can reduce the ambiguity in $D_m$ retrievals caused by the variable aspect ratio AR of ice particles. While we found some influence of AR on $D_m$ retrievals in the region between both radar instruments and at high elevation angles (e.g., 30°), ZDR from POLDIRAD was very helpful to improve $D_m$ retrievals above the Ka-band cloud radar, or in the areas between both systems where the elevation angles of both radars are low. In these regions, ZDR measurements are essential to reduce the uncertainty in $D_m$ retrieval from DWR measurements of horizontally aligned ice oblates.

The current version of the ice microphysics retrieval scheme considers only dry ice particles. In future studies, this methodology will be extended to include wet particles as well. In this way, we aim for a better understanding of microphysical processes of ice growth, such as aggregation or riming, to improve their representation in future weather and climate models.

**Appendix A: Radar measurements error assessment**

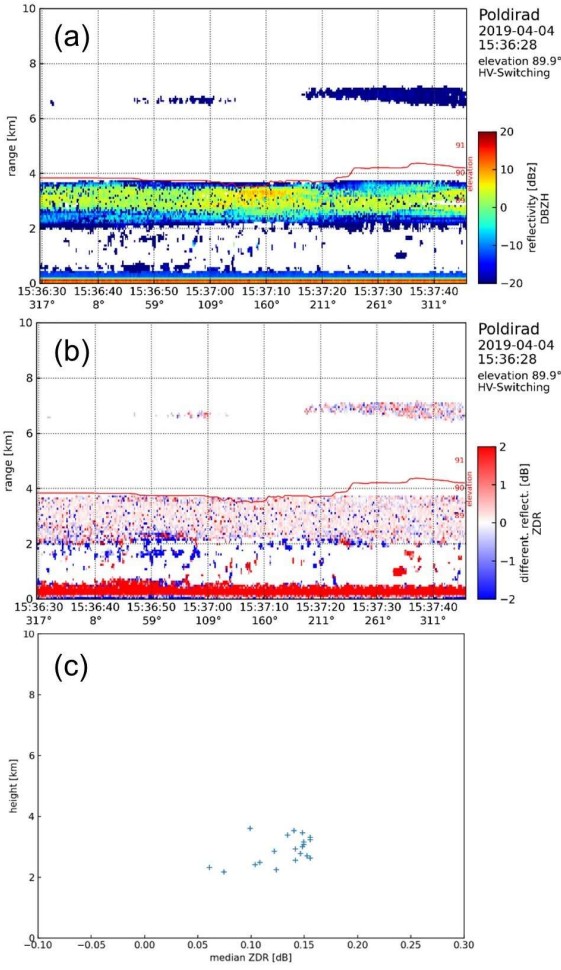

**Figure A1:** POLDIRAD (a) $Z_e$, and (b) ZDR measurements of a liquid cloud layer for different times and azimuth angles with a vertical pointing antenna on 4th April 2019. Panel (c) shows the offset of the averaged ZDR for the range where the liquid layer was detected.


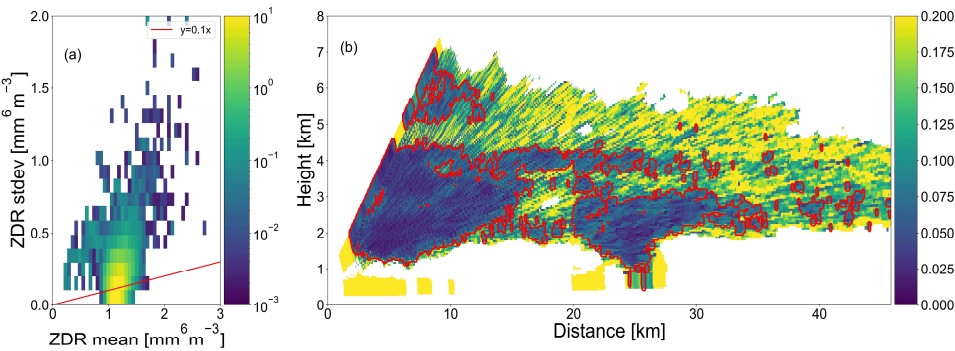

**Figure A2:** (a) The local standard deviation of ZDR is plotted as a function of the local mean of ZDR using a 2d density histogram of the calculated parameters (colorbar indicates the density values). (b) The ratio $a =$ ZDR$_{stdev}$ /ZDR$_{mean}$ can be used to filter out noisy ZDR measurements (values of ZDR are indicated with the corresponding colorbar). In the red encircled areas ($a < 0.1$), the retrieval results are considered to be reliable enough to be aggregated into statistical results.

**Appendix B: Development of an ice mask**

For the ice mask implementation, variables from both radars, i.e., the LDR from MIRA-35 as well as the ZDR and ρHV from POLDIRAD, were used. These variables are known to have distinct polarimetric signatures when a ML is present. The mask was applied to each vertical profile of the common grid for every pair of RHI scans. Below 4 km a ML is detected for the following condition: MIRA-35 LDR is in the range $-22$ dB $\leq$ LDR $\leq -15$ dB and POLDIRAD ρHV as well as ZDR are in the range $0.75 \leq$ ρHV $\leq 0.95$ and $1.5$ dB $\leq$ ZDR $\leq 2.5$ dB, respectively. As we merely focus on stratiform snowfall precipitation

cases and as we assume that riming or melting ice is unlikely to occur, all hydrometeors above 4 km (height above MSL) and/or above ML were accounted dry. When the criteria were not met, the isotherm of 0 ºC was used as an auxiliary information for ice above that height. The temperature data were obtained from the Oberschleißheim sounding station (about 13 km north of Munich, source: Deutscher Wetterdienst, data provided by University of Wyoming; http://weather.uwyo.edu/upperair/sounding.html, last access: 08 April 2022). Although the thresholds used in the ice mask

were evaluated in precipitation cases where a ML was observed, it is required either to investigate more precipitation cases obtaining more precise thresholds, or use already established ML detection algorithms exploiting polarimetric radar observations, e.g., Wolfensberger et al. (2016).

The necessity of sharpening our ice mask's thresholds is highlighted from Fig. B1 where an example of ZDR observations during a thunderstorm observed over Munich on 7th July 2019 at 08:22 UTC with a ML detected at 3 km, is presented. Figure

B1a shows ZDR without the application of any noise filters (described in Sect. 3.1), while in Fig. B1b the filtered and masked ZDR is plotted. Figure B1c presents the origin of the masked ZDR values. The greater part of the cloud cross-section is masked using the 0 ºC isotherm revealing the need for more precise ice thresholds with evaluating more case studies with mixed-phase cloud cross-sections. In our investigated case studies, a ML was never detected and only a very small part of the cloud cross-section was masked using the 0° isotherm at some cases.


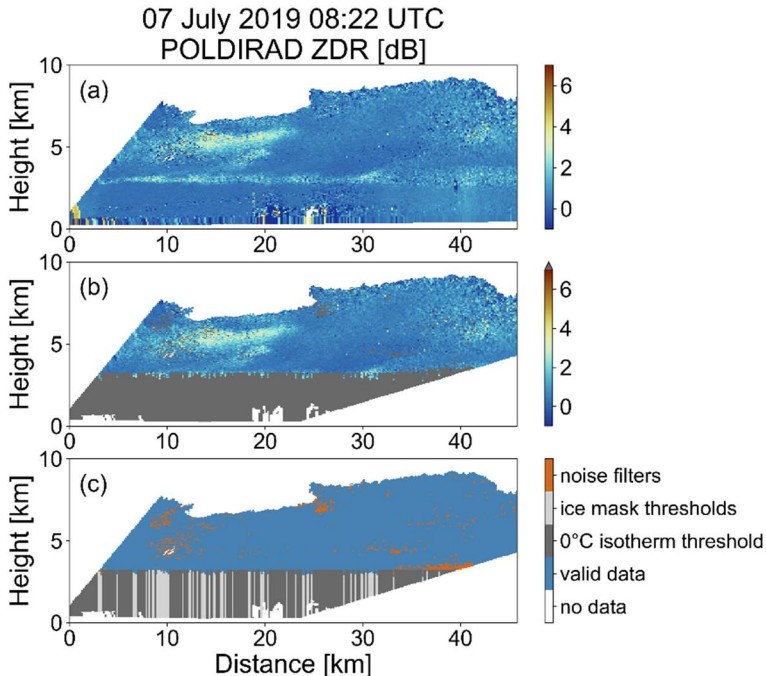

**Figure B1:** (a) Unfiltered POLDIRAD ZDR measurements from 7th July 2019 at 08:22 UTC. (b) Noise-filtered and ice masked values of POLDIRAD ZDR plotted with grey color. (c) Different origin of filtered and masked values plotted with different color.

### Appendix C: Estimation of minimum retrievable $D_m$

For sensitivity purposes regarding DWR measurements we had to consider a minimum $D_m$ in our simulations. For this reason, we assumed a minimum value of DWR = 0.1 dB that can be observed by the two radars. The minimum retrievable $D_m$ depends not only on the viewing geometry of the two radars but also on the AR and the $m(D_{max})$ used for the calculation of the ice spheroids density. In Fig. C1 examples of the minimum retrievable $D_m$ for different radar geometries and $m(D_{max})$ are presented. For this figure, the $m(D_{ma})$ of aggregates is used for red and dark red line plots for the mass estimation of the ice

spheroids. When the two radar beams are both simulated to be emitted horizontally the horiz-horiz definition is used, while when C-band beam is simulated be emitted horizontally and Ka-band towards zenith the horiz-vert label is used in the legend. As all ice spheroids are assumed to be aligned to the horizontal plane with small oscillations of up to 20° out of this plane, the

minimum retrievable $D_m$ is, in general, smaller when the radar beams are passing through the ice spheroids from the side. Assuming C-band beam emitted horizontally and for ice particles with the same size, Mie effects can be stronger for Ka-band

beam when it penetrates the particles from the side (horiz-horiz geometry) rather than from below (horiz-vert geometry), as the beam path is longer inside the particle. For horiz-horiz geometry, $Z_{e,Ka}$ values are lower and thus, DWR is higher than for horiz-vert geometry for same particle size. Therefore, the lowest minimum retrievable $D_m$ is smaller in horiz-horiz than in horiz-vert geometry. From the comparison of red (ice spheroids that follow m($D_{max}$) of aggregates) and blue (ice spheroids that follow m($D_{max}$) of BF95) color line plots, in which the radar beams are simulated to be emitted horizontally (C-band)

and vertically (Ka-band), the minimum retrievable $D_m$ using ice spheroids with m($D_{max}$) analog to this of aggregates is larger compared to this of BF95, due to the higher effective density of aggregates assumption for ice spheroids of the same size. The less dense the particles are, the smaller the $D_m$ will be for the minimum DWR threshold of 0.1 dB. For the same geometry (C-band emitted horizontally and Ka-band emitted vertically) and both mass-size relations, the more aspherical the particles the larger the minimum retrievable $D_m$ due to the larger cross-section of the horizontally aligned spheroids for the Ka-band beam.


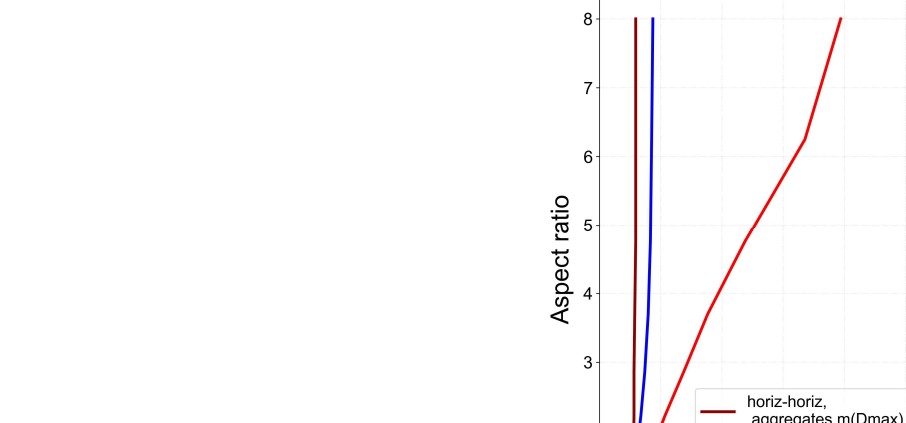

**Figure C1:** Minimum possible retrieved $D_m$ using ice spheroids with $\mathbf{m(D_{max})}$ analog to aggregates when C-band and Ka-band beam are emitted horizontally (dark red). With red and blue color, the minimum possible retrieved $D_m$ for ice spheroids with $\mathbf{m(D_{max})}$ analog to aggregates and BF95 when C-band beam is emitted horizontally and Ka-band is emitted towards zenith is plotted.

**Data availability**

Hersbach et al. (2018) was downloaded from the Copernicus Climate Change Service (C3S) Climate Data Store (CDS). doi: 10.24381/cds.bd0915c6. The author did not download the data to distribute them. The results contain modified Copernicus Climate Change Service information 2020. Neither the European Commission nor ECMWF is responsible for any use that may be made of the Copernicus information or data it contains. Ice water path data were taken from the MODIS MYD06_L2

product with the identifier doi: 10.5067/MODIS/MYD06_L2.NRT.061 (Platnick, S., Ackerman, S., King, M. et al., 2017). The temperature and wind speed profile data used in this study were provided by the University of Wyoming; http://weather.uwyo.edu/upperair/sounding.html (source: Deutscher Wetterdienst). Radar data from POLDIRAD weather radar from DLR as well as MIRA-35 cloud radar from LMU are available upon request to the authors.

## Author contribution

In the framework of the IcePolCKa project ET, FE, MH and GK performed radar measurements during snowfall events. ET developed the method used in this work and wrote the final paper under the consultation of FE. SG and TZ contributed to this work with productive discussions during the development of the ice microphysics retrieval. All authors contributed helpful comments to the manuscript.

## Competing interests

The authors declare that they have no conflict of interest.

## Acknowledgements

Eleni Tetoni gratefully acknowledges the "Investigation of the initiation of convection and the evolution of precipitation using simulations and polarimetric radar observations at C- and Ka-band (IcePolCKa)" project funded by the Deutsche Forschungsgemeinschaft (DFG, German Research Foundation) under grant number HA 3314/9-1. Some of the figures of the present manuscript were created using the freely available "Scientific colour maps 7.0" package kindly provided by Crameri et al. (2020), with the identifier doi: 10.1038/s41467-020-19160-7. Panels (a) and (c) in Fig. 3 were created using the ARM Radar Toolkit (Py-ART) developed by Helmus and Collis (2016), with the identifier doi: 10.5334/jors.119. The authors would like to thank Dr. Klaus Gierens and Dr. Luca Bugliaro for the internal review of the manuscript.

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
