# Peer review of "Retrievals of ice microphysical properties using dual-wavelength polarimetric radar observations during stratiform precipitation events"

_Atmospheric Measurement Techniques, 2021_

## Referee Comment (RC1)

Review of amt_2021_216

The manuscript "Retrievals of ice microphysics using dual-wavelength polarimetric radar observations during stratiform precipitation events" is substantially improved from its previous version. However, the authors need to address the issue regarding the gamma distribution shape factor before the final publication. A major revision is recommended.

Major comments:

A major issue that remains unaddressed is the value of the shape factor μ of the gamma distribution used in this study, μ = 4. That is very high for aggregated snow – it is usually close to 0 (hence gamma PSD is practically reduced to exponential), as numerous previous studies suggest (Gunn and Marshal, 1958; Sekhon and Srivastava, 1970; Lo and Passarelli, 1982; Mitchell et al. 1990; Field and Heymsfield, 2003; Tiira et al. 2016; Matrosov and Heymsfield, 2017). The authors need to explain why there is such a large discrepancy in μ regarding previous studies. See the specific comment.

Specific comments:

Line 31: Comma is missing after the "incoming solar radiation".

Line 70-71: Reflectivity factor Z is close to the 4th PSD moment in low-density snow, where the snow density is inversely proportional to particle diameter; it is proportional to the 6th DSD/PSD moment in rain. The authors should briefly comment on this.

Lines 274-275: There is a typo in the sentence: "...exceeds the noise ZDRstdv by on magnitude".

Lines 365-366: The shape parameter μ = 4 seems too high for the aggregates, it is usually close to 0. Why is the value μ = 4 chosen? Explain the rationale and provide some evidence why μ = 4 works for your cases. Is this because you are using Dmelted in PSD calculations? Do other parameters need adjustments to be used with Dmelted?

Lines 460-465: You did not take into account how the usage of AR=1.60 instead of AR=1.67 affects your results. Briefly comment on this, provide some estimates.

Figure 17: You should add a panel for sphericity, median mass diameter, and ice water content retrieved with the initial aggregate density (for easier comparison) and reflect the results in Table 4.

References:

Gunn, K. L. S., and J. S. Marshall, 1958: The distribution with size of aggregate snowflakes. *J. Meteor.*, **15**, 452–461.

Sekhon, R. S., and R. C. Srivastava, 1970: Snow size spectra and radar reflectivity. *J. Atmos. Sci.*, **27**, 299–307.

Lo, K. K., and R. E. Passarelli Jr., 1982: Growth of snow in winter storms: An airborne observational study. *J. Atmos. Sci.*, **39**, 697–706.

Mitchell, D. L., R. Zhang, and R. Pitter, 1990: Mass-dimensional relationship for ice particles and the influence of riming on snowfall rates. *J. Appl. Meteor.*, **29**, 153–163.

Field, P. R., and A. J. Heymsfield, 2003: Aggregation and scaling of ice crystal size distributions. *J. Atmos. Sci.*, **60**, 544–560.

Tiira, J., D. N. Moisseev, A. Von Lerber, D. Ori, A. Tokay, L. F. Bliven, and W. Petersen, 2016: Ensemble mean density and its connection to other microphysical properties of falling snow as observed in southern Finland. *Atmos. Meas. Tech.*, **9**, 4825–4841.

Matrosov, S., and A. Heymsfield, 2017: Empirical relations between size parameters of ice hydrometeor populations and radar reflectivity. *J. Appl. Meteor. Climatol.*, **56**, 2479–2488.

---

## Community Comment (CC1)

This study introduces a method for the retrieval of snow microphysical properties (ice water content, mean size, shape) from polarimetric and dual-wavelength radar observations. The method as introduced here is based on a bunch of rather rude assumptions, in particular the shape model (homogeneous spheroids) and the mass-size relation. In my view, the study lacks to show the effect that these assumptions have on the retrieval results and to properly state the limitations of the method.

To detail and justify my concerns:

The authors write (L588) "we have to assume a suitable m(Dmax) relationship" and I cannot agree more.

The authors obviously define "suitable" from the agreement of their scattering calculations with statistics of observations of DWR and ZDR. The scattering calculations are based on a range of assumptions, including e.g the size distribution, the mass-size relation, the spheroidal particle model, the shape of the spheroids.

Out of these, mass-size relation is rather well constrained by observations of a range of different measurement techniques (including 3D imaging of falling snow "in the wild" (Leinonen et al., 2021)) and its range of variation is comparably small. Moreover, it is a crucial microphysical parameter in weather and climate models.

On the other hand, the spheroidal particle model is a highly artificial model that is known to be not well suited to represent scattering properties of particles of low effective density like snow aggregates. This regards microwave scattering properties in general (e.g. Eriksson, 2015; Eriksson, 2018), but polarimetric properties in particular (eg. Schrom and Kumjian, 2018). Aspect ratio, in addition, is a highly simplistic parameter to describe the shape of typically irregularly shaped particles. That is, it is questionable how well aspect ratios observed from irregularly shaped particles can constrain reasonable values for homogeneous soft particles (like spheroids or even plates).

To summarize, mass-size relation is a well-constrained parameter, while aspect ratio is not and is a rather artificial parameter. Hence, I wonder (and question), why the authors chose to use a mass-size relation that is unreasonable for snow aggregates but insist on keeping the aspect ratio within "reasonable" bounds.

As I understand, the results of the retrieval method presented here strongly depend on the choice of mass-size relation. This poses the question how reliable retrieved microphysical properties (primarily IWC) are when selecting a relation that is far off the range of commonly accepted values. As stated above, I miss a comprehensive estimate of the uncertainties (or, actually, errors) that this unreasonable assumption results in - not within the (forward modelling-retrieval) system here, which is self-consistent, but in more realistic retrievals (if independently measured IWC are not available, e.g. by analysing the retrieved IWC based on a range of for different m(Dmax) in the retrieval).

The introduction to the discussion section points out two a priori assumptions on the particle properties as limitations of the method: the already discussed mass-size relation and the choice of particle model between oblate and prolate spheroids, missing to point out the much more

crucial assumption of spheroids in general. This continues in the discussion of "unsuitability" of the BF95 mass-size relation, where no other reasons for the very low ZDR than the low density, seemingly exclusively resulting from the m(Dmax) assumption is discussed, which is re-iterated in the conclusions.

As source for their aggregate model, the authors cite Yang et al. (2000). I find this highly misleading. Yang et al. (2000) (as well as a range of follow-up papers building on it and extending it, including Hong et al., 2009, and Ding et al., 2017) targets the explicit modeling of scattering properties of irregularly shaped particles. The irregular, non-spherical and(!) non-spheroidal(!) shape is the crucial aspect of their shape model, the core element of that research. The authors of this study, however, reduce this complexity to the minor aspect of the underlying mass-size relation (L356: "another m(Dmax) that we use is the aggregates from Yang et al. (2000)", falsely implying the equivalence of the m(Dmax) with the entire Yang et al. aggregate definition). Mass-size relation is likely the most un-aggregate-ish characteristic of the Yang aggregate model, was - probably - selected with not much care at that time (is that even originating from Yang et al. themselves, or where has it been taken from by Yang et al.?), and has been pointed out as a shortcoming of the Yang-aggregates by other authors (e.g. Eriksson, 2018).

To justify reference to Yang, it would be interesting to see how the scattering properties of the authors' aggregates compare to microwave properties of the actual, irregularly shaped aggregate of Yang. As far as I am aware, they are not available for preferential orientation, but both Ding et al. (2017) and Eriksson et al. (2018) provide scattering properties for this habit ("8-column aggregate") at radar wavelengths for totally randomly orientation, which should allow for a comparison of the predicted reflectivities and dual-wavelength ratios.

**Minor comments:**

L376: Could you provide a reference for that Dm from the equation you provide is the Median mass diameter? In my understanding, when D therein is the melted mass equivalent diameter, then Dm is the mean mass diameter. Are median and mean really equal here?

L384: Please specify more precisely, what Maxwell-Garnett approach was used, in particular what material forms the matrix, what the inclusions, as this can make a lot of a difference (see e.g. Eriksson, 2015).

**References**

Ding, J., Bi, L., Yang, P., Kattawar, G. W., Weng, F., Liu, Q., and Greenwald, T.: Single-scattering properties of ice particles in the microwave regime: Temperature effect on the ice refractive index

with implications in remote sensing, J. Quant. Spectrosc. Ra., 190, 26–37, 2017.

Eriksson, P., Jamali, M., Mendrok, J., and Buehler, S. A.: On the microwave optical properties of randomly oriented ice hydrometeors, Atmos. Meas. Tech., 8, 1913–1933, https://doi.org/10.5194/amt-8-1913-2015, 2015.

Eriksson, P., Ekelund, R., Mendrok, J., Brath, M., Lemke, O., and Buehler, S. A.: A general database of hydrometeor single scattering properties at microwave and sub-millimetre wavelengths, Earth Syst. Sci. Data, 10, 1301–1326, https://doi.org/10.5194/essd-10-1301-2018, 2018.

Hong, G., Yang, P., Baum, B. A., Heymsfield, A. J., Weng, F., Liu, Q., Heygster, G., and Buehler, S. A.: Scattering database in the millimeter and submillimeter wave range of 100–1000 GHz for nonspherical ice particles, J. Geophys. Res., 114, D06201, https://doi.org/10.1029/2008JD010451, 2009.

Leinonen, J., Grazioli, J., and Berne, A.: Reconstruction of the mass and geometry of snowfall particles from multi-angle snowflake camera (MASC) images, Atmos. Meas. Tech., 14, 6851–6866, https://doi.org/10.5194/amt-14-6851-2021, 2021.

Schrom, R. S. and Kumjian, M. R.: Bulk-Density Representations of Branched Planar Ice Crystals: Errors in the Polarimetric Radar Variables, Journal of Applied Meteorology and Climatology, 57, 333–346, https://doi.org/10.1175/JAMC-D-17-0114.1, 2018.

---

## Author Comment (AC2)

**Answers to Reviewer 1**

The manuscript "Retrievals of ice microphysics using dual-wavelength polarimetric radar observations during stratiform precipitation events" is substantially improved from its previous version. However, the authors need to address the issue regarding the gamma distribution shape factor before the final publication. A major revision is recommended.

Major comments:
A major issue that remains unaddressed is the value of the shape factor of the gamma distribution μ used in this study, μ = 4. That is very high for aggregated snow – it is usually close to 0 (hence gamma μ PSD is practically reduced to exponential), as numerous previous studies suggest (Gunn and Marshal, 1958; Sekhon and Srivastava, 1970; Lo and Passarelli, 1982; Mitchell et al. 1990; Field and Heymsfield, 2003; Tiira et al. 2016; Matrosov and Heymsfield, 2017). The authors need to explain why there is such a large discrepancy in regarding previous studies. See the specific comment.

Thank you for this comment. We acknowledge your point also justified by literature and have revised the parameter $\mu = 4$ to $\mu = 0$ in our pre-defined PSD throughout the study and updated our calculations and plots. Consequently, some retrieval results show significant differences in the retrieved $D_m$ between the two shape parameter values. In the following figures, the retrieved $D_m$ is larger when the PSD shape parameter is assumed 4 (left figure) instead of 0 (right figure). In both figures, the ice hydrometeors are assumed to be represented by oblate spheroids and the $m(D_{max})$ relationship is chosen to be analog to the aggregates of Yang et al. (2000).

[Figure]

Specific comments:
Line 31: Comma is missing after the "incoming solar radiation".

Thank you for pointing this out. A comma is now added after the "*incoming solar radiation*" phrase.

Line 70-71: Reflectivity factor Z is close to the 4th PSD moment in low-density snow, where the snow density is inversely proportional to particle diameter; it is proportional to the 6th DSD/PSD moment in rain. The authors should briefly comment on this.

Thank you for this comment. We have now added a sentence clarifying that the described definition of reflectivity factor Z refers to melted ice particles.

Lines 274-275: There is a typo in the sentence: "...exceeds the noise ZDRstdv by on magnitude".

The sentence is now rephrased as "*the signal ZDR_mean exceeds the noise ZDR_stdv by one order of magnitude*".

Lines 365-366: The shape parameter = 4 seems too high for the aggregates, it is usually close to 0. M Why is the value = 4 chosen? Explain the rationale and provide some evidence why = 4 works for μ

your cases. Is this because you are using Dmelted in PSD calculations? Do other parameters need adjustments to be used with Dmelted?

Considering the aforementioned literature in your major comments, $\mu$ is now set to 0, a characteristic value for snow aggregates. Moreover, the affected Fig. 7, Fig. 8 and Fig. 9 are now reproduced.

Lines 460-465: You did not take into account how the usage of AR=1.60 instead of AR=1.67 affects your results. Briefly comment on this, provide some estimates.

Thank you for this comment. In the latest version of LUTs we changed the AR values so that we include 1.67 for oblates (corresponding to the typical 0.6 from Hogan et al., 2012) in our LUTs. This is also changed in the text as well as in the affected plots. As for reference, the use of AR = 1.67 instead of AR = 1.6 results 5.03 dB instead of 5.06 dB for DWR and 0.36 dB instead of 0.31 dB for ZDR. Moreover, when we assume AR = 1.67 instead of the previous AR = 1.6 we obtain 9.76 dBZ instead of 9.72 dBZ for $Z_{eC}$ and 3.06 dBZ instead of 3.03 dBZ for $Z_{eKa}$. These values are calculated for PSD $D_m$ = 1 mm, IWC = 0.01 g m$^{-3}$ and both radar beams are simulated to be emitted horizontally.

Figure 17: You should add a panel for sphericity, median mass diameter, and ice water content retrieved with the initial aggregate density (for easier comparison) and reflect the results in Table 4.

A panel for the initial effective density of Yang et al. (2000) is now added in Fig. 17 and indeed helps the reader to directly have a look on the differences of the retrieved parameters. Moreover, RMSE results for DWR, ZDR and Ze using the initial aggregates m(Dmax) are added in Table 4. Thank you a lot for suggesting this.

References:
Gunn, K. L. S., and J. S. Marshall, 1958: The distribution with size of aggregate snowflakes. J. Meteor., 15, 452–461.
Sekhon, R. S., and R. C. Srivastava, 1970: Snow size spectra and radar reflectivity. J. Atmos. Sci., 27, 299–307.Lo, K. K., and R. E. Passarelli Jr., 1982: Growth of snow in winter storms: An airborne observational
study. J. Atmos. Sci., 39, 697–706.
Mitchell, D. L., R. Zhang, and R. Pitter, 1990: Mass-dimensional relationship for ice particles and the influence of riming on snowfall rates. J. Appl. Meteor., 29, 153–163.
Field, P. R., and A. J. Heymsfield, 2003: Aggregation and scaling of ice crystal size distributions. J. Atmos. Sci., 60, 544–560.
Tiira, J., D. N. Moisseev, A. Von Lerber, D. Ori, A. Tokay, L. F. Bliven, and W. Petersen, 2016: Ensemble mean density and its connection to other microphysical properties of falling snow as observed in southern Finland. Atmos. Meas. Tech., 9, 4825–4841.
Matrosov, S., and A. Heymsfield, 2017: Empirical relations between size parameters of ice hydrometeor populations and radar reflectivity. J. Appl. Meteor. Climatol., 56, 2479–2488.

References:

Hogan, R. J., Tian, L., Brown, P. R. A., Westbrook, C. D., Heymsfield, A. J., and Eastment, J. D.: Radar Scattering from Ice Aggregates Using the Horizontally Aligned Oblate Spheroid Approximation, J. Appl. Meteorol. Clim., 51, 655–671, https://doi.org/10.1175/JAMC-D-11-074.1, 2012.

Yang, P., Liou, K. N., Wyser, K., and Mitchell, D.: Parameterization of the scattering and absorption properties of individual ice crystals, J. Geophys. Res.: Atmos., 105, 4699–4718, https://doi.org/10.1029/1999JD900755, 2000.

---

## Author Response (AR1)

**Answers to Reviewer 1**

The manuscript "Retrievals of ice microphysics using dual-wavelength polarimetric radar observations during stratiform precipitation events" is substantially improved from its previous version. However, the authors need to address the issue regarding the gamma distribution shape factor before the final publication. A major revision is recommended.

Major comments:
A major issue that remains unaddressed is the value of the shape factor of the gamma distribution μ used in this study, μ = 4. That is very high for aggregated snow – it is usually close to 0 (hence gamma μ PSD is practically reduced to exponential), as numerous previous studies suggest (Gunn and Marshal, 1958; Sekhon and Srivastava, 1970; Lo and Passarelli, 1982; Mitchell et al. 1990; Field and Heymsfield, 2003; Tiira et al. 2016; Matrosov and Heymsfield, 2017). The authors need to explain why there is such a large discrepancy in regarding previous studies. See the specific comment.

Thank you for this comment. We acknowledge your point also justified by literature and we have now changed $\mu = 4$ to $\mu = 0$ in our pre-defined PSD. Some preliminary results show significant differences in the retrieved $D_m$ between the two shape parameter values of PSD. In the following figures, the retrieved $D_m$ is larger when the PSD shape parameter is assumed 4 (left figure) instead of 0 (right figure). In both figures, the ice hydrometeors are assumed to be represented by oblate spheroids and the aggregates $m(D_{max})$ relationship from Yang et al. (2000) is used.

[Figure]

Specific comments:
Line 31: Comma is missing after the "incoming solar radiation".

Thank you for pointing this out. A comma is now added after the "*incoming solar radiation*" phrase.

Line 70-71: Reflectivity factor Z is close to the 4th PSD moment in low-density snow, where the snow density is inversely proportional to particle diameter; it is proportional to the 6th DSD/PSD moment in rain. The authors should briefly comment on this.

Thank you for this comment. We have now added a sentence clarifying that the described definition of reflectivity factor Z refers to melted ice particles.

Lines 274-275: There is a typo in the sentence: "...exceeds the noise ZDRstdv by on magnitude".

The sentence is now rephrased as "*the signal $ZDR_{mean}$ exceeds the noise $ZDR_{stdv}$ by one order of magnitude*".

Lines 365-366: The shape parameter = 4 seems too high for the aggregates, it is usually close to 0. M Why is the value = 4 chosen? Explain the rationale and provide some evidence why = 4 works for μ

your cases. Is this because you are using Dmelted in PSD calculations? Do other parameters need adjustments to be used with Dmelted?

Considering the aforementioned literature in your major comments, $\mu$ is now set to 0, a characteristic value for snow aggregates. Moreover, the affected Fig. 7, Fig. 8 and Fig. 9 are now reproduced.

Lines 460-465: You did not take into account how the usage of AR=1.60 instead of AR=1.67 affects your results. Briefly comment on this, provide some estimates.

Thank you for this comment. In the latest version of LUTs we changed the AR values so that we include 1.67 for oblates (corresponding to the typical 0.6 from Hogan et al., 2012) in our LUTs. This is also changed in the text as well as in the affected plots.

Figure 17: You should add a panel for sphericity, median mass diameter, and ice water content retrieved with the initial aggregate density (for easier comparison) and reflect the results in Table 4.

A panel for the initial effective density of Yang et al. (2000) is now added in Fig. 17 and indeed helps the reader to directly have a look on the differences of the retrieved parameters. Moreover, RMSE results for DWR, ZDR and Ze using the initial aggregates m(Dmax) are added in Table 4. Thank you a lot for suggesting this.

References:
Gunn, K. L. S., and J. S. Marshall, 1958: The distribution with size of aggregate snowflakes. J. Meteor., 15, 452–461.
Sekhon, R. S., and R. C. Srivastava, 1970: Snow size spectra and radar reflectivity. J. Atmos. Sci., 27, 299–307.Lo, K. K., and R. E. Passarelli Jr., 1982: Growth of snow in winter storms: An airborne observational
study. J. Atmos. Sci., 39, 697–706.
Mitchell, D. L., R. Zhang, and R. Pitter, 1990: Mass-dimensional relationship for ice particles and the influence of riming on snowfall rates. J. Appl. Meteor., 29, 153–163.
Field, P. R., and A. J. Heymsfield, 2003: Aggregation and scaling of ice crystal size distributions. J. Atmos. Sci., 60, 544–560.
Tiira, J., D. N. Moisseev, A. Von Lerber, D. Ori, A. Tokay, L. F. Bliven, and W. Petersen, 2016: Ensemble mean density and its connection to other microphysical properties of falling snow
as observed in southern Finland. Atmos. Meas. Tech., 9, 4825–4841.
Matrosov, S., and A. Heymsfield, 2017: Empirical relations between size parameters of ice hydrometeor populations and radar reflectivity. J. Appl. Meteor. Climatol., 56, 2479–2488.

References:

Hogan, R. J., Tian, L., Brown, P. R. A., Westbrook, C. D., Heymsfield, A. J., and Eastment, J. D.: Radar Scattering from Ice Aggregates Using the Horizontally Aligned Oblate Spheroid Approximation, J. Appl. Meteorol. Clim., 51, 655–671, https://doi.org/10.1175/JAMC-D-11-074.1, 2012.

Yang, P., Liou, K. N., Wyser, K., and Mitchell, D.: Parameterization of the scattering and absorption properties of individual ice crystals, J. Geophys. Res.: Atmos., 105, 4699–4718, https://doi.org/10.1029/1999JD900755, 2000.

**Answers to Reviewer 2**

**General comments:** This is a tough paper to review. On one hand the measurement setup and collected dataset is very interesting. On the other hand, there are several things that causes concerns.

I understand the problem the authors are facing and that is behind one of the weak points in the study. The authors are using the "soft spheroid" particle model in combination with T-matrix to compute single-scattering ice particle properties that are used for the retrievals. However, given a large number of articles published in the last decade that argue that such approach could lead to significant errors, this argument is not easy to make. It is not impossible, since one could argue that 35 GHz measurements are not suffering from the "non spheroidal" effect that much. However, the authors tried to avoid the existing literature and use not well supported arguments, see detailed comments below. I suggest that they improve this part of the paper. Please use more up-to-date literature and make your arguments using the current understanding of the problem.

Thank you a lot for your comment. We agree with you that our ice model is a weak point in this study. In this study, however, the authors major intension was to investigate the feasibility two combine two spatially separated radars to obtain microphysics information about the ice hydrometeors detected into clouds, above the melting layer. The combination of two radars located in different areas is a challenging task due to the different radar scanning geometries, time and spatial mismatch of the radar beams and different path integrated attenuation (PIA) along the radar beams. In our feasibility study we aim to examine the combination of radar measurements from a weather and a cloud radar within a common retrieval framework for particle size, mass and shape. For this purpose, we develop an ice microphysical retrieval for the estimation of the median size, the apparent shape and the ice water content of the detected ice populations making some a-priori assumptions about their particle size distribution and mass-size relation that they follow. Hence, to avoid the addition of more assumptions in the developed ice retrieval, we selected a simple ice particle model, i.e., the soft spheroid, which is known to require the least assumed parameters compared to more complex ice particle models.

We now dedicated a new subsection (1.2 *Representation of ice atmospheric hydrometeors using spheroids*) to include more up-to-date literature and to make a more convincing argument for the use of the simple soft spheroid approximation.

1)  The revised manuscript now includes the following paragraph with a literature review:

*"Single scattering simulations are an indispensable tool to bridge the gap between microphysical properties of hydrometeors and polarimetric radar observations. In the case of ice particles, however, the calculation of scattering properties can be challenging due to their large complexity, variety in shape, structure, size and density. One of the most sophisticated methods, the Discrete-Dipole Approximation (DDA, Draine and Flatau, 1994), can be used to calculate the scattering properties of realistic ice crystals and aggregates. However, this approximation can be computational demanding. To reduce computation cost and complexity, ice particles are often assumed to be spheres and their scattering properties are calculated using the Mie theory or they are assumed to be spheroids using the T-Matrix method (Waterman, 1965) or the Self-Similar Rayleigh-Gans Approximation (SSRGA, e.g., Hogan and Westbrook, 2014; Hogan et al., 2017; Leinonen et al., 2018a) for scattering simulations. The calculations when SSRGA is used are known to be affected by the way that ice mass is distributed throughout the particle's volume. As we aim for a simple ice particle model, we extensively used the T-Matrix method in this study, assuming the ice particles to be soft spheroids. It is a common approach in model studies that ice particles are represented by homogeneous spheroids with density equal or smaller of bulk ice. Due to its simplicity, the limitations of the spheroid approximation have been a heavily researched and debated topic in the last decade. While Tyynelä et al. (2011) showed an underestimation of the backscattering for large snowflakes, Hogan et al. (2012) suggested that horizontally aligned oblate spheroids with a sphericity of 0.6 can reliably reproduce the scattering properties of realistic ice aggregates which are smaller than the radar wavelength. Nevertheless, the same study also*

*concluded that for larger particles spheroids are an improvement to Mie spheres which can lead to a strong underestimation of $Z_e$ and, in turn, strong overestimation of IWC. Leinonen et al. (2012) showed that the spheroidal model cannot always explain the radar measurements as more sophisticated particle models do, e.g., snowflake models. Later on, Hogan and Westbrook, (2014) indicated that the soft spheroid approximation underestimates the backscattered signal of large snowflakes (1 cm size) – measured with a 94 GHz radar – up to 40 and 100 times for vertical and horizontal incidence, respectively. In contrast, the simple spheroidal particle model could successfully explain measurements of slant-45° linear depolarization ratio, SLDR, as well as SLDR patterns on the elevation angles (Matrosov, 2015) during the Storm Peak Laboratory Cloud Property Validation Experiment (StormVEx). In Liao et al. (2016) it was found that randomly oriented oblate ice spheroids could reproduce scattering properties in Ku- and Ka-band similar to these from scattering databases when large particles were assumed to have a density of 0.2 g cm$^{-3}$ and a maximum size up to 6 mm. Furthermore, although Schrom and Kumjian (2018) showed that some ice crystal shapes as branched planar particles could be better represented by plate crystals than spheroids, the simple spheroidal model has been used in recent studies to represent ice aggregates as in Jiang et al. (2019) or to retrieve shape from LDR as in Matrosov (2020). In all these studies, it is recognized that the spheroidal model requires less assumed parameters compared to more complex particle models."*

2) We also wrote a new paragraph to support our decision to use the soft spheroid approximation for this study nonetheless. You can find our answer to one of your more specific comments further below.

The second problem is related to how reliable the retrieved values are. Because of the measurement setup the radar observations volumes are mismatched. Potentially because of this, the observed DWR and retrieved Dm values show artificial looking patterns. The authors use the retrievals to generate statistics of particle properties, see Fig 12, which is one of the main stated goals of the study. I would like to see a discussion what retrieved values can be trusted and why. Ideally, problematic data should be excluded.

Thank you for pointing this out. The different locations of the two radars are an opportunity and challenge at the same time as we aim to obtain an oblique perspective on the cross-section area between both instrument. For this reason, we conducted a sub-study investigating all possible sources of errors and uncertainties in our measurement dataset, i.e., radar reflectivity, differential reflectivity and dual-wavelength ratio. An extended sub-section can be now found in our Sect. 3.1.2:

*"3.1.2 Assessment of radar observations errors*

[revised manuscript text omitted]

Therefore, only measurements fulfilling the aforementioned criteria are considered for statistical results of the ice microphysics retrieval. For visualization purposes in the case study, we only applied the ice mask and noise/error filter using grey colors, and left later filtered regions due to ZDR error. In the following figures we demonstrate the effect of the ZDR filter, with IWC results shown throughout the manuscript (left panel) and after the filter application of the ZDR noise filter (right panel). Only the valid values of the right plot are these considered in the statistics of this paper.

[Figure]

Finally, there are statements in the manuscript that are not correct. A good example is the definition of the reflectivity factor. These should be corrected.

Thank you for your correction. These formulas are now better described as:

*"The radar reflectivity factor z is defined as the sixth moment of the particle size distribution $N(D)$ and is thus designed to be proportional to the to the Rayleigh scattering cross section of small – size much smaller comparing to the radar wavelength – liquid spheres:*

$$z \, [mm^6 \, m^{-3}] = \int_0^\infty N(D)D^6 dD \qquad (1a)$$

*where,*
*z: the radar reflectivity in linear scale,*
*N: the number concentration,*
*D: the geometric diameter of the particles.*
*This formula can be also expressed in logarithmic terms:*

$$Z \, [dBZ] = 10 \cdot log_{10}(z). \qquad (1b)$$

*This definition, however, cannot be directly applied to snow due to the varying density, the irregular shape and larger size of ice particles which cause deviations from the Rayleigh into the Mie scattering regime. Moreover, $N(D)$ for ice particles is referring to the size distribution of their melted diameters. Nevertheless, an equivalent radar reflectivity factor $Z_e$ can be derived from the measured radar reflectivity $\eta$ ($\eta = \sum_{Vol} \sigma_n$; normalized to a specific volume summation of backscattering cross-section of all detected hydrometeors) when the dielectric factor of water $|K|^2 = 0.93$ is assumed:*

$$z_e [mm^6 m^{-3}] = \frac{\lambda^4}{\pi^5 |K|^2} \cdot \eta \ and \ Z_e [dBZ] = 10 \cdot log_{10}(z_e) \qquad (1c)$$

*where:*
*λ: the radar wavelength. In the Rayleigh regime, the radar reflectivity factor Z or the equivalent radar reflectivity factor $Z_e$ (for simplicity reasons referred also as radar reflectivity in this paper) is proportional to the sixth power of the particle size, while in the Mie regime $Z_e$ scales with the second power of the particle size. In both regimes $Z_e$ scales linearly with the particle number concentration."*

Overall, in my opinion the manuscript needs significant improvements before it can be considered for the publication. I encourage the authors to revise it significantly and resubmit.

**Specific comments:**
Line 30 " Ice clouds can cause a cooling effect at the surface by reflecting the shortwave, incoming solar radiation but they can also contribute to warming of the atmosphere by trapping the longwave, terrestrial radiation (Liou, 1986). "

Are you sure about this statement? To my knowledge, ice clouds have a net warming effect. It is possible that in some particular cases they would lead to cooling, but the warming effect is more common.

*Thank you a lot for this comment. The sentence is now rephrased providing more general information about the role of ice clouds in the Earth's energy budget:*

> *"Ice clouds are known to reflect the shortwave, incoming solar radiation, but they can also trap the longwave, terrestrial radiation interfering to the Earth's energy budget (Liou, 1986). Their influence on the radiation budget of the climate system strongly depends on their top height as well as on ice crystals habits and effective ice crystal size (Zhang et al., 2002)."*

Line 49-50: "Another way to gain microphysics information is to use multi-frequency radar observations as they exploit the scattering properties of ice particles in both Rayleigh and Mie regime."

Strictly speaking, Mie regime is not a correct term, a better one is the resonance scattering regime or non-Rayleigh scattering. The Mie solution is only applicable to spherical droplets.

*Thank you for pointing this out.* The sentence is now rephrased as "*Another way to gain microphysics information is to use multi-frequency radar observations as they exploit the scattering properties of ice particles in both Rayleigh and non-Rayleigh regime.*"

Equation 1a – This equation is derived assuming small (much smaller than the wavelength) spherical water droplets.

*The phrase "small – size much smaller comparing to the radar wavelength –" is now added to the sentence describing Eq. 1a.*

Line 64-65 "Similarly, the equivalent radar reflectivity factor Ze can be calculated when the radar reflectivity η is measured as well as the dielectric factor of water |ð    ¾|2 = 0.93 and Rayleigh scattering is assumed: "

This is incorrect. Please check the definition of the radar reflectivity factor and equivalent reflectivity. Also, what is η, you have not defined it.

*Thank you a lot for your comment. The sentence is now rephrased and the definition of measured radar reflectivity η is now added.*

Line 96, Straka's paper is not presenting a method to distinguish hydrometeor types but summarizes characteristics of hydrometeors and corresponding dual-polarization radar signatures. There are many papers that actually present the method.

*Thank you, that sentence is now rephrased and states that Straka et al. (2000) had summarized the characteristics of atmospheric hydrometeors at a wavelength of 10 cm.*

Line 107 "Snow hydrometeors with size in the millimeters order of magnitude are found to have low densities and therefore, the Self-Similar Rayleigh-Gans Approximation (SSRGA, e.g. Hogan and Westbrook, 2014; Hogan et al., 2017; Leinonen et al., 2018a), that is applicable for "soft spheres", can be used. "

Actually, the exact opposite is argued in (Hogan et al., 2017; Leinonen et al., 2018a) and many other studies i.e. by Kneifel et al. (2015). SSRGA is introduced exactly because we cannot use the "soft sphere or spheroid" approximation. SSRGA takes into account internal distribution of ice particle mass that in its turn affect RGA's form-factor and therefore the scattering properties. "Soft-sphere or spheroid" approximation assumes that mass is uniformly distributed, and inclusions are much smaller than the wavelength and hence an effective media approximation, such as Maxwell Garnett, can be used. These approximations seem to fail at mm-wavelengths.

Thank you for this valuable comment. After an extended research on that topic, we corrected our statement. We have now made clear the difference between the Self-Similar Rayleigh-Gans Approximation and PyTMatrix. This part of the text is now modified as follows:

*"One of the most sophisticated methods, the Discrete-Dipole Approximation (DDA; Draine and Flatau, 1994), can be used to calculate the scattering properties of realistic ice crystals and aggregates. However, this approximation can be computational demanding. To reduce computation cost and complexity, ice particles are often assumed to be spheres and their scattering properties are calculated using the Mie theory or they are assumed to be spheroids using the T-Matrix method (Waterman, 1965) or the Self-Similar Rayleigh-Gans Approximation (SSRGA; e.g., Hogan and Westbrook, 2014; Hogan et al., 2017; Leinonen et al., 2018a) for scattering simulations. The calculations when SSRGA is used are known to be affected by the way that ice mass is distributed throughout the particle's volume. As we aim for a simple ice particle model, we extensively used the T-Matrix method in this study, assuming the ice particles to be soft spheroids."*

Line 110: " A well-proven approach is the soft spheroid particle model which uses the effective medium approximation (EMA) to model the refractive index of ice crystals and aggregates, e.g. the Bruggeman or Maxwell-Garnett models as in Garnett and Larmor (1904). "

What are the assumptions and for which conditions EMA are proven to work? This is a blank statement implying that EMA always work regardless of conditions.

Thank you for this comment. More information about EMA along with references is now added in Sect. 3.2.1 *Soft spheroid model*. In particular, the following paragraph is now added:

*"Our soft spheroid model uses the effective medium approximation (EMA) to model the refractive index of the composite material as an ice matrix with air inclusions following the Maxwell-Garnett (MG) mixing formula given in Garnett and Larmor (1904):*

$$\frac{e_{eff}-e_i}{e_{eff}+2e_i} = f_i \frac{e_i-e_m}{e_i+2e_m} \tag{4}$$

*with,*
$e_m$, $e_i$: the permittivities of the medium and the inclusion, respectively,
$e_{eff}$: the effective permittivity,
$f_i$: the volume fraction of the inclusions.

*The complex refractive index, m, is then calculated from $m = \sqrt{e_{eff}}$. In the framework of the EMA, the electromagnetic interaction of an inhomogeneous dielectric particle (components with different refractive indices) can be approximated with one effective refractive index of a homogeneous particle (e.g., Liu et al., 2014; Mishchenko et al., 2016). In Liu et al. (2014), internal mixing was proven to best represent the scattering properties of hydrometeors. Here, the refractive index is modelled as an internal mixing of ice with air inclusions which are arranged throughout the ice particle. The same work also pointed out that the size parameter $D_{crit} = \frac{\pi d}{\lambda}$ for each of these air inclusions should not be larger than 0.4 (with d as the diameter of the inclusion). "*

Line 112 : "Many studies, e.g. Hogan et al. (2012), have …"

There are many studies after that which argue the opposite. As I have mentioned above the development of SSRGA was motivated by the inability of a "soft spheroid" particle

model to reproduce the observations. You should use more recent literature to argue the point and make more convincing argument.

Thank you for pointing out this deficiency in the old version of the manuscript. As we already mentioned to one of your general remarks, we now included more recent literature in a new, dedicated subsection (Sect. 1.2). We would like to answer your second remark to make a more convincing argument why we use the soft spheroid approximation together with your next remark:

Line 122 and up to line 116 "Moreover, using spheroids we can better understand the ambiguities between the aforementioned degrees of freedom. Here, more sophisticated models of specific ice crystals could introduce additional challenges to sort a collection of ice shapes along these degrees of freedom or to define variables like the aspect ratio."

The aspect ratio and orientation angle are part of the "soft-spheroid" particle model. These parameters are irrelevant for a more complex particle representation. One should keep this in mind, while interpreting observations and using different particle models for such interpretations. If you have selected "soft-spheroid" as your model of a more complex ice particle, then you should expect that your model is a (possibly over) simplified representation. Whether this is an advantage or not, it is a matter of discussion. So please make a stronger argument of your point?

Thanks for stressing this point! The new Sect. 1.2 *Representation of ice atmospheric hydrometeors using spheroids* also includes a paragraph in which we make a stronger argument why we use the soft spheroid approximation instead of a more complex particle representation:

> *"Although more complex ice particle and scattering models are available, this work will use the soft spheroid approximation out of the following reasons: (1) In this work we aim to provide a feasibility study to combine two spatially separated radars to better constrain the ice crystal shape in microphysical retrievals using simultaneous DWR and ZDR observations from an oblique angle. Besides instrument coordination, the actual measurements and the assessment of measurement errors, the ice crystal and scattering model are just one component. Due to its simple and versatile setup, this work will utilize the soft spheroid approximation to study the benefit of additional ZDR measurements and the role of the observation geometry. (2) More importantly, to our knowledge, the more accurate SSRGA described by Hogan and Westbrook (2014) does not (yet) provide polarimetric variables used in this study, namely the ZDR. (3) In anticipation of a prognostic aspect ratio of ice crystals in bulk microphysical models (e.g., the adaptive habit prediction; Harrington et al., 2013), we aim to keep a minimal set of degrees of freedom to remain comparable with these modelling efforts. (4) Using ice spheroids we are able to vary parameters such as median size, aspect ratio and ice water content independently, which serve as degrees of freedom of the ice spheroids, and calculate their optical properties without much computational cost as in other scattering algorithms (e.g., DDA) that are used in more realistic ice crystal shapes simulations. Moreover, using spheroids we can better understand the ambiguities between these simple, aforementioned degrees of freedom."*

**Aspect ratio**, page 8: Please use commonly used definitions of AR. At the moment, it is very confusing.

Thank you for this comment. The text is now modified and contains aspect ratio (AR) and sphericity ($S$) definitions. Particularly, it has been updated as:

> *"For the scattering simulations we assumed that ice hydrometeors can be represented by ice spheroids. The shape of the particles is defined using the aspect ratio parameter. In PyTMatrix this is the ratio of the horizontal to rotational axis of the particle. From the description of the simulated ice spheroids in Fig. 6, it is obvious that oblate (shaped like lentil) and prolate particles (shaped like rice) have AR larger and lower than 1.0, respectively, as z*

*axis is selected to be the rotational axis. Using this principle, the representative value of sphericity = 0.6 for oblate ice spheroids from Hogan et al. (2012) is calculated as AR = 1.67 for oblate ice particles in this study and therefore, this number was used as a reference value for the simulation plots (Fig. 7, Fig. 8 and Fig. 9a). In this study, we used S additionally to AR to compare retrieval results when the oblate and prolate shape assumption is used. S for oblates and prolates is found to be smaller than 1, while for spheres is equal to 1."*

Line 276 "Furthermore, small oscillations out of this plane with a standard deviation of 20° are included to consider the flutter of ice crystals. "

How? Please explain what you mean.

Thank you for your point. After your comment we have now replaced this sentence with the following part:

*"Here, all prolate ice particles were assumed to fall with their maximum diameter aligned to the horizontal plane. Hence, all ice prolates (hereafter referred as horizontally aligned prolates or horizontally aligned prolate ice spheroids) are rotated 90° (mean canting angle) in the yz plane (Fig. 6), while ice oblates have a 0° mean canting angle. The canting angle, i.e. angle between the particle's major dimension and the horizontal plane, of the falling hydrometeors has been the topic of several studies. This value in nature is not so easy to estimate and thus, a standard deviation (e.g., 2°–23° as in Melnikov, 2017) is often additionally used. Here, we used a fixed standard deviation of 20° to describe the oscillations of the particles maximum dimension around the selected canting angle. Then the calculation of the scattering properties is performed using an adaptive integration technique for all possible particle's geometries, ignoring the Euler angles alpha and beta of the scattering orientation."*

Line 304, equation 6: What was the motivation of using the melted snowflake diameter?

Thank you for this comment. In Eq. 6, $D_{eq}$ was only used as an interim parameter to obtain an $m(D_{max})$ relationship since Yang et al (2000) only provided a fit for an equivalent volume sphere. The formula is now modified as:

*"... The $D_{eq}$ is used to describe the diameter of a spherical water particle with the same mass as an ice particle with maximum dimension $D_{max}$.*

$$m(D_{max}) = \frac{\pi \cdot \rho_w \cdot D_{eq}^3}{6} = \frac{\pi \cdot \rho_w}{6} e^{\sum_{n=0}^{4} b_n (\ln(D_{max}))^n \, 3} \qquad (7)$$

*where $b_n$ is taken from Table 2 in Yang et al. (2000), the water density $\rho_w$ = 1 g cm$^{-3}$ and $D_{eq}$ as well as $D_{max}$ are in microns."*

Page 11. Look-up table structure

Given that you are using the "soft spheroid" particle model and T-matrix to compute singlescattering ice particle properties, I was expecting a discussion on how the refractive index is defined for different AR values. Are you preserving particle mass or density (and therefore the refractive index)? This should be explicitly discussed. What are PSD integration limits used in calculations? How did you select the maximum Dvalue? The selection of maximum D has a direct influence on Zdr. Is that the reason why you are having an issue with reproducing Zdr observations?

Thank you for your comment on this topic. In our study a soft spheroid is a homogeneous mixture of ice and air, it's ice mass is evenly distributed all over the spheroid's volume and EMA is used for the refractive

index calculation. As already quoted to your question regarding EMA above, we now included a more detailed description how we calculate the refractive index. To answer your question here, we are preserving particle mass and not density / refractive index when increasing the AR until reaching the bulk density of ice. Then particle mass is clipped. Regarding integration limits: for our work we a-priori chose the maximum diameter of PSD (D_max) to be 20 mm, as we are interested to retrieve ice microphysics for ice particles that are detected within the cloud and above the melting layer and not for large aggregated snowflakes that reach the ground. In the following plot we calculated ZDR and DWR for AR = 1.67, shape parameter of PSD $\mu = 0$ and median mass diameter, $D_m$, to vary between 0.1–3.02 mm.

[Figure]

We show that for even larger D_max, i.e., D_max = 25 mm, the simulated ZDR-DWR don't differ from the results using D_max = 20 mm. Minor differences in ZDR and DWR are observed when D_max is selected to be lower, i.e., 15 mm.

The ice spheroids construction is now added in the text (Sect. 3.2.1 *Soft spheroid model: Mass-size relation*). In particular:

> *"The maximum dimension, $D_{max}$, and the sphericity values for the spheroids were a-priori defined and their mass was calculated according to the formula that describes the relation between mass and $D_{max}$ (mass-size relation), providing information about the mass of the ice crystals and therefore, their effective density with respect to their size. Mass m of an ice particle is usually connected to its maximum diameter $D_{max}$ with a power-law formula,*
>
> $$m(D_{max}) = a D_{max}^b \qquad (5)$$
> *where,*
> *a: the prefactor of the $m(D_{max})$, refers to the density scaling at all particles sizes,*
> *b: the exponent of the $m(D_{max})$, relates to the particles shape and growth mechanisms. With the mass and the spheroid dimensions known, the density of the ice spheroid was calculated. In the special case when the density was found to exceed the density of solid ice (0.917 g cm⁻³), the mass of the spheroid was clipped and its density was set equal to the ice density. With the ice spheroid mass known, we also calculated the mass of the bulk ice spheroid (same-dimensions spheroid with density of solid ice) and from the ratio mass$_{sph}$/mass$_{bulk\_sph}$ we calculated the ice fraction of the simulated spheroid. Using the ice fraction, we then calculated the particle's refractive index from the Maxwell-Garnet mixing formula."*

Fig 11, page 15 How physical are high Dm values closer to the cloud top? Please explain what data can be trusted. Ideally you should mask questionable data. This affects the results presented in Fig. 12.

Thank you for pointing this out. In the latest version of our manuscript in the already mentioned Sect. 3.1.2: *Assessment of radar observations errors* as well as in our Appendix A: *Radar measurements error assessment* you

may find a detailed description of our approach to consider only reliable measurements in our ice microphysics retrieval and thus, for the development of Fig. 12 (current Fig. 14). As we already mentioned to one of your previous answers, the ZDR noise filter is not yet applied to the case study plots shown in Fig. 12 which would remove most of these large Dm values closer to cloud tops. For the statistical results shown in Fig. 14, we applied all described filters.

Page 17. Unknown mass-size relationship

Please explain, what was the logic behind the selection of the m(D) relations. Are they representative of the events you have observed, i.e. represent particles observed in this temperature regime? Are they sufficiently different to cover a possible range of values? On lines 290  300 you just state that you use them without much discussion why.

You are right, the motivation for BF and Yang was not really described until now. As we laid this out as a feasibility study of the measurement combination, we aimed for two sufficiently different m(D) relationships to get a better understanding of the limitations when soft spheroids are used for that task. Since BF95 was heavily used in previous studies we started off with their m(D) to represent more 2D ice crystals (b=1.9) resulting in soft spheroids of lower density. To contrast them, we choose the m(D) analog to the aggregates from Yang et al (2000) which result in soft spheroids with a higher and constant density. The corresponding introduction now reads:

> *"While the effective density of a spheroid decreases strongly with its size due to the exponent b=1.9 in BF95, we contrast this with a second $m(D_{max})$ with a higher and constant density. To that end we borrowed the $m(D_{max})$ from the irregular aggregate model from Yang et al. (2000) to create soft spheroids with an analog mass-size ratio."*

Our choice to continue the study with this m(D) can also be understood when answering your next question:

"The BF95 mass-size relation was found to model to low ice particle densities … BF95 prescribes near zero density values for large particles. This leads to very low simulated ZDR values with increasing particles size (Fig. 6)"

Hogan et al (2012) was able to reproduce observed Zdr values. Could you please explain what is the difference between their and your study?

Simulations using the BF95 mass-size relation and the typical AR = 1.67 in current Fig. 8 (previously Fig. 6) show that BF95 cannot reproduce our radar dataset. Especially for large particles the ZDR signal is very low due to the low density. Although we can achieve higher ZDR values like in Hogan et al (2012) when spheroids with higher density are used, these spheroids are then so compact that they cannot simultaneously achieve large DWR values. With our soft spheroids we can reproduce ZDR simulations similar to Fig. 5 from Hogan et al., 2012 for S-band and different axial ratio and ice fraction values. For the typical AR = 1.67 (axial ratio 0.6), the maximum DWR for $D_{max}$ = 1 mm was, however, not larger than 0.45 dB when ice fraction was 0.2.

[Figure]

To conclude, we further used the aggregates mass-size relation after a radiative closure study from Ewald et al. (2021) where it was shown that aggregate habit reconciled better the lidar, radar, and solar radiance measurements against columns or plates. We also noticed that the aggregates could better explain our ZDR-DWR measurement space than the well-known BF95. Hence, we continued our ice studies using the aggregates mass-size relation from Yang et al. (2000). In this way we managed to obtain simultaneously larger ZDR and DWR that better match our radar measurements.

Fig. 16 page 20. I think this is the most interesting plot of the paper. I am not sure how practical this is, but the difference between a and b panels indicates that there is extra information that can be retrieved by using different measurement geometries. Very interesting.

Thank you for your interest in this plot (current Fig. 18). Our intention was indeed to show the different radar geometries effect on the retrieved parameters. Moreover, the valuable contribution of ZDR in different areas of the cloud cross-section is highlighted in this plot. For very low radar elevation angles from both radars, e.g., at the lowest part of the cloud cross-section, and also above MIRA-35, i.e., when POLDIRAD is scanning and MIRA-35 is pointing to zenith, ZDR can not only provide apparent shape information but also helps to narrow down the solution space for the size retrieval.

References

[revised manuscript text omitted]

**Answers to Dr. Jana Mendrok**

This study introduces a method for the retrieval of snow microphysical properties (ice water content, mean size, shape) from polarimetric and dual-wavelength radar observations. The method as introduced here is based on a bunch of rather rude assumptions, in particular the shape model (homogeneous spheroids) and the mass-size relation. In my view, the study lacks to show the effect that these assumptions have on the retrieval results and to properly state the limitations of the method.

Thank you for your comment about our manuscript. As already mentioned in the paper, but maybe it needed to be more emphasized (and it is now), this work serves as a feasibility study exploring the combination of two spatially separated radars to derive microphysics information about the detected – in the radar beams cross-section – atmospheric hydrometeors. Our major focus is to ensure that we are able to obtain high quality dual-wavelength ratio measurements and as a further extension of this work, to use the radar measurements combined with a simple ice particle model and assumptions for the particle size distribution (PSD) and mass-size relation (m(Dmax)) of the ice particles, to develop an ice microphysics retrieval. Using spheroid as a particle model, we are then able to test our a-priori assumptions for the ice particles as its simplicity allows for easy calculations of the particles mass (estimated from the mass-size relation) and thus, effective density. Therefore, we are able to investigate the effect of our guesses for the particles shape, size and mass on the retrieved parameters (apparent shape as well as median size and ice water content of the ice particles PSD), using the soft spheroid model to represent the detected ice hydrometeors.

After a more up-to-date literature review about the limitations of the soft spheroid approximation we now included a more detailed argumentation why we still decided to stick to this rather simple model:

> *"Although more complex ice particle and scattering models are available, this work will use the soft spheroid approximation out of the following reasons: (1) In this work we aim to provide a feasibility study to combine two spatially separated radars to better constrain the ice crystal shape in microphysical retrievals using simultaneous DWR and ZDR observations from an oblique angle. Besides instrument coordination, the actual measurements and the assessment of measurement errors, the ice crystal and scattering model are just one component. Due to its simple and versatile setup, this work will utilize the soft spheroid approximation to study the benefit of additional ZDR measurements and the role of the observation geometry. (2) More importantly, to our knowledge, the more accurate SSRGA described by Hogan and Westbrook (2014) does not (yet) provide polarimetric variables used in this study, namely the ZDR. (3) In anticipation of a prognostic aspect ratio of ice crystals in bulk microphysical models (e.g., the adaptive habit prediction; Harrington et al., 2013), we aim to keep a minimal set of degrees of freedom to remain comparable with these modelling efforts. (4) Using ice spheroids we are able to vary parameters such as median size, aspect ratio and ice water content independently, which serve as degrees of freedom of the ice spheroids, and calculate their optical properties without much computational cost as in other scattering algorithms (e.g., DDA) that are used in more*

*realistic ice crystal shapes simulations. Moreover, using spheroids we can better understand the ambiguities between these simple, aforementioned degrees of freedom."*

The resulting limitations are now also mentioned more prominently throughout the discussion and we included the following paragraph to mention the limited scope of this work already in the introduction:

*"Due to this simplification, this study will focus on the feasibility to combine DWR and ZDR from spatially separated radar instruments into a common retrieval framework. Due to the missing internal structure of soft spheroids, the known underestimation of the radar backscatter and generally lower ZDR for larger snowflakes will limit this study to ice aggregates with sizes in the millimeter regime. This will include the onset of ice aggregation within clouds above the melting layer (ML) but will exclude heavy snowfall close to the ground. Anyhow, this region is rarely included in the measurement region with an overlap between the two scanning radar instruments."*

To detail and justify my concerns:

The authors write (L588) "we have to assume a suitable m(Dmax) relationship" and I cannot agree more. The authors obviously define "suitable" from the agreement of their scattering calculations with statistics of observations of DWR and ZDR. The scattering calculations are based on a range of assumptions, including e.g the size distribution, the mass-size relation, the spheroidal particle model, the shape of the spheroids.

The paper gives a first glimpse that the results of the ice retrieval are definitely affected by the different assumptions that we use in our ice scattering simulations performed by T-Matrix (e.g., Mischenko and Travis (1994); Mischenko et al. (1996) and more). In particular, Fig. 8 as well as Sect. 4.3.1 *Unknown mass-size relationship* show differences in the retrieved parameters when the well-known Brown and Francis (Brown and Francis, 1995; hereafter BF95) and aggregates (Yang et al., 2000) m(Dmax) are used. Moreover, the different shape (oblate or prolate) assumption also affects the ice retrieval's results (Sect. 4.1). However, to be more specific on how and how much the a-priori simulation assumptions about m(Dmax), particles oblate or prolate shape, shape parameter $\mu$ of particle size distribution (PSD), wobbling of ice particles etc. can alter the ice retrieval's results, we are already working on a second manuscript which will serve as a sensitivity analysis on the aforementioned assumptions. To be more specific, in the current version of our manuscript we have now stressed out that *the retrieved parameters would be .... under these assumptions…*

Out of these, mass-size relation is rather well constrained by observations of a range of different measurement techniques (including 3D imaging of falling snow "in the wild" (Leinonen et al., 2021)) and its range of variation is comparably small. Moreover, it is a crucial microphysical parameter in weather and climate models.

Thank you for pointing out this concern. As mentioned above, our study is not focused on falling snow close to the ground. While the mass-size relationship of falling snow might be rather well constrained by observations with multi-angle snowflake cameras (MASC) on the ground, the mass size relationship is one of the least constrained and strongly varying ice crystals properties higher up within clouds. The large variability of in-cloud mass-size relation has been found in various in situ studies (e.g., Heymsfield et al., 2010; Xu et al., 2016) and has been stressed in numerous studies as one of the largest sources of uncertainties in ice microphysical retrievals (e.g., Deng et al., 2013; Delanoë et al., 2014; Ham et al., 2017).

On the other hand, the spheroidal particle model is a highly artificial model that is known to benot well suited to represent scattering properties of particles of low effective density like snow aggregates. This regards microwave scattering properties in general (e.g. Eriksson, 2015; Eriksson, 2018), but polarimetric properties in particular (eg. Schrom and Kumjian, 2018). Aspect ratio, in addition, is a highly simplistic parameter to describe the shape of typically irregularly shaped particles. That is, it is

questionable how well aspect ratios observed from irregularly shaped particles can constrain reasonable values for homogeneous soft particles (like spheroids or even plates).

As already mentioned, our intention is to use a simple and easy-handled ice particle model so that we can seamlessly change our assumptions for ice particle populations and check their effect on the ice microphysics retrieval algorithm. Using a more realistic but more complicated ice particle model could lead to additional assumptions in our study. The use of the spheroid model to represent ice particles has been debated in many studies in the past. Therefore, we have now included a part presenting related literature (Sect. 1.2 *Representation of ice atmospheric hydrometeors using spheroids*):

*"Single scattering simulations are an indispensable tool to bridge the gap between microphysical properties of hydrometeors and polarimetric radar observations. In the case of ice particles, however, the calculation of scattering properties can be challenging due to their large complexity, variety in shape, structure, size and density. One of the most sophisticated methods, the Discrete-Dipole Approximation (DDA; Draine and Flatau, 1994), can be used to calculate the scattering properties of realistic ice crystals and aggregates. However, this approximation can be computational demanding. To reduce computation cost and complexity, ice particles are often assumed to be spheres and their scattering properties are calculated using the Mie theory or they are assumed to be spheroids using the T-Matrix method (Waterman, 1965) or the Self-Similar Rayleigh-Gans Approximation (SSRGA; e.g., Hogan and Westbrook, 2014; Hogan et al., 2017; Leinonen et al., 2018a) for scattering simulations. The calculations when SSRGA is used are known to be affected by the way that ice mass is distributed throughout the particle's volume. As we aim for a simple ice particle model, we extensively used the T-Matrix method in this study, assuming the ice particles to be soft spheroids. It is a common approach in model studies that ice particles are represented by homogeneous spheroids with density equal or smaller of bulk ice. Due to its simplicity, the limitations of the spheroid approximation have been a heavily researched and debated topic in the last decade. While Tyynelä et al. (2011) showed an underestimation of the backscattering for large snowflakes, Hogan et al. (2012) suggested that horizontally aligned oblate spheroids with a sphericity of 0.6 can reliably reproduce the scattering properties of realistic ice aggregates which are smaller than the radar wavelength. The same study also concluded that for larger particles spheroids are an improvement to Mie spheres which can lead to a strong underestimation of $Z_e$ and, in turn, strong overestimation of IWC. Leinonen et al. (2012) on the other hand showed that the spheroidal model cannot always explain the radar measurements as more sophisticated particle models do, e.g., snowflake models. Later on, Hogan and Westbrook, (2014) indicated that the soft spheroid approximation underestimates the backscattered signal of large snowflakes (1 cm size) – measured with a 94 GHz radar – up to 40 and 100 times for vertical and horizontal incidence, respectively. In contrast, the simple spheroidal particle model could successfully explain measurements of slant-45° linear depolarization ratio, SLDR, as well as SLDR patterns on the elevation angles (Matrosov, 2015) during the Storm Peak Laboratory Cloud Property Validation Experiment (StormVEx). In Liao et al. (2016) it was found that randomly oriented oblate ice spheroids could reproduce scattering properties in Ku- and Ka-band similar to these from scattering databases when large particles were assumed to have a density of 0.2 g cm$^{-3}$ and a maximum size up to 6 mm. Although Schrom and Kumjian (2018) showed that some ice crystal shapes as branched planar particles could be better represented by plate crystals than spheroids, the simple spheroidal model has been used in recent studies to represent ice aggregates as in Jiang et al. (2019) or to retrieve shape from LDR as in Matrosov (2020). In all these studies, it is recognized that the spheroidal model requires less assumed parameters compared to more complex particle models."*

To summarize, mass-size relation is a well-constrained parameter, while aspect ratio is not and is a rather artificial parameter. Hence, I wonder (and question), why the authors chose to use a mass-size relation that is unreasonable for snow aggregates but insist on keeping the aspect ratio within "reasonable" bounds.

We value your critique. As mentioned above we do not imply that our approach will yield reasonable results for large snowflakes. This work was designed as a feasibility study to combine simultaneous DWR and ZDR measurements within a common retrieval framework. While people have used these joint observables to constrain or rule out specific ice crystal shapes during their discussions in the past, to our knowledge, this study is the first attempt to combine both observables in a microphysical retrieval. To that end we employed two strongly different mass-size relationships to test and to understand the interrelationship between DWR and ZDR for a very simple and intuitive ice crystal model.

As I understand, the results of the retrieval method presented here strongly depend on the choice of mass-size relation. This poses the question how reliable retrieved microphysical properties (primarily IWC) are when selecting a relation that is far off the range of commonly accepted values. As stated above, I miss a comprehensive estimate of the uncertainties (or, actually, errors) that this unreasonable assumption results in – not within the (forward modelling-retrieval) system here, which is self-consistent, but in more realistic retrievals (if independently measured IWC are not available, e.g. by analyzing the retrieved IWC based on a range of for different m(Dmax) in the retrieval).

Using the same assumptions and testing two different m(Dmax) we find that ice spheroids with mass-size relation corresponding to aggregates from Yang et al. (2000) can explain better our radar observations against BF95, especially for larger ice particles and thus, higher DWR values, e.g., our current Fig. 16, also attached below. Panels (a) and (b) present residual values between the measured and the simulated DWR when ice spheroids are oblates, follow exponential PSD and have a mass-size relation corresponding to aggregates and BF95, respectively.

[Figure]

In particular, we write:

> "*Figure 16 shows the residuals between the simulated and measured DWR for aggregates (Fig. 16a) and BF95 $m(D_{max})$ (Fig. 16b). For ice spheroids that follow the $m(D_{max})$ of aggregates, the residuals are evenly distributed around 0 (mean value of +0.08 dB) suggesting that this mass-size relation can better explain our measurements in this case. In contrast, the measured DWR appeared to be higher than the simulated one for BF95 for the larger part of the cloud cross-section (reddish areas) with a mean value of –0.923 dB*"

A similar study is also conducted in our manuscript testing mass-size relations similar to that of aggregates which considers and almost constant effective density with the size ($\sim 0.2$ g cm$^{-3}$). The retrieval results are shown in our current Fig. 17, also attached here.

[Figure]

For a better evaluation of the retrieved parameters, i.e., IWC, and as we do not have auxiliary data from January 2019 to evaluate the ice retrieval results, we used snapshots from the MODIS MYD06 product (source: https://worldview.earthdata.nasa.gov/, Platnick et al, 2017) for ice water path (IWP). In the following figure snapshots for Oberpfaffenhofen (location of POLDIRAD weather radar) and Munich (location of MIRA-35 cloud radar) are presented. Snapshots of the colorbar indicating the respective IWP value for each location are also shown.

[Figure]

The figure shows a gradient of the retrieved MODIS IWP which decreases from west to east. Therefore, an averaged value of IWP ~ 90 g m$^{-2}$ was considered from MODIS for the whole radar cross-section. Using our retrieved IWC for the three cases (0.5x, 1x, 2x aggr $\rho_{eff}$) and integrating with height we obtain IWP ~ 46 g m$^{-2}$, IWP ~ 83 g m$^{-2}$  and IWC ~ 137 g m$^{-2}$ when the effective density is considered 0.5x, 1x, 2x times of aggr $\rho_{eff}$, making the 1x aggr $\rho_{eff}$ the mass-size relation which can best explain our radar measurements for that scene. In all cases, the ice particles were assumed oblates and that they follow an exponential PSD.

The introduction to the discussion section points out two a priori assumptions on the particle properties as limitations of the method: the already discussed mass-size relation and the choice of particle model between oblate and prolate spheroids, missing to point out the much more crucial assumption of spheroids in general. This continues in the discussion of "unsuitability" of the BF95 mass-size relation, where no other reasons for the very low ZDR than the low density, seemingly exclusively resulting from the m(Dmax) assumption is discussed, which is re-iterated in the conclusions.

Thank you very much for this comment. The BF95 mass-size relation has been widely used in ice cloud studies before (e.g., Hogan et al., 2012 and many more) yielding good results in comparison to observations. Investigating how suitable the BF95 is, we notice that using BF95, which is known to

predict low density for large particles when combined to ice spheroids, we cannot produce high ZDR signals but only for very small and dense particles, and not for less dense and larger ice particles with larger DWR values. Therefore, the "unsuitability" of this mass-size relation when ice spheroids are used lies in the fact that it cannot explain our ZDR measurements due to the homogeneity that the spheroid model suggests and due to the missing sharp edges of the ice aggregates, which are known to give high ZDR, when these are considered to be approximately represented by a spheroid. This explanation is now added in the current version of our manuscript.

As source for their aggregate model, the authors cite Yang et al. (2000). I find this highly misleading. Yang et al. (2000) (as well as a range of follow-up papers building on it and extending it, including Hong et al., 2009, and Ding et al., 2017) targets the explicit modeling of scattering properties of irregularly shaped particles. The irregular, non-spherical and(!) non-spheroidal(!) shape is the crucial aspect of their shape model, the core element of that research. The authors of this study, however, reduce this complexity to the minor aspect of the underlying mass-size relation (L356: "another m(Dmax) that we use is the aggregates from Yang et al. (2000)", falsely implying the equivalence of the m(Dmax) with the entire Yang et al. aggregate definition). Mass-size relation is likely the most un-aggregate-ish characteristic of the Yang aggregate model, was – probably – selected with not much care at that time (is that even originating from Yang et al. themselves, or where has it been taken from by Yang et al.?), and has been pointed out as a shortcoming of the Yang-aggregates by other authors (e.g. Eriksson, 2018).

Thank you for pointing out this misleading wording. We indeed only adapted the mass-size relationship corresponding to aggregates from Yang et al. (2000) to construct corresponding soft spheroids. You are right that the aggregates were initially designed as a specific aggregation of 8 hexagonal elements (Yang and Liou, 1998) for which an m(D) was only fitted afterwards (Eq. 12 with parameters from Table 2 in Yang et al., 2000). In numerous studies, however, their microphysical (e.g., Baum et al., 2005) as well as their scattering properties (e.g., Eichler et al., 2009; Ewald et al., 2021) have turned out to be a versatile tool to explain remote sensing data from ice clouds. Moreover, after reading Eriksson et al. (2018, or 2015?), we could not find the mentioned shortcoming of Yang-aggregates. On the contrary, their triple-frequency signature (DWR$_{Ku,Ka}$ vs DWR$_{Ka,W}$) in Fig. 13 (Eriksson et al., 2018) seem to be quite capable to reproduce also radar measurements, e.g. compare Fig. 10 in Kulie et al. (2014). In this work, we chose this m(D) relationship with a higher and constant ice crystal density as a contrast to the lower-density m(D) of BF95.

To avoid the misleading confusion between the m(D) with the entire Yang et al. aggregate definition, we changed the wording throughout the manuscript and revised the paragraph which introduce this second m(D) relationship:

> *"While the effective density of a spheroid decreases strongly with its size due to the exponent b=1.9 in BF95, we contrast this with a second $m(D_{max})$ with a higher and constant density. To that end we borrowed the $m(D_{max})$ from the irregular aggregate model from Yang et al. (2000) to create soft spheroids with an analog mass-size ratio. Originally, the construction of these aggregates was fully described in Yang and Liou (1998) as an aggregated collection of geometrical hexagonal columns. In our study, this second soft spheroid model only emulates the maximum dimension and mass of the underlying aggregates."*

To justify reference to Yang, it would be interesting to see how the scattering properties of the authors' aggregates compare to microwave properties of the actual, irregularly shaped aggregate of Yang. As far as I am aware, they are not available for preferential orientation, but both Ding et al. (2017) and Eriksson et al. (2018) provide scattering properties for this habit ("8-column aggregate") at radar wavelengths for totally randomly orientation, which should allow for a comparison of the predicted reflectivities and dual-wavelength ratios.

Thank you for this comment. After your interesting suggestion, we compared microphysical properties of our ice spheroids that follow the aggregates mass-size relation assumption from Yang et al. (2000) to aggregates from the Atmospheric Radiation Measurement (ARM) scattering database (Lu et al., 2016). In the following plots we attach some of the comparison results. Both figures show

comparisons between soft spheroids (dots in the scatter plot) and low-density ARM aggregates (LD-P1d, crosses in the scatter plot).

[Figure]

In the left plot we used the soft spheroids analog to the m(D) of aggregates from Yang, while we used the soft spheroids from our study with double the density of m(D) from Yang for the right plot. With the doubled density, the agreement between T-matrix soft spheroids and ARM aggregates is obviously better. In the first case, with the original effective density, the soft spheroids produce 1 dB lower ZDR than these simulated with the generalized multiparticle Mie method (GMM) in ARM. We are planning to continue these kind of studies in our next paper (also including the suggested database from Ding et al. (2017) and Eriksson et al. (2018)), which is already in preparation. In case of interest, we could already include such a comparison in our manuscript Appendix.

Minor comments:

L376: Could you provide a reference for that Dm from the equation you provide is the Median mass diameter? In my understanding, when D therein is the melted mass equivalent diameter, then Dm is the mean mass diameter. Are median and mean really equal here?

The present formula indeed referred to mean mass diameter. Thank you for pointing this out. The definition has been changed now to $\int_0^{Dm} mN(D)dD = \frac{1}{2}IWC$ from e.g., Ding et al. (2020).

L384: Please specify more precisely, what Maxwell-Garnett approach was used, in particular what material forms the matrix, what the inclusions, as this can make a lot of a difference (see e.g. Eriksson, 2015).

In this work, ice spheroids were assumed to have been formed with the inclusion of air in the medium of ice. We also added a new paragraph in the introduction (Sect. 3.2.1 *Soft spheroid model: Refractive index*) describing the way that these particles are formed. In particular we write:

*"Our soft spheroid model uses the effective medium approximation (EMA) to model the refractive index of the composite material as an ice matrix with air inclusions following the Maxwell-Garnett (MG) mixing formula given in Garnett and Larmor (1904):*

$$\frac{e_{eff}-e_i}{e_{eff}+2e_i} = f_i \frac{e_i-e_m}{e_i+2e_m} \qquad (4)$$

*with,*
*$e_m$, $e_i$: the permittivities of the medium and the inclusion, respectively,*
*$e_{eff}$: the effective permittivity,*
*$f_i$: the volume fraction of the inclusions.*
*The complex refractive index, m, is then calculated from $m = \sqrt{e_{eff}}$. In the framework of the EMA, the electromagnetic interaction of an inhomogeneous dielectric particle (components with different refractive indices) can be approximated with one effective refractive index of a homogeneous particle (e.g., Liu et al., 2014; Mishchenko et al., 2016). In Liu et al. (2014),*

*internal mixing was proven to best represent the scattering properties of hydrometeors. Here, the refractive index is modelled as an internal mixing of ice with air inclusions which are arranged throughout the ice particle. The same work also pointed out that the size parameter $D_{crit} = \frac{\pi d}{\lambda}$ for each of these air inclusions should not be larger than 0.4 (with d as the diameter of the inclusion). "*


Data availability

Ice water paths were taken from the MODIS MYD06 product with the identifier doi:10.5067/MODIS/MOD06_L2.061 (Platnick et al, 2017)

---

## Referee Report (RR1)

Review of amt-2021-216

The manuscript "Retrievals of ice microphysics using dual-wavelength polarimetric radar observations during stratiform precipitation events" is improved from its previous version. A minor revision is recommended.

Specific comments:

Line 90: Define $\sigma_n$.

Line 162: Add "differential" to specific propagation phase (KDP – the name is usually defined as specific differential phase).

Line 172-173: You use capital $Z_H$ and $Z_V$ (which are common for a log scale), whereas you use linear $z_H$ and $z_V$ in the definition, eq. (3). Try to be consistent with the definition.

Lines 210-211: The sentence is somewhat confusing, reformulate it.

Line 255: The word "this" is repeated, fix the typo.

Lines 1062-1064: This sentence is unclear.

Lines 1073-1074: The right parenthesis/bracket ")" at the end of the sentence is missing.

---

## Editor Decision (ED1)

Decision on amt-2021-216

Dear authors,
Thank you for submitting your revised manuscript and your responses to the reviewers.
After consideration I am again asking for minor revisions. Specific points to address are below.

Regards,
Raquel Evaristo.

Comments from reviewer 4 were responded carefully, however, I noticed there was little or no effort in including the points raised in the manuscript.

In particular, concerning the comments numbered 2 and 4, the authors took a lot of effort to respond to the reviewer, but none of that was included in the document. The reviewers comments are supposed to be a way to improve your manuscript, and not merely a response to certain personal doubts.

Very specifically, when responding to comment 2. you say that "This advantage has been stressed out in a lot of parts of our manuscript, e.g., lines 26-29 or 886-889", but these two instances are describing results. I believe the reviewer meant that you should formulate clearly the advantage of your approach. Ideally this should be included in the introduction as a motivation point for this study (your section 1.3), while comparing and stressing the advantages over other approaches.

Then include the response to comment 4 in your discussion, effectively showing you compare your calculation with the calculation from other methods and obtain a result that seems to be more accurate (at least according to your comparison with the IWP from MODIS, for example). Ideally in situ observations should be taken for validation. Would be good to add a comment on this point.

Same with comment 6. Lines 684-693 discuss the error propagation and how it affects Dm but never mention how it affects IWC. Your response to this comment, and particularly the end, acknowledging that this point needs sensitivity studies to improve the methodology in the future should be included in the paper.

Minor comments:
Lines 32-33:
" terrestrial radiation interfering to the Earth's evergy budget" replace with:
" terrestrial radiation interfering with the Earth's evergy budget"

Line 55: Include references concerning dual frequency.

Line 57: "to the" is repeated

Line 110: replace "constrained" with "constrain"

Line 123: Include a reference where the readers can find information on polarimetric variables.

Line 234: replace "resulting to a single radar cross-section" with:
"resulting in a single radar cross-section"

Line 236: The sentence is not constructed properly. Maybe add "In another approach…"

Line 240:  replace "This approach can result nine…" with:
"This approach can result in nine…"

Line 359: Appendix B is mentioned before Appendix A. Switch the Appendixes accordingly so that A comes first.

Lines 376 -380: You are using an event from the 4[th] of April 2019 to find the calibration bias for events in January 2019. How can you be sure of the stability of the calibration more than 3 months later?

Line 401: I suspect that here it should be "calculate DWR errors" instead of "calculate DWR values". Please check.

Line 418 (legend of figure 5): "the POLDIRAD and MIRA-35 spatial mismatch" is repeated.

Line 559: replace "Combining the PSD and with the m(Dmax)" with "Combining the PSD with the m(Dmax)"

Figure 11 a: Adjust the color scale between reasonable values for your variability.

Line 812: This formulation "once with twice…" is confusing.
Suggest: "...for oblate ice particles, 1) with twice and 2) with half the density…"

Line 949 Legen of figure A1: panel c) shows a scatterplot of the median Zdr vs height. Why are there only a few points? How were these points selected?

Lines 965 to 974, Appendix B:
There are polarimetric algorithms available for melting layer detection designed for RHIs. One example is Wolfensberger et al. 2016. This discussion could be shorter if you just mentioned other optimized methodologies could be used instead of your hard thresholds.

Wolfensberger, D., Scipion, D. and Berne, A. (2016), Detection and characterization of the melting layer based on polarimetric radar scans. Q.J.R. Meteorol. Soc., 142: 108-124.
https://doi.org/10.1002/qj.2672

Line 974: "only few ice hydrometeors were detected above the 0°C isotherm". This does not make sense to me. Please check.

---

## Author Response (AR2)

Review of amt-2021-216  (authors reply to Anonymous Referee #2)

The manuscript "Retrievals of ice microphysics using dual-wavelength polarimetric radar observations during stratiform precipitation events" is improved from its previous version. A minor revision is recommended.

Dear Reviewer,

On behalf of the co-authors of this manuscript I would like to thank you for your review. Our replies to your comments can be found below in blue color. Changed sentences in our manuscript are written in italics.

Thank you a lot once again for your feedback.

Kind regards,
Eleni Tetoni

Specific comments:
Line 90: Define σ$_n$.
Thank you for this comment. The phrase (line 70) is now changed to: "$\eta = \sum_{Vol} \sigma_n$; normalized to a specific volume summation of backscattering cross-section, $\sigma_n$, of all detected hydrometeors".

Line 162: Add "differential" to specific propagation phase (KDP – the name is usually defined as specific differential phase).
Thank you for pointing this out. "differential" is now added to – currently – line 123.

Line 172-173: You use capital ZH and ZV (which are common for a log scale), whereas you use linear zH and zV in the definition, eq. (3). Try to be consistent with the definition.
Thank you we have now fixed that typo (lines 132–133).

Lines 210-211: The sentence is somewhat confusing, reformulate it.
Thank you for this comment. After reformulating the sentence, it now reads: "*The same study also concluded that, spheroids are more suitable to represent larger particles (maximum diameter up to 2.5mm) in simulations rather than Mie spheres can, as the latter can lead to a strong overestimation of $Z_e$*" (lines 168–170).

Line 255: The word "this" is repeated, fix the typo.
We have now fixed that, thank you (line 217).

Lines 1062-1064: This sentence is unclear.
After rephrasing, the sentence is now changed to: "*Statistical results of the retrieved S, $D_m$ as well as IWC are presented in Fig. 14*. For these results it is assumed that *ice hydrometeors are represented by ice spheroids following an exponential PSD and with $m(D_{max})$ corresponding to this of aggregates.*" (lines 724–726).

Lines 1073-1074: The right parenthesis/bracket ")" at the end of the sentence is missing.
A bracket is now added, thank for pointing this out (line 743).

*All line numbers refer to the file with track changes.

Manuscript ID: acp-2021-216 (amt-2021-216, authors reply to Anonymous Referee #4)

Review of "Retrievals of ice microphysics using dual-wavelength polarimetric radar observations during stratiform precipitation events"

Dear Reviewer,

On behalf of all co-authors I would like to thank you for your time to review our manuscript and for the valuable feedback you provided to us. In the following lines you can find our replies to your comments/suggestions written with blue color. New or changed sentences/paragraphs in our manuscript are quoted in italics.

Thank you a lot once again.

Kind regards,
Eleni Tetoni

General Comment

This is my first review of this manuscript, while the manuscript has been revised accounting for comments from the reviews in the previous review process. The manuscript proposed a novel approach for IWC and Dm retrievals that utilize multi parameters available for observation using two polarimetric Doppler radars. Application of this technique is limited to vertical slices of one location only. The authors sincerely revised the manuscripts, and the technique proposed in this study has been evaluated carefully. There are a few points that the authors should address before publication.

Specific comments

1. I suggest changing the title to "Retrievals of ice microphysical properties….". First I saw the current title, I thought that the study discussed microphysical processes such as particle growth processes in cloud. However, this manuscript focused more on retrievals of IWC, Dm, etc., which are microphysical properties.

Thank you very much for this suggestion. We have now changed the title of the manuscript to: *"Retrievals of ice microphysical properties using dual-wavelength polarimetric radar observations during stratiform precipitation events"*.

2. What is the advantage of this method over the conventional method such as Z-only and Z+polarimetric variable-only retrievals? I agree that the DWR could be an additional constraint, but this study did not demonstrate comparisons.

To our opinion, our approach has multiple advantages. First of all, in a possible Z-only retrieval we should keep in mind that the radar reflectivity can be sometimes affected by non-Rayleigh/Mie effects especially at higher-frequency radars (e.g., Ka- band) resulting to lower values than Z measured by lower frequency radars (e.g., C-, X-band). However, we exploit this difference in radar reflectivities by using DWR to infer information about the size of the atmospheric hydrometeors. DWR has been used so far in many conventional size retrievals, usually by making an a-priori assumption of the ice particles shape, e.g., specific value of aspect ratio assumed to retrieve the particles size. In our approach, next to the DWR which is used to infer size information, polarimetry variables, i.e., ZDR, obtained from a scanning radar can be efficiently used to provide shape

information, especially when the other scanning radar is pointing upwards. This advantage has been stressed out in a lot of parts of our manuscript, e.g., lines 26–29 or 886–889. It is common that multi-wavelength techniques are applied to vertically pointing radars. This option though, doesn't allow for polarimetric measurements as e.g., oblate ice particles would appear like spheres. On the contrary, our scanning radars can provide such an observations combination making the simultaneously size and shape retrieval of the detected hydrometeors, possible (see also our new Fig. 19). Furthermore, in our study we prove that two spatially separated radars can be combined to provide ice microphysics information. Therefore, operational weather radars located throughout Germany can be used in synergy with already established cloud radar sites to monitor precipitation but also to obtain microphysical properties of atmospheric hydrometeors.

3. Figure 19: I do not think that this image represents a natural situation properly. This image seems to assume that there is a single ice particle in a sampling volume and presents only one radar beam incidence at a single point of the particle. However, the radar observables represent ensembles of PSD and scattering by multiple incident points.

Thank you for pointing this out. We have now created a new version of Fig. 19 (also attached below). The radars setup is also visible in our new plot showing how the radar beams detect ice hydrometeors populations which are assumed to be soft spheroids. We then zoom in in the spheroids ensembles and use a single oblate spheroid to which represents the average aspect ratio of the whole spheroids PSD to describe the effect of the different radar geometries on the apparent shape retrieval.

[Figure]

4. I understand that using a cost function is effective to reduce errors. But, I wonder whether the results from this technique can be consistent with the theory-based retrievals. For example, was the aspect ratio retrieved in this study consistent with that from Matrosov's et al. (2017) technique? Was the IWC retrieved in this study consistent with that from Bukovcic's et al. (2018) technique?

Thank you for this comment. Studying the suggested literature, we proceeded with some comparison plots for the ice water content retrieval.

Using our C-band radar KDP (and not S-band as the literature suggests) along with Ze for the presented – in the manuscript – case study, we calculated the IWC from the Bukovčić et al. (2018) formula: $IWC(KDP, Ze) = 0.71KDP^{0.65}Ze^{0.28}$. The results are shown below. Our retrieval shows lower IWC values using oblate spheroids than the theoretical formula.

[Figure]

To further evaluate both results we calculated the ice water path (IWP) in both cases and compared it to IWP data from MODIS MYD06_L2 product (Platnick, S., Ackerman, S., King, M. et al., 2017). From MODIS, an averaged value of IWP ~ 90 g m$^{-2}$ was retrieved for the whole radar cross-section. Using our retrieved IWC and integrating with height we obtain IWP ~ 83 g m$^{-2}$ , while we obtain 3622 g m$^{-2}$ when retrieved IWC from Bukovčić et al. (2018) was used.

Deeper investigation of this difference shows that for Bukovčić's retrieval of IWC$_{Bukovčić}$ = 2 g m$^{-3}$ at 1 km, approximately 5 times smaller particles (melted equivalent size is meant here) would be needed to explain the observed reflectivity of e.g., $Z_e$ ~ 15 dBZ.  More precisely for the retrieved IWC at 1 km, IWC$_{Bukovčić}$ = 2 g m$^{-3}$ and IWC$_{Tetoni}$ = 0.02 g m$^{-3}$. Assuming particles size 1 mm for our retrieved IWC and solving $Z_e = ND^6$ for particle size, we obtain a melted equivalent size ~ 200 microns that could explain the IWC$_{Bukovčić}$ retrieval. As Bukovčić's retrieval includes no information about particle size, we consider our IWC to be closer to reality since we retrieve particles size before the IWC.

To answer how consistent our aspect ratio retrieval is, we compared our scattering calculations to Matrosov et al., 2017 their Fig. 3b (attached below). Comparing the simulation lines we observe differences between our calculations and those of Matrosov. For BF95 mass-size relation when the standard deviation would be 3° and the median volume diameter = 0.02 cm, ZDR = 1 dB would give aspect ratio = 0.55 in Matrosov's approach, while the retrieved aspect ratio according our calculations would be 0.45 (i.e., 20% lower values). Differences between our AR-ZDR simulations and Matrosov could arise from the way that a mixture of air-ice is considered in the mixing formula used to calculate the refractive index, i.e., an ice matrix with inclusions of air or an air matrix with inclusions of ice, of the soft spheroid. In our study *"Our soft spheroid model uses the effective medium approximation (EMA) to model the refractive index of the composite material as an ice matrix with inclusions of air following the Maxwell-Garnett (MG) mixing formula given in Garnett and Larmor (1904)"* (lines 439–441), while for Matrosov approach we couldn't spot this information. In our manuscript we mainly used the aggregates m(Dmax) from Yang et al., 2000. In this case, ZDR = 1 dB would give an aspect ratio of ~0.38 when median volume diameter = 0.02 cm. In general, for our

both mass-size relation assumptions, the retrieved aspect ratio is lower than this retrieved using Matrosov's approach.

[Figure]

[Figure]

5. Was the attenuation A calculated from the attenuated reflectivity and Zdr values? If so, how can the uncertainty impact the IWC and Dm retrievals?

During our scattering simulations we performed specific attenuation *A* calculations with respect to the three degrees of freedom (mass, size, shape) used for our soft spheroids. Therefore, this parameter was calculated during the retrieval using the first step retrieval's output (Fig. 10) and the indices that minimized the two cost functions for $Z_e$, ZDR and DWR in our 3D LUTs, and not estimated by empirical relations from literature. The specific attenuation for our presented case study was retrieved relatively small as our ice spheroids were considered to be dry (we consider mainly dry ice particles in our analysis) and with low-density. This resulted to small values of total attenuation at both radar bands (see our Fig. S2, in our submitted supplementary material) and in turn, to almost no difference in the correction of radar reflectivities and thus, in the retrieved parameters. However, we are aware that attenuation from wet snow can be significant, especially for our Ka-band radar. In this case, the largest part of DWR would origin from attenuation effects, resulting to large uncertainties in size retrieval.

6. A region of relatively large Dm in Fig. 11b seems to correspond to a region of estimated DWR error due to the volumetric mismatch error (Fig. 5f). Are the Dm values intrinsic? Can they be used for the IWC retrieval?

Thank you for your question. All retrieved parameters are described along with their errors originated by the thoroughly analyzed different sources (Sect. 3.1.2 of our manuscript). For instance, we attach the Dm as well as IWC results along with their errors, highlighting the area that you indicated. The errors for IWC are indeed large in that area, too. However, we consider the Dm values intrinsic. The DWR error is calculated considering two components. The spatiotemporal and the beam width differences in the measured – from the two radars – volume. In the presence of strong

gradient of DWR originated from microphysics, one has to expect large DWR errors. Our error estimation method, in the case of the very thin ~500 m layer around the altitude of 2.5 km, considers this mismatch to origin from the volume mismatch. In these areas the retrieved ice microphysical properties should be carefully treated. In a possible next version of this algorithm we can exclude areas where the calculated DWR error exceeds some limits (also suggested in lines 931–933 of our manuscript). The selection of such criteria needs in depth sensitivity studies but for sure will improve the performance of our retrieval.

[Figure]

7. Provide brief case description for each case to see what kind of snow events this technique was applied to.

Thank you for your comment. Throughout the main body of our manuscript we use one case study to demonstrate our methodology and 59 case studies in total to derive statistics. We have now added a brief meteorological description of the presented case in lines 307–311. The addition is the following: "*At 04:00 UTC of that night, an ice cloud started forming at an altitude of 9 km. During the time of our coordinated measurements the cloud's vertical extension was up to 7 km. Throughout that day, the ambient temperature was mostly below 0°. The wind speed at the surface was very low, while at higher altitudes exceeded 15 m s⁻¹ at some cases. The vertical gradient of the wind favored the development of fall streaks (also shown in our radar observations in Fig. 3) and thus, ice particle growth within the ice cloud.*"

8. This technique can be applied to only one vertical cross section (i.e. RHIs directed toward only one azimuth). Please discuss how this can be used to observe microphysical processes and general characteristics of clouds.

Thank you for pointing this out. A small discussion is now added in lines 233–241. In particular we write: "*Our approach considers single RHI scans from each radar instrument resulting to a single radar cross-section. In the special case when the wind direction in this area is aligned to our radars cross-section, we can monitor the evolution of precipitation and the development of fall streaks inside*

*the clouds by performing continuous RHI scans according to the precipitation rate. Another approach to deeply investigate the initiation of convection as well as to better observe ice microphysical processes in clouds, in a separate study, we performed sector range-height indicator (S-RHI) using POLDIRAD and MIRA-35 to monitor precipitation cells during convection. In this way, a first scan was executed towards the cell of interest at a specific azimuth. Then, two additional fast RHI scans were executed from each radar deviated ±2° from the initial azimuth. This approach can result nine vertical profiles within the precipitation cell providing additional microphysical information (Köcher et al., 2022, their Fig. 1).”*

Technical comments

1. Fig. 11c, Fig. 12a, and Fig.12c look exactly same. Why?

Figure 11c (sphericity), Fig. 12a (aspect ratio) and Fig. 12c (sphericity) referring to shape results for prolate ice spheroids are indeed identical. This originates from the aspect ratio definition (AR=horizontal to rotational axis; lines 456-457 and Fig. 6) which results aspect ratio values <1 for prolate soft spheroids. The sphericity $S$ is defined as the minor-to-major axis ratio (line 167) and in the case of prolate ice spheroids it is found <1. For oblate ice spheroids the sphericity is derived from the reverse of the AR value (see also Fig. 6). Moreover, in line 583–584 we state that for our calculations we use *“0.125, 0.16, 0.21, 0.27, 0.35, 0.45, 0.6, 0.8, 1.0) for the horizontally aligned prolates and the inverse values for the oblate particles”*, and resulting the same $S$ values for both shape assumptions.

E.g.,

Oblate: $S = 0.6$ , AR = 1.67

Prolate: $S = 0.6$, AR = 0.6

2.Please provide scan speeds for each radar observation.

Thank you for this comment. We have now added information about the scan speed of the two radars in line 278–279 (POLDIRAD; elevation velocity 1°/s) and 290 (MIRA-35; elevation velocity 4°/s).

3. Table 2: Provide sample size. How many RHI slices were available for each case?

Thank you. We have now added this information in Table 2 as well as Table 4.

4. How was the DWR from different radar coordinates taken? Were the reflectivity data gridded? If so, how were the data gridded?

The way that DWR was obtained using two radars at different locations is described in lines 322–331. The method how we interpolated the radar data is now added in these lines.
*“During the snow events, $Z_e$ measurements from the two radars were performed and interpolated, using the nearest-neighbor interpolation method, onto a common rectangular grid (50 × 50 m$^2$). The 0-height of this grid is defined to be the height above MSL, while POLDIRAD and MIRA-35 locate at 602.5 m and 541 m height above MSL. In Fig. 3a and Fig. 3c, the measured $Z_e$ from the two radar systems during the RHI scans from 30$^{th}$ January 2019 at 10:08 UTC is presented. For the MIRA-35 $Z_e$ measurements we applied a calibration offset of 4 dBZ as derived in Ewald et al. (2019). Studying only snow cases no strong effects of hydrometeor attenuation are expected (e.g., Nishikawa et al., 2016). However, an iterative method to estimate hydrometeor attenuation has been developed.*

*Additionally, both $Z_e$ datasets are corrected for gaseous attenuation using the ITU-R P.676-11 formulas provided by International Telecommunication Union (ITU) in September 2016 (ITU-R P.676, 2016). Both methods are fully described in Sect. 3.3. After the interpolation of both radar reflectivities in the common radar grid, we calculated the DWR (Fig. 3b) using Eq. (2)."*

5. Did you use weights for each term in the cost function?

Thank you for this comment. At the beginning we used weighting factors and one cost function for our ice microphysics retrieval. Therefore, the mass, size and shape of the ice particles were retrieved at once. This selection of the weighting factor values was not objective though and one could reasonably argue about these numbers. Moreover, the residuals between the simulated and measured values for the three different parameters, i.e., radar reflectivity, dual-wavelength ratio and differential radar reflectivity had not the same units and thus, they were not compatible. Hence, we proceeded without the use of weights, but instead of normalized differences, and we additionally introduced one more cost function. In this way we proceeded with the retrieval of the size and shape simultaneously, as we noticed that the DWR and ZDR remain the same for different values of IWC. At the second step we retrieved the mass, i.e., IWC, of the ice hydrometeors. (the invariance of DWR and ZDR can be found in Fig. 9 as well as in lines 589–592)

*All line numbers refer to the file with track changes.

---

## Author Response (AR3)

**Decision on amt-2021-216**

Dear Editor,

Thank you a lot for your effort and comments on our manuscript. In the following lines you may find our replies written with blue color. The indicated lines refer to the manuscript version with track changes.

Thank you for your time.

Kind regards,
Eleni Tetoni

Dear authors,
Thank you for submitting your revised manuscript and your responses to the reviewers.
After consideration I am again asking for minor revisions. Specific points to address are below.
Regards,
Raquel Evaristo.

Comments from reviewer 4 were responded carefully, however, I noticed there was little or no effort in including the points raised in the manuscript.
In particular, concerning the comments numbered 2 and 4, the authors took a lot of effort to respond to the reviewer, but none of that was included in the document. The reviewers comments are supposed to be a way to improve your manuscript, and not merely a response to certain personal doubts.

Very specifically, when responding to comment 2. you say that "This advantage has been stressed out in a lot of parts of our manuscript, e.g., lines 26-29 or 886-889", but these two instances are describing results. I believe the reviewer meant that you should formulate clearly the advantage of your approach. Ideally this should be included in the introduction as a motivation point for this study (your section 1.3), while comparing and stressing the advantages over other approaches.
Thank you a lot for your suggestion. We have now added a new paragraph in Sect. 1.3 mentioning the advantages of our approach. (Lines: 238–247)

Then include the response to comment 4 in your discussion, effectively showing you compare your calculation with the calculation from other methods and obtain a result that seems to be more accurate (at least according to your comparison with the IWP from MODIS, for example). Ideally in situ observations should be taken for validation. Would be good to add a comment on this point.
Thank you for your comment. We have now added a new section in our discussion part which includes parts of our replies to this referee comment 4. You can find this new section in lines: 959–988.

Same with comment 6. Lines 684-693 discuss the error propagation and how it affects Dm but never mention how it affects IWC. Your response to this comment, and particularly the end, acknowledging that this point needs sensitivity studies to improve the methodology in the future should be included in the paper.

We have now included a new Fig. 13 showing the Dm and IWC error considering the calibration and beam width as well as spatiotemporal mismatch errors. We also added some new lines in the text including information to this respect (Lines: 728–749). Thank you for pointing this out.

Minor comments:

Lines 32-33:
" terrestrial radiation interfering to the Earth's evergy budget" replace with:
" terrestrial radiation interfering with the Earth's evergy budget"
"To" is now replaced with "with". Thank you. (Line: 37)

Line 55: Include references concerning dual frequency.
Thank you for your suggestion. We have now included literature using dual-wavelength methods. (Lines: 59)

Line 57: "to the" is repeated
The sentence is now corrected. (Line: 61)

Line 110: replace "constrained" with "constrain"
The typo is now fixed, thank you. (Line: 115)

Line 123: Include a reference where the readers can find information on polarimetric variables.
Thank you for your suggestion. We have now included two references giving a description about the polarimetric radar variables mentioned in the text. (Line: 126–127)

Line 234: replace "resulting to a single radar cross-section" with:
"resulting in a single radar cross-section"
The typo is corrected. (Line: 225)

Line 236: The sentence is not constructed properly. Maybe add "In another approach…"
Thank you for pointing this out. We have now improved the sentence according your suggestion. (Lines: 232)

Line 240: replace "This approach can result nine…" with:
"This approach can result in nine…"
Thank you. The phrase is now changed as you suggest. (Line: 236)

Line 359: Appendix B is mentioned before Appendix A. Switch the Appendixes accordingly so that A comes first.
Although Appendix A is mentioned in parenthesis in Section 3.1 (Line: 357), it indeed comes before Appendix B (first mentioned in Line: 375). However, we have now removed the parenthesis in line 357 so that the reference to Appendix A to be more obvious.

Lines 376 -380: You are using an event from the 4th of April 2019 to find the calibration bias for events in January 2019. How can you be sure of the stability of the calibration more than 3 months later?
Thank you for your comment. We have now added some lines advocating the use of the same calibration bias in ZDR. (Lines: 391–398)

Line 401: I suspect that here it should be "calculate DWR errors" instead of "calculate DWR values". Please check.

Thank you for pointing this out. We have now corrected the indicated phrase. (Line: 418)

Line 418 (legend of figure 5): "the POLDIRAD and MIRA-35 spatial mismatch" is repeated.

Unfortunately we couldn't spot the repetition of "the POLDIRAD and MIRA-35 spatial mismatch". However, we now rephrased the sentence deleting the second "the POLDIRAD and MIRA-35" part. (Line: 459)

Line 559: replace "Combining the PSD and with the m(Dmax)" with "Combining the PSD with the m(Dmax)"

We have how rephrased the sentence. Thank you a lot. (Line: 605)

Figure 11 a: Adjust the color scale between reasonable values for your variability.

Thank you for pointing this out. We have now replaced Fig. 11.

Line 812: This formulation "once with twice…" is confusing.
Suggest: "...for oblate ice particles, 1) with twice and 2) with half the density…"

Thank you a lot for this suggestion. We improved the text accordingly. (Line: 905)

Line 949 Legen of figure A1: panel c) shows a scatterplot of the median Zdr vs height. Why are there only a few points? How were these points selected?

Thank you for your comment. We have now changed lines 394–398 to include information about your questions. Specifically we write: *"In Fig. A1 (Appendix A), examples of radar reflectivity $Z_e$, differential reflectivity ZDR as well as a scatter plot showing the average ZDR offset are presented. The scatters in the last panel (Fig. A1c) indicate the median ZDR value averaged over the full measurement period, shown in Fig. A1a, for each vertical radar bin within the cloud layer. The data were acquired by super sampling the 150 m pulse in 75 m range steps to enhance the signal statistics."*

Lines 965 to 974, Appendix B:
There are polarimetric algorithms available for melting layer detection designed for RHIs. One example is Wolfensberger et al. 2016. This discussion could be shorter if you just mentioned other optimized methodologies could be used instead of your hard thresholds.
Wolfensberger, D., Scipion, D. and Berne, A. (2016), Detection and characterization of the melting layer based on polarimetric radar scans. Q.J.R. Meteorol. Soc., 142: 108-124.
https://doi.org/10.1002/qj.2672

Thank you for your suggestion. We have now included a comment that an already established melting layer detection algorithm (using your suggested literature) could also be used as our thresholds need more investigation and evaluation using more "ML case studies". However, we still keep Fig. B1 as an example for this statement (Lines: 1103–1112).

Line 974: "only few ice hydrometeors were detected above the 0°C isotherm". This does not make sense to me. Please check.

Thank you very much for pointing this out. This sentence was indeed confusing. We have now replaced it with the new sentence *"In our investigated case studies, a ML was never detected and only a very small part of the cloud cross-section was masked using the 0° isotherm at some cases."* (Lines: 1111–1112)

---

## Author Response (AR4)

**Comments to the author**:

Decision on Retrievals of ice microphysical properties using dual-wavelength polarimetric radar observations during stratiform precipitation events

By Eleni Tetoni, Florian Ewald, Martin Hagen, Gregor Köcher, Tobias Zinner, and Silke Groß
Submitted to AMT

Thank you again for your reviewed manuscript, which is again improved compared to the previous version.

There is still one problem left: Zdr bias may not be stable from January to April, and there is no way to be certain of that unless the bias is checked regularly. The papers you cited do not show stability over periods as long as 3 months. I understand this comes at a late stage in this process, and I apologize for that. However, this needs to be addressed.

Bellow are a few minor points I still found in the manuscript that should be considered.

Dear Editor,

Thank you once again for your time and your comments to our manuscript. In the following lines you can find our replies written with blue color. Regarding your comment about the ZDR bias stability, after studying the suggested book we found Fig. 6.7 where the ZDR bias remains stable for almost a year for the S-band Columbia WSR-88D (0.2 dB). Moreover, Fig.11 in Ryzhkov et al. (2005) shows that ZDR calibration accuracy during JPOLE was estimated about 0.2 dB. After your recommendations, we created a ZDR histogram following Ryzhkov's approach and the results are shown below as well as in the supplementary of this manuscript. We have also added the following lines to our Sect. 3.1.2:

*"To further ensure the stability of ZDR bias, an additional calibration validation was conducted following the Ryzhkov and Zrnic (2019) approach (described in their Sect. 6.2.4). Our measurement dataset from January 2019 was filtered for large Ze regions and intermediate temperatures for dry and large aggregates. This analysis yielded a median ZDR = 0.2 dB for these areas, where ice aggregates are expected, indicating that POLDIRAD was well calibrated during the period of this study."*

Thank you for your time and feedback once again.

Kind regards,
Eleni Tetoni

*Ryzhkov, A. V. and Zrnic, D. S.: Data Quality and Measurement Errors. In: Radar Polarimetry for Weather Observations, Springer, Cham., https://doi.org/10.1007/978-3-030-05093-1_6, 2019.*

*Ryzhkov, A. V., Giangrande, S. E., Melnikov, V. M., & Schuur, T. J. (2005). Calibration Issues of Dual-Polarization Radar Measurements, Journal of Atmospheric and Oceanic Technology, 22(8), 1138-1155. Retrieved Jun 8, 2022, from https://journals.ametsoc.org/view/journals/atot/22/8/jtech1772_1.xml*

Line 65: Instead of defining N as the number concentration I think it is more illustrative to define N(D) as the particle size distribution.
Thank you, we corrected accordingly.

Lines 253-254: The acronym was just defined in the previous page, just use PROM afterwards.
Thank you a lot, the acronym was replaced.

Line 348: The title of the subsection 3.1.1 is "Ice mask and noise filters application", but you do not really describe any of the other filters in this section. I suggest that you remove the second part of the title and just call this section "Application of the ice mask".
Thank you for stressing this out. We have now replaced the title considering your suggestion.

Line 515: "and thus, from the mass" should be replaced with "and thus, by the mass".
We now corrected that, thank you.

Line 844: There's a double period mark here.
Double period mark was removed and a typo was also fixed.

Line 847: In the IWC formula using KDP, which was defined to be used at S-band, did you scale the KDP with the wavelength?
Thank you very much for your valuable comment. We haven't scaled our KDP with the wavelength. After doing so, we noticed also differences in our comparisons which were also included in the text.

Line 975: The figure B1c has different colors for different origin of filters, that are not all grey, so remove "grey" in the legend referring to c.
The word "grey" is now removed. Thank you for pointing this out.

Non-public comments to the Author: About the Zdr calibration issue, are there any vertical scans even performed (even in snow) at a closer time (less than 1 month)? Alternatively, could you plot a histogram of Zdr for all the snow events you use in the manuscript and check that you have a reasonable value as expected for aggregates? There is a methodology somehow like this described in (Ryzhkov and Zrnic 2019("Radar Polarimetry for Weather Observations" book, section 6.2.4)), and the expected value should be below 0.2 dB. If you find that your Zdr bias indeed changed compared to April, hopefully not too much, just comment on how that should affect your retrievals. You do not need to do all calculations again.